**1** **Soil bacterial community and functional shifts in response to**

**2** **altered snow pack in moist acidic tundra of Northern Alaska**

**3** **M. P. Ricketts[1], R. S. Poretsky[1], J. M. Welker[2], and M. A. Gonzalez-Meler[1]**

[1] {Biological Sciences Department, Ecology and Evolution, University of Illinois at Chicago,
Chicago, IL, USA}
[2] {Department of Biological Sciences, University of Alaska, Anchorage, AK, USA}
Correspondence to: Michael P Ricketts (rickett4@uic.edu)

## Abstract

Soil microbial communities play a central role in the cycling of carbon (C) in Arctic tundra ecosystems, which contain a large portion of the global C pool. Climate change predictions for Arctic regions include increased temperature and precipitation (i.e. more snow), resulting in increased winter soil insulation, increased soil temperature and moisture, and shifting plant community composition. We utilized an 18-year snowfence study site designed to examine the effects of increased winter precipitation on Arctic tundra soil bacterial communities within the context of expected ecosystem response to climate change. Soil was collected from three pre-established treatment zones representing varying degrees of snow accumulation, where Deep ~100% and Int. ~50% increased snow pack relative to the control, and Low ~25% decreased snow pack relative to the control. Soil physical properties (temperature, moisture, active layer thaw depth) were measured, and samples were analysed for C concentration, nitrogen (N) concentration, and pH. Soil microbial community DNA was extracted and the 16S rRNA gene was sequenced to reveal phylogenetic community differences between samples and determine how soil bacterial communities might respond (structurally and functionally) to changes in winter precipitation and soil chemistry. We analysed relative abundance changes of the six most abundant phyla (ranging from 82-96% of total detected phyla per sample) and found four (Acidobacteria, Actinobacteria, Verrucomicrobia, and Chloroflexi) responded to deepened snow. All six phyla correlated with at least one of the soil chemical properties (%C, %N, C:N, pH), however a single predictor was not identified suggesting that each bacterial phylum responds differently to soil characteristics. Overall bacterial community structure (beta diversity) was found to be associated with snow accumulation treatment and all soil chemical properties. Bacterial functional potential was inferred using ancestral state reconstruction to approximate functional gene abundance, revealing a decreased abundance of genes required for soil organic matter (SOM) decomposition in the organic layers of the deep snow accumulation zones. These results suggest that predicted climate change scenarios may result in altered soil bacterial community structure and function, and indicate either a reduction in decomposition potential, or alleviated temperature limitations on extracellular enzymatic efficiency, or both. The fate of stored C in Arctic soils ultimately depends on the balance between these mechanisms.

# 1 Introduction

Broad and rapid environmental changes are driving both above- and belowground community shifts in the Arctic (Elmendorf et al., 2012a, 2012b; Tape et al., 2006, 2012; Wallenstein et al., 2007). It is well established that soil microbial communities may alter their composition in response to changing environmental factors such as nutrient availability, moisture, pH, temperature, and aboveground vegetation shifts (Lauber et al., 2009; Morgado et al., 2015; Semenova et al., 2015), and ecological and climate induced changes to Arctic soil microbial community structure and function have important effects on ecosystem carbon (C) cycling and nutrient availability for plant growth (Deslippe et al., 2012; Graham et al., 2012; Waldrop et al., 2010; Zak and Kling, 2006). Because many of these environmental features are rapidly changing in Arctic tussock tundra ecosystems (Anisimov et al., 2007; Liston and Hiemstra, 2011), and because of the large amounts of C stored in Arctic soils (Hugelius et al., 2013; Ping et al., 2008; Schuur et al., 2009; Tarnocai et al., 2009), it is imperative to examine microbial responses in this system.

Soil microorganisms play a key role in the decomposition of soil organic matter (SOM) on a global scale, releasing nutrients into the soil and stored C into the atmosphere in the forms of $CO_2$ and $CH_4$, two major greenhouse gases that contribute to global warming (Anisimov et al., 2007). Decomposition of SOM by soil microorganisms amounts to at least half of the 80-90 Gt C released each year by soil respiration, the second largest terrestrial flux after gross primary productivity (GPP; Davidson and Janssens, 2006; Hopkins et al., 2013; Raich et al., 2002). Because global soils contain about 2,000 Gt of C, ~1,500 Gt of which is in the form of SOM (Batjes, 1996; IPCC, 2000), large scale changes in the rate of microbial decomposition will have an impact on the rate at which $CO_2$ accumulates in the atmosphere (Schimel and Schaeffer, 2012).

The decomposition rate of SOM, resulting in heterotrophic respiration from soils ($R_h$), has been shown to be sensitive to temperature and moisture (Davidson and Janssens, 2006; Frey et al., 2013; Hopkins et al., 2012, 2013; Xia et al., 2014). As the Arctic climate warms, increasing $R_h$ may be capable of producing a positive feedback on the climate system as C stored in soils over millennia is released back to the atmosphere (Czimczik and Welker, 2010; Jonasson et al., 1999; Lupascu et al., 2014b; Mack et al., 2004; Nowinski et al., 2010; Shaver and Chapin, 1980, 1986).

Northern latitude permafrost soils may house over 50% of the world's soil organic C (SOC; the C component of SOM), approximately twice the amount of C present in the atmosphere (Hugelius et al., 2013; Ping et al., 2008; Schuur et al., 2009; Tarnocai et al., 2009). In addition, Arctic ecosystems are more susceptible to the effects of climate change, warming at approximately twice the rate as temperate zones and exhibiting increased winter precipitation patterns (Anisimov et al., 2007; Liston and Hiemstra, 2011). Deeper snow has a suite of cascading consequences in tundra ecosystems as snow acts to insulate soil from extreme winter air temperatures resulting in soil temperatures under deeper snow pack up to 10°C warmer than soils under ambient snow depths (Schimel et al., 2004). Altered soil conditions under deeper snow may thus lead to increased SOM decomposition, causing changes in SOC stocks while also releasing nutrients for plant and microbial growth (Anisimov et al., 2007; Leffler and Welker, 2013; Rogers et al., 2011; Welker et al., 2005). The predicted increase in soil temperature as a result of deeper winter snow accumulation should enhance the rate of SOM decomposition by: 1) a direct temperature effect on enzyme kinetics, and 2) by increasing substrate availability to decomposers as the active layer deepens and permafrost thaws (Lützow and Kögel-Knabner, 2009; Nowinski et al., 2010; Schuur et al., 2008). Therefore, warming and deeper snow in the Arctic are likely to expose C stored over millennia to decomposers, resulting in a major source of C to the atmosphere.

However, ecosystem C loss may be offset by increased soil moisture, causing hypoxic conditions and limiting $R_h$ (Blanc-Betes et al. 2016). Also, microbial mineralization of plant nutrients, such as nitrogen (N) and phosphorus (P), from SOM decomposition are likely to contribute to increased net primary productivity (NPP; Hinzman et al., 2005; Natali et al., 2012; Pattison and Welker, 2014) and cause shifts in vegetation from herbaceous species (Cottongrass tussock-*Eriophorum vaginatum*) towards woody species (Arctic shrubs – *Betula nana* and *Salix pulchra*) that may produce a larger amount of plant litter compounds that are more resistant to decomposition (Bret-Harte et al., 2001; Pearson et al., 2013; Sturm et al., 2005; Wahren, 2005). The balance between these processes will determine the extent to which Arctic tundra ecosystems feedback on the global climate, making the fate of this stored C unclear (Sistla et al., 2013).

This study examined changes in soil bacterial community composition due to increased winter snow accumulation and subsequent altered biotic and abiotic factors using a long-term snow

fence manipulation experiment that mimics changes in winter precipitation by creating a gradient
of snow depths from much deeper than ambient to shallower than ambient levels (Jones et al.,
1998; Pattison and Welker, 2014; Welker et al., 2000). We postulated that increased soil thermal
insulation from deeper winter snow accumulation would elicit bacterial community response via:
1) altered soil physical characteristics such as soil temperature, moisture, or $O_2$ availability, and
2) altered soil chemistry produced by increased microbial mineralization of SOM resulting in
increased nutrient availability and changes in plant species composition and litter. Here we
evaluated phylum level shifts in bacterial community phylogeny using 16S rRNA gene analysis
and predicted bacterial functions using the program PICRUSt (Langille et al., 2013) to test
whether increased snow accumulation and associated changes in soil conditions (warmer
temperatures, altered plant inputs, and increased hypoxia) would cause shifts in bacterial
community structure and functional potential that reflect increased SOM decomposition and
nutrient mineralization.
**2    Methods**
**2.1    Site description and sample collection**
The study utilized a long-term snow depth manipulation experiment site (Jones et al., 1998;
Walker et al., 1999) established in 1994 in a moist acidic tundra ecosystem located near Toolik
Lake Field Station, Alaska (68º37'N, 149º32'W). It consists of a strategically placed snow fence
designed to simulate the increased precipitation patterns and continuous snow-cover episodes
predicted under global warming scenarios, resulting in a gradient of increasing snow
accumulation (and thus increasing soil thermal insulation, soil temperatures, and active layer
thaw depth/permafrost thaw) with proximity to the fence. While snowfall varied from year to
year, the wind drift from the fence provided consistent relative snow accumulation at similar
distances from the fence every winter (Fahnestock et al., 2000; Welker et al., 2005). The soil is
classified as Typic Aquiturbel, exhibiting characteristics of cryoturbation and poor drainage
(Ping et al., 1998; Soil Survey, 2015). Four experimental zones were identified according to their
snow accumulation regime: Control ("Control", taken >30m outside the effects of the
snowfence), deep snow ("Deep" ~ 100% increase in snow pack relative to the control),
intermediate snow ("Int.", ~50% increase in snow pack relative to the control), and low snow

("Low", ~25% decrease in snow pack relative to the control; Fig. 1). The Deep snow zone is unique in that it is waterlogged during thaw periods, and dominated not by Cottongrass tussock or woody shrub species (e.g. *Eriophorum vaginatum, Betula nana,* or *Salix pulchra*), but by a sedge species, *Carex bigelowii*. However, the vegetative history of this plot includes a transition from tussock cottongrass to woody shrub species, and finally to wet sedge species (Arft et al., 1999; Walker and Wahren, 2006).

Three replicate soil cores were taken approximately 15-20m apart from each experimental snow zone (totalling 12 soil cores) in August of 2012 and analysed separately. All soil coring equipment was cleaned and sterilized in the field between each sample using water and 100% ethanol. The top 10cm representing the organic layers was taken first using a sharpened steel pipe (5.08cm diameter X 12cm length) and serrated knife to cut through surface vegetation and to minimize soil compaction. A slide hammer with 2x12" split soil core sampler (AMS Inc., ID, USA) was used to obtain the remainder of the active layer down to permafrost (~35–65cm soil depth), including mineral soil layers. The soil cores were stored in sterile Whirl-pak® bags, immediately frozen on site, and shipped to the Stable Isotope Laboratory at the University of Illinois at Chicago where they were sectioned horizontally into 2cm depth segments using a sterilized ice-core cutter, providing a 2cm resolution soil depth profile for each core. A portion of each segment was ground into a fine powder using a Spexmill mixer/mill 8000 (SPEX SamplePrep, NJ, USA) and analysed for C and N concentration and stable isotopes using a Costech Elemental Analyser (Valencia, CA, USA) in line with a Finnigan Deltaplus XL IRMS (isotope ratio mass spectrometer; Bremen, Germany). Soil pH was measured from portions of the same segments by creating a soil slurry mixture (2ml $H_2O$:1g soil) and using an Accumet Basic AB15 pH meter with a calomel reference pH electrode (Thermo Fisher Scientific Inc., MA, USA). In addition, at the time of collection, soil temperature, soil moisture, and active layer thaw depth were measured and recorded at four points around each soil core hole (n=12 per treatment) to characterize the soil environment. Soil temperatures (°C) were measured using a 12cm Taylor TruTemp Digital Instant Read Probe Thermometer (Taylor Precision Products, Inc., NM, USA), surface (top 12cm) volumetric water content (%) was measured using an HS2 HydroSense II Soil Moisture Measurement System (Campbell Scientific Inc., UT, USA), and active layer thaw depths (cm) were measured by inserting a meter stick attached to a metal rod into the ground until it hit ice.

## 2.2 DNA extraction, sequencing, and analysis

Samples from organic and mineral layers of each soil core, as well as the transition between the two, were selected for DNA extraction initially based on visual examination of each individual core section and further classified by %C in saturated soils as per the Soil Survey Division Staff, (1993; Organic ≥ 12% SOC, Mineral: < 12% SOC). Organic samples were collected just below where plant tissue transitioned into dark brown/black soil (mean soil depth ± standard error [S.E.] = 5.6±1.3cm; Control n=4, Deep n=4, Int. n=3, Low n=4), transitional samples were taken from the visual border between organic and mineral layers based on change in soil colour (mean soil depth ± S.E. = 14.8±1.8cm; Control n=3, Deep n=3, Int. n=4, Low n=3), and mineral samples were collected 10cm below this transition (mean soil depth ± S.E. = 25.1±1.7cm; Control n=3, Deep n=4, Int. n=3, Low n=3),, totalling 41 samples. To maintain consistency, only these samples were used to analyse %C, %N, and pH relationships. Samples were sent to Argonne National Laboratory for DNA extraction, amplification, and sequencing as per standards used by the Earth Microbiome Project (Gilbert et al., 2014). DNA extractions were performed using MoBio's PowerSoil®-htp 96 Well Soil DNA Isolation Kit as per protocol, the V4 region of the 16S rRNA gene was amplified using PCR primers 515F/806R (Caporaso et al., 2012), DNA quantification was performed using PicoGreen, and 2x150bp paired-end sequencing was performed using an Illumina MiSeq instrument.

Samples were barcoded prior to sequencing for downstream sample identification and paired-end assembly, demultiplexing, quality filtering, operational taxonomic unit (OTU) picking, and preliminary diversity analyses were performed using the QIIME software package version 1.8.0 (Caporaso et al. 2010). Forward and reverse reads were assembled using fastq-join (Aronesty, 2011) with 15bp overlap at 15% maximum difference. Quality filtering included removal of reads that didn't have at least 75% consecutive high quality (phred > q20) base calls and truncation of reads with more than three consecutive low quality (phred < q20) base calls. This resulted in an assembled-read median sequence length of 253bp.

To reveal phylogenetic abundance and relationships, sequence reads were assigned taxonomic identities using closed reference OTU picking that clusters and matches each read to a reference database. Any read that did not match a sequence in the reference database was discarded. All

default QIIME parameters were used (reference database = Greengenes (13_8), OTU picking method = uclust, and sequence similarity threshold = 97%). Because many organisms are known to possess multiple copies of the 16S rRNA gene in their genome, the abundance assignments were corrected based on known copy numbers using PICRUSt's *normalize_by_copy_number.py* script. The relative abundances of the six most abundant phyla, comprising 82% - 96% of total detected phyla per sample, were analysed for treatment effects, and alpha and beta diversities were examined using the Shannon diversity index to estimate within sample diversity, and Bray-Curtis dissimilarity matrices to determine community structure differences.

The genetic functional potential of bacterial communities was determined using the software package PICRUSt version 1.0.0 (Langille et al., 2013) which predicts functional gene copy numbers in a community based on 16S rRNA sequencing results. Recent advances in sequencing technologies and bioinformatics has greatly enhanced our current knowledge of the genetic potential of soil microorganisms, allowing us to determine what genes a group of organisms is likely to possess based on ancestral state reconstruction of metagenome assemblies from current genomic databases (Langille et al., 2013; Martiny et al., 2013). PICRUSt utilizes this knowledge, revealing functional potential, in the form of gene abundance, associated with phylogenetic community structure. For this study, we targeted Kyoto Encyclopedia of Gene and Genomes (KEGG) ortholog assignments for enzymatic genes commonly associated with SOM decomposition, nutrient (nitrogen and phosphate) mobilization, and environmental stress responses (Sinsabaugh et al., 2008; Waldrop et al., 2010; full list in Table S1). These genes were then grouped according to functional role, resulting in the following nine gene groups: 1) lignin degradation, 2) chitin degradation, 3) cellulose degradation, 4) pectin degradation, 5) xylan degradation, 6) arabinoside degradation, 7) nitrogen mobilization, 8) phosphate mobilization, and 9) superoxide dismutation.

## 2.3   Statistical analyses

Differences between soil layers (Organic, Transition, Mineral) and snow accumulation treatments (Control, Deep, Int., Low), including abiotic measurements and relative abundance of bacterial 16S rRNA and functional genes, were determined using the Kruskal-Wallis test in the R statistical software package with a significance threshold of $p < 0.05$. Due to significant

differences between soil layers (Table S2), each layer was analysed separately. Only organic and mineral layers are reported. All abiotic factors, phyla relative abundances and relative abundances of functional genes were analysed individually to elucidate the treatment effects for each group, and pairwise comparisons were made to determine significant differences between treatments using the Nemenyi post hoc test. In addition, linear regressions were performed to determine relationships between soil chemical properties (%C, %N, C:N, and pH) and bacterial abundance at the phylum level, as well as the gene abundances of SOM degrading enzymes (Supplementary Figs. S1-S15). To ensure accurate comparisons, soil chemical properties were measured from the same samples that DNA was extracted from. Only $R^2$ values > 0.30 are discussed.

Bacterial diversity statistics were calculated using QIIME (Caporaso et al. 2010), specifically the *compare_alpha_diversity.py, compare_categories.py,* and *compare_distance_matrices.py* scripts*.* Pairwise comparisons of the Shannon alpha diversity metrics from soil layer and each treatment group were made using non-parametric two-sample t-tests with 999 Monte Carlo permutations. Beta diversity was analysed by comparing Bray-Curtis dissimilarity matrices of bacterial abundance data from each sample to soil chemical properties, and between soil layers and snow accumulation treatments using adonis tests with 999 permutations. Organic and mineral layers were also analysed separately when comparing snow accumulation treatments and soil chemical properties (Table 2). Analyses of soil chemical properties were further substantiated by Mantel tests, again using 999 permutations. This data was visualized by creating a non-metric multidimensional scaling (NMDS) plot (Stress=0.090, Shepard plot non-metric $R^2$=0.992) in the R package phyloseq (McMurdie and Holmes, 2013) using the same Bray-Curtis dissimilarity matrices (Fig. 2).

## 3    Results

### 3.1    Environmental changes

Significant differences in soil temperature (n=12, H=33.29, df=3, p<0.001), active layer thaw depth (n=12, H=21.35, df=3, p<0.001), and organic layers %C (n=4, H=9.74, df=3, p=0.021) were associated with the four different snow zones. Post hoc tests revealed higher temperatures in the Deep snow zone relative to the Control (p=0.009), the Int. (p=0.001), and the Low snow

zone (p<0.001; Table 1). Active layer depth data revealed similar results, increasing in the Deep snow accumulation zone and decreasing as snow pack was experimentally reduced. Only in the Deep zone was the active layer thaw depth significantly (p=0.020) deeper than the Control zone. However, along the snow accumulation gradient, thaw depth significantly increased from Low to Deep plots (Low/Int. - p=0.021, Low/Deep - p<0.001; Table 1). Soil moisture was not correlated with snow accumulation, possibly the result of surface hydrology at the site, which was largely saturated throughout the growing season. In the organic soil layers, the %C concentration of soil declined with increased snow accumulation (Low/Deep - p=0.03), while the %N concentration only slightly increased (Low/Deep - p=0.32). This resulted in a decreasing trend in C:N ratios across snow accumulation treatment zones and relative to the control (Control/Deep - p=0.14; Table 1). Soil pH tended to increase (became more neutral) with increased snow accumulation (Low/Deep - p=0.06). The changes in the mineral soil layers were less pronounced than in the organic layers. C:N ratios again showed a decreasing trend as snow accumulation increased, while soil pH increased in the Deep zone but did not show a trend along the treatment gradient (Table 1).

## 3.2    Bacterial community shifts

Some bacteria exhibited shifting trends in response to snow depth, both across treatments and relative to the control, while other community shifts were either not significant or did not appear to be the result of the snow depth treatments (Figs. 3 and S16-S20). Noticeable trends at the phylum level included a 1.6-fold increased abundance in Verrucomicrobia (p=0.068), a 2.1-fold increase in Actinobacteria (p=0.083), and a 329.0-fold increase in Chloroflexi (p=0.010) in the organic layers from the Low to Deep snow zones. Acidobacteria showed decreased abundance in all treatments relative to the Control, with the Deep zone exhibiting the largest difference with a 1.98-fold decrease (p=0.055; Fig. 3). In the mineral layers, significant increases in the phylum Chloroflexi (7.18-fold increase; p=0.011) occurred from the Control to Deep zones, while significant decreases (2.84-fold decrease; p=0.019) were observed from Control to Deep zones in the phylum Verrucomicrobia (Fig. 3).

Bacterial abundance in each phylum correlated with at least one of the soil chemical properties we measured (%C, %N, C:N, or pH). The best overall predictor was %C, correlating with four out of the six phyla. It showed negative relationships with Actinobacteria ($R^2$=0.38, p<0.001;

Fig. S4) and Chloroflexi ($R^2$=0.34, p<0.001; Fig. S6), and positive relationships with Bacteroidetes ($R^2$=0.33, p<0.001; Fig. S5) and Proteobacteria ($R^2$=0.32, p<0.001; Fig. S2). Actinobacteria was also negatively correlated with %N ($R^2$=0.34, p<0.001; Fig. S4), and Chloroflexi, positively with soil pH ($R^2$=0.34, p<0.001; Fig. S6). The best and only predictor for Acidobacteria abundance was soil pH, which correlated negatively ($R^2$=0.46, p<0.001; Fig. S1). Verrucomicrobia abundance correlated positively with %N ($R^2$=0.36, p<0.001; Fig. S3).

Soil depth significantly affected bacterial relative abundance in all phyla except for Acidobacteria (Table S2). The organic layers were more abundant in Proteobacteria (1.59-fold difference; p<0.001), Verrucomicrobia (1.48-fold difference; p<0.001), and Bacteroidetes (2.27-fold difference; p=0.001). Phyla that were more abundant in the mineral layers were Actinobacteria (4.48-fold difference; p<0.001) and Chloroflexi (14.21-fold difference; p<0.001).

Alpha diversity, measured using the Shannon index, was found to differ between soil layers (organic / mineral – p=0.003), but not between snow accumulation treatments (Table S3). However, beta diversity of bacterial communities visualized by a NMDS plot of Bray-Curtis dissimilarity indices constructed from community matrices (Stress=0.090, Shepard plot non-metric $R^2$=0.992; Fig. 2) revealed significant differences in community structure between all samples (organic, transition, and mineral) associated with winter snow pack (adonis $R^2$=0.13, p = 0.017), %C (adonis $R^2$=0.24, p < 0.001; Mantel r statistic=0.63, p < 0.001), %N (adonis $R^2$=0.14, p < 0.001; Mantel r statistic=0.34, p < 0.001), C:N (adonis $R^2$=0.19, p < 0.001; Mantel r statistic=0.42, p < 0.001), and pH (adonis $R^2$=0.15, p < 0.001; Mantel r statistic=0.49, p < 0.001). In addition, analysis of each soil layer separately showed that soil chemical properties and snow accumulation treatment affected bacterial community structure more in the organic layers than in the mineral layers, and that in the organic layer, the snow pack treatment (p<0.001), %C (p=0.004), and pH (p<0.001) are the main drivers of community shifts (Table 2).

**3.3 PICRUSt functional analysis**

Of the functional gene groups examined, the most significant treatment effects occurred in the organic soil layers. A 1.27-fold decrease in the abundance of genes involved in cellulose degradation (p=0.018) and a 1.56-fold decrease in the abundance of genes involved in chitin degradation (p=0.029) was observed in the Deep zone relative to the Control (Fig. 4). Also, across treatments from Low to Deep, lignin degrading gene abundance decreased 12.29-fold

(p=0.023), pectin degrading gene abundance decreased 1.41-fold (p=0.018), and xylan degrading gene abundance decreased 1.63-fold (p=0.014; Fig. 4). A similar trend was observed in enzymes responsible for the regulation of oxygen radicals with a 1.05-fold decrease in the Deep zone compared to the Low (p=0.083). Shifts along the snow accumulation gradient were also observed in gene groups involved in nutrient mobilization with a 1.18-fold increase in genes necessary for N mobilization (p=0.14), and a 1.12-fold decrease in genes necessary for phosphate mobilization (p=0.39) in the Deep zone relative to the Control.

Trends in the mineral layers were less clear. Significant shifts included a 2.18-fold increase in genes  encoding for enzymes involved in arabinoside degradation (p=0.049) and a 1.23-fold decrease in enzymes involved in N mobilization (p=0.019) in the Deep zone relative to the Control (Fig. 4). Genes for lignin-degrading enzymes again showed decreasing abundance along the treatment gradient from Low to Deep (16.23-fold decrease; p=0.051). However, relative to the Control, lignin-degrading genes in both Int. and Low zones exhibited much greater abundances than they did in the organic layers (Fig. 4).

All soil chemical properties were found to be poor predictors of gene abundance, with the exception of genes associated with lignin degradation. Both %C and C:N showed positive relationships ($R^2$=0.32, p<0.001 and $R^2$=0.54, p<0.001, respectively; Fig. S10), and soil pH showed a negative relationship ($R^2$=0.41, p<0.001; Fig. S10).

While the analysis did reveal significant changes in enzyme gene abundance across the snow zones, many of the KEGG ortholog groups of enzymes targeted in this study were either not found in any of the samples or were found in very low quantities, including phenol oxidases, peroxidases, and laccases (Table S1).

## 4	Discussion

This study documents changes in soil bacterial community structure in the active layer of moist acidic tundra in response to long-term (18 year) experimental changes in winter precipitation. We examined how changes in bacterial community functional potential as a result of climate forcing factors might affect SOM degradation and alter the C balance of this Arctic tundra ecosystem. Low temperatures in Arctic ecosystems limit soil C availability and decomposability

(Conant et al., 2011; Davidson and Janssens, 2006). However, global warming-induced permafrost thaw may partially alleviate this temperature limitation, potentially releasing large amounts of C into the atmosphere via SOM decomposition and further increasing the rate of global warming (Lupascu et al., 2013, 2014a; Lützow and Kögel-Knabner, 2009; Schuur et al., 2008).

After 18 years of experimental winter snow addition, bacterial community structure and functional potential in Arctic moist acidic tundra changed under deeper winter snow accumulation. Our results indicate that increased snow pack reduced the abundance of genes associated with SOM decomposition in the organic soil layers, suggesting a reduced SOM decomposition potential. Possible explanations for this functional shift may include: 1) altered bacterial C substrate preferences towards more labile sources under lowered $O_2$ availability that would result in a decreased abundance of genes associated with SOM decomposition, and 2) a reduced amount of enzymatic machinery (and fewer gene copies; Rocca et al., 2014) necessary to accomplish similar metabolic results, as increased soil temperatures under snow accumulation may alleviate kinetic limitations of enzyme functioning (German et al., 2012; Sinsabaugh et al., 2008).

## 4.1  Bacterial community shifts

Our results indicate that altered snow accumulation has a significant effect on soil bacterial community structure in Arctic moist acidic tussock tundra ecosystems. While large differences in relative abundances were found between soil layers (Table S2), the most notable effects of snow accumulation occurred in the organic layers. For instance, we observed shifts in the relative abundance in many of the most abundant phyla including Verrucomicrobia, Acidobacteria, and Actinobacteria, particularly in the Deep snow zone (Fig. 3). Shifts in Verrucomicrobia were primarily driven by increases in the order Chthoniobacterales in the Deep snow zones relative to the Low snow zones. This order contains facultative aerobic heterotrophs able to utilize saccharide components of plant biomass, but unable to use amino acids or organic acids other than pyruvate (Sangwan et al., 2004). Shifts in Actinobacteria were dominated by the order Actinomycetales, gram-positive facultative bacteria that have been linked to the stimulation of ectomycorrhizal growth which degrade recalcitrant C (Goodfellow and Williams, 1983; Maier et al., 2004; Pridham and Gottlieb, 1948). While not as abundant, the phylum Chloroflexi also

responded to snow pack treatments, increasing in abundance from Low to Deep snow zones (Fig. 3). Shifts in Chloroflexi were the result of increasing abundance of the class Anaerolineae in the Deep zone. Anaerolineae include green non-sulfur bacteria able to thrive in anaerobic environments and have previously been found in similar cold, water-saturated soils (Costello and Schmidt, 2006). These results appear consistent with the increased soil moisture and decreased partial pressure of $O_2$ documented under increased snow pack at the study site (Blanc-Betes et al., 2016).

These shifts in bacterial phyla indicate that even at the coarsest level of phylogeny and a high degree of variance between samples, deeper snow in winter and associated changes in soil conditions may be driving changes in the belowground community. Bacterial community shifts may be resulting in potentially altered substrate use preference by decomposers, and different genetic functional activity. This is supported by other studies from Arctic soil and permafrost ecosystems that provide evidence of altered microbial community composition and rapid functional response to temperature manipulations, thawing soils, or fertilization treatments (Deslippe et al., 2012; Koyama et al., 2014; Mackelprang et al., 2011). For example, Actinobacteria abundance was found to increase in response to both increased temperature (Deslippe et al., 2012) and in freshly thawed permafrost soils (Mackelprang et al., 2011), similar to the response we observed in the Deep zone (Fig. 3). Mackelprang et al. (2011) also reported varying shifts in a wide array of functional genes in response to permafrost thaw. In addition, Koyama et al. (2014) documented a decrease in the Acidobacteria phylum in response to fertilizer soil inputs which they attributed to be a direct result of competition with α-, β−, and γ-Proteobacteria (oligotrophic vs. copiotrophic bacteria, respectively) which increased in abundance with fertilizer treatment. While oligotrophic organisms such as Acidobacteria are adapted to survive in low nutrient environments, they are often outcompeted in more fertile soils by generalist copiotrophs (such as Proteobacteria) who are better equipped to harvest available nutrients. Our results did not show a clear pattern for Proteobacteria, but they do show that Acidobacteria abundance shifts associate negatively with Proteobacteria shifts in the Deep zone where C:N soil values are lowest (most fertile; Table 1 and Fig. 3).

Correlations between soil chemical characteristics (%C, %N, C:N, and pH) and bacterial phylum abundance partially support findings reported in Fierer et al. (2007). They identified C

mineralization rates (a proxy for C availability) to be the best predictor of bacterial abundance in the dominant phyla, including positive relationships with Bacteroidetes and β-Proteobacteria, and a negative relationship with Acidobacteria (Fierer et al., 2007). We acknowledge that C mineralization and availability differ from %C in that regardless of carbon concentration, physical and chemical factors in the Arctic such as temperature limitations, and high tannin concentrations may limit C mineralization (Davidson and Janssens, 2006; Schimel et al., 1996). Physical protection of SOM by soil aggregates and associations with organo-minerals, also known to limit C mineralization, does not play as large of a role in Arctic soils compared to other soil types (Höfle et al., 2013; Ping et al., 2015). Regardless of these difference in C measurement, our study did find weak positive relationships between %C and Proteobacteria (Fig. S2) as well as Bacteroidetes (Fig. S5), similar to Fierer et al., 2007. Interestingly, although N can be a limiting factor for microbial growth, %N only correlated to two phyla: positively with Verrucomicrobia (Fig. S3) and negatively with Actinobacteria (Fig. S4). While identifying individual abiotic factors that may predict bacterial abundance at the phylum level is informative, it is important to recognize that often a variety of interacting factors determine microbial community composition, and effects at the phylum scale may be too coarse for adequate interpretation. Our results suggest that while C:N (a proxy for SOM quality) is a poor indicator of individual bacterial phylum abundance, %C and %N (and in some cases soil pH) alone may be more relevant in these acidic tundra soils. More detailed studies that address the relationships between soil chemical/abiotic characteristics and microbial community composition at finer phylogenetic scales are needed to adequately identify dependable predictors.

While the alpha diversity of soil bacterial communities via the Shannon index did differ between soil layer, it did not differ between snow pack treatment zones. Also, it does not elucidate community structural or functional differences between samples, and it fails to distinguish shifts in genetic potential among treatments. In contrast, beta diversity analyses better revealed soil bacterial community responses to snow accumulation. Bacterial community structure significantly shifted between snow pack treatment zones at all soil depths / layers (Table 2). The NMDS plot (Fig. 2) shows bacterial community structures to be associated with the snow accumulation treatment as soil chemical properties changed (%C, %N, C:N, and pH), indicating that bacterial β-diversity may respond to indirect changes in soil chemistry in response to winter snow accumulation. The initial effects of increased snow pack result in altered physical factors

(greater active layer thaw depth and increased soil temperatures and moisture; Blanc-Betes et al., 2016) which may lead to increased SOM availability and faster enzyme activities with the potential to enhance SOM decomposition. Higher SOM mineralization may promote the documented shifts in aboveground plant communities and increased NPP (Natali et al., 2012; Sturm et al., 2005, Anderson-Smith 2013), and vegetation shifts to more shrubby species may alter the chemistry and quality of new litter inputs, ultimately affecting decomposer communities. Moreover, soil moisture and compaction can reduce $O_2$ diffusion into the soil, inhibiting aerobic SOM decomposition (Blanc-Betes et al., 2016; O'Brien et al., 2010), and altering bacterial community composition by selecting for microorganisms that utilize simple C substrates, leaving behind complex organic compounds and plant polymers. In addition, tannins produced by expanding woody shrubs may act to inhibit microbial activity (Schimel et al., 1996), further slowing decomposition. This is supported by the lower relative abundance of genes required for SOM decomposition in the Deep snow accumulation zone where we observed the most significant shifts in bacterial community composition (Figs. 3&4). The balance between these two competing processes, and the functional shifts associated with them, will ultimately influence the C balance of the system.

## 4.2    Functional shifts

To examine the influence of shifting bacterial abundances on soil community functioning and the C balance of Arctic ecosystems, we focused on the genetic potential of the bacterial community to produce enzymes required for the degradation of various forms of SOM. We did this by using PICRUSt software to estimate functional gene abundance via ancestral state reconstruction (Langille et al., 2013). While this method does not provide direct measurements of gene abundance (e.g. does not account for horizontal gene transfer or unknown functional / taxonomic linkages that may exist in the sampled tundra soils), it does offer valuable insights into the functional capacities of bacterial communities using 16S rRNA data (Langille et al., 2013). Furthermore, gene abundance in itself is not a direct measurement of gene expression or enzyme activity (Wood et al., 2015). However it does provide a measure of genetic potential and may be positively correlated to enzyme activity and gene expression (Morris et al., 2014; Neufeld et al., 2001; Rocca et al., 2014). To accurately measure enzymatic functional potential or gene expression would require a targeted metagenomic and metatranscriptomic approach.

Many bacterial genes encoding for enzymes associated with the degradation of lignin and other complex plant compounds (such as peroxides, phenol oxidases, and laccases) were not detected in this study. This suggests that bacterial communities preferentially degrade microbial biomass and polysaccharide polymers, and that the decomposition of more recalcitrant forms of C in Arctic soils is performed by other microorganisms such as fungi. Fungi typically play a key role in the degradation of recalcitrant organic matter by specializing in the production of oxidative enzymes (Deslippe et al., 2012; Morgado et al., 2015). The absence of bacterial genes that encode for peroxides, phenol oxidases, and laccases, could also be due to the presence of tannins in the soil, which are common in the Alaskan floodplain and are produced by encroaching shrub species (DeMarco et al., 2014; Schimel et al., 1996). Tannic compounds have been shown to inhibit microbial activity and decrease decomposition by binding to vital enzymes (Schimel et al., 1996). If production of phenol oxidases and peroxides yield little to no benefit for bacteria in this ecosystem due to competition with fungi and interference from tannins and other phenolic compounds, genes encoding for these enzymes may be reduced (Rocca et al., 2014).

The PICRUSt predicted copies of genes for enzymes responsible for SOM decomposition, while generally more abundant in the organic layers (Table S2), were less abundant in the organic layers of the Deep snow zone than in the Control and Low snow accumulation zones (Fig. 4). The genes most affected encode enzymes required for the breakdown of plant derived litter, such as cellulose, xylan, or pectin, all major constituents of plant cell walls. Xylans in particular are common in woody plant tissues (Timell, 1967). The observed decrease of these genes in Deep snow pack suggests bacterial preference of readily available substrates, such as microbial biomass or root exudates (Sullivan and Welker, 2005; Sullivan et al., 2007, 2008). Production of these substrates may have been stimulated by increased soil temperatures and NPP predicted under a climate change scenario, and require less energetic investment in exo-enzyme production (Schimel, 2003). The production of enzymes for the degradation of complex polysaccharides is energetically demanding. Therefore, in an energy and nutrient limited ecosystem such as the Arctic tundra (Hobbie et al., 2002; Jonasson et al., 1999; Mack et al., 2004; Shaver and Chapin, 1980, 1986; Sistla et al., 2012), more labile substrates are likely preferable, which may lead to accumulation of SOM, and thus SOC (Lupascu et al., 2013, 2014a).

Our results indicating reduced decomposition potential under deeper snow pack is consistent
with other long-term warming and snowfence studies from Arctic tundra ecosystems that report
zero net C loss (or even C gain) during the growing season (Natali et al., 2012, 2014; Sistla et al.,
2013). We speculate that initial soil conditions likely favour decomposer activity and
decomposition rates increase in response to increased temperatures, resulting in C loss. Over
time changing soil conditions (e.g. increased moisture, decreased $O_2$ availability, changes in
chemistry of litter inputs) may select for microorganisms that use anaerobic metabolic pathways
such as methanogenesis (Blanc-Betes et al. 2016). These hypoxic soil conditions would limit
aerobic decomposition. As bacterial communities increase the abundance of genes encoding for
enzymes involved in N mobilization, newly available N would enhance microbial biomass
production, plant NPP, leaf litter N content, and induce plant community shifts (Pattison and
Welker, 2014; Schimel, 2003; Welker et al., 2005). A decrease in SOM decomposition is
possibly supported by data from this study, which shows a decreased abundance of genes
involved in SOM decomposition in conjunction with trends suggesting increased abundance of N
mobilization genes in the organic layers as snow pack increases (Fig. 4).
Increased temperature may provide an alternate explanation to the decreased PICRUSt predicted
abundance of genes associated with SOM decomposition in the organic layers of the Deep snow
accumulation zone (Fig. 4) . Enzyme activity is partially regulated by the rate of gene expression
as well as by post-transcriptional regulating factors, which include environmental factors (Gross
et al., 1989). Michaelis-Menten enzyme kinetics are sensitive to temperature (German et al.,
2012), increasing the maximum rate of enzyme activity ($V_{max}$) by increasing the catalytic
constant of the reaction (Razavi et al., 2015). Increased $V_{max}$ may represent an excess potential
enzyme activity for the given substrate or growth conditions, resulting in a down regulation of
genes required for the enzyme (e.g. Gonzalez-Meler et al., 1999, 2001), because fewer enzymes
are needed to achieve similar $V_{max}$ at higher temperatures. Therefore, increases in soil
temperature under deeper snow may partially explain the decrease in PICRUSt predicted
abundance of genes required for SOM decomposition (Table 1 and Fig. 4).
**4.3    Ecosystem response to  snow accumulation**
Whether bacterial communities are responding to changing plant inputs and corresponding
altered SOM quality (decreased C:N; Table 1) or whether they are directly altering SOM

chemistry through selective decomposition remains unclear. From the results of our study, it is clear that increased snow accumulation may lead to changes in both bacterial community composition and SOM chemistry in the organic soil layers (Table 1 and Fig. 3). Unlike other ecosystems where plants are the first responders to abiotic climate change factors, in the Arctic, microbes are likely the first responders to changes in temperature by initially increasing nutrient mineralization. These released nutrients facilitate plant community shifts and increase ecosystem NPP (Chapin III et al., 1995). Over time, the combination of increased snow accumulation and soil compaction may lead to hypoxic/anaerobic soil conditions (e.g., Blanc-Betes et al., 2016) and further vegetative shifts to wet-sedge (*Carex)* species, limiting SOM decomposition. This in combination with a recent history of more recalcitrant plant litter inputs could result in re-accrual of SOC (e.g., Sistla et al., 2012), ultimately mitigating the positive feedback loop hypothesized in current literature (Davidson and Janssens, 2006; Natali et al., 2014; Schuur et al., 2009; Sturm et al., 2005).

## 5    Conclusions

The results presented here support the hypothesis that bacterial community structure and function shift as a result of consistently deepened snowpack. Increases in soil hypoxia under deepened snow may have resulted in an increased abundance of anaerobic or facultative bacteria, slowing decomposition. Decreases in PICRUSt predicted gene copies suggest that SOM decomposition may be slowed under accumulated snow, and bacterial community substrate preference may shift to more labile compounds. Concentrations of C and N, as opposed to C:N, better explained bacterial community responses to snow pack treatments. Together these results strongly suggest that soil decomposers of moist acidic tundra are key in determining the direction and magnitude of permafrost C feedbacks on the climate system.

## Author Contributions

M. P. Ricketts, J. M. Welker, and M. A. Gonzalez-Meler designed the experiment. R. S. Poretsky provided expertise and insight into the bioinformatics and data analyses. M. P. Ricketts performed all sample collections, lab work, and data analyses. M. P. Ricketts prepared the manuscript with contributions from all co-authors.

## Data Availability

All data used in this report are publically accessible through two separate data repositories. The 16S rRNA gene sequences derived from Illumina Mi-Seq sequencing have been deposited in the NCBI Sequence Read Archive (SRA) under accession number SRP068302. All computer script text files used in QIIME and R packages, as well as BIOM and Excel files, are available via the NSF Arctic Data Center, doi:10.18739/A2DP96.

## Acknowledgements

This research was supported by funding from the Department of Energy, Terrestrial Ecosystem Science Program to M. A. Gonzalez-Meler (DE-SC 0006607). J. M. Welker initiated and maintained the snow fence experiment with NSF funds beginning in 1994 (9412782, 0632184, 0612534, 0856728). Michael Ricketts received additional support from the Elmer Hadley Graduate Research Award, the W.C. and May Preble Deiss Award for Graduate Research, and the Bodmer International Travel Award. We would like to thank members of the Stable Isotope lab at the University of Illinois at Chicago (UIC), particularly Jessica Rucks and Gerard Aubanell for assistance with field work/sample collection and processing. We thank Elena Blanc-Betes, Douglas Johnston, Dr. Douglas Lynch, and Dr. Charlie Flower for advice on experimental design, analyses, and comments on the manuscript. We would also like to thank Dr. D'Arcy Meyer Dombard (Earth and Environmental Sciences department, UIC) for use of her lab and soil expertise, and Imrose Kauser (Microbial Ecology lab, UIC) for help with the bioinformatics. Finally we thank Olivia L Miller (Purdue University), Niccole Van Hoey (University of Alaska, Anchorage), all of the staff at Toolik Field Station, and field and lab UIC undergraduate assistants: Ben Thurnhoffer, Andres Davila, and Briana Certa.

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

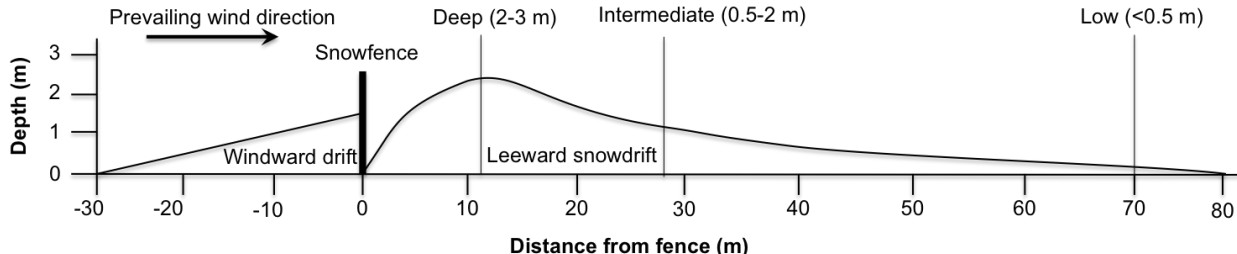

Figure 1. Modified from Walker et al., 1999.  Schematic of snow accumulation depth at moist
acidic tundra site from snow fence manipulation. Three soil cores were obtained from each
treatment zone (labeled Deep, Intermediate, and Low) and a Control zone located >30m outside
the effect of the snowfence.

Table 1. Abiotic characteristics of soil from snow accumulation treatments (Low = ~25% less snow pack than the Control, Int. = ~50% more snow pack than the Control, Deep = ~100% more snow pack than the Control). Values are means ± standard errors. Soil chemical properties were obtained from samples used for DNA extraction, while temperature and thaw depth were measured *in situ* (n=12). Organic and mineral samples were analysed separately using the Nemenyi post hoc test. Results are indicated by [a,b,c] only where $p<0.05$.

| Treatment | Soil Layers | Sample Depths (cm) | %C | %N | C:N | pH | Temp @ 12cm (°C) | Thaw Depth (cm) |
|---|---|---|---|---|---|---|---|---|
| Control | Organic (n=4) | 6.75±3.12 | 45.21±1.09 [ab] | 1.01±0.20 | 50.04±9.44 | 4.59±0.09 | 4.32±0.27 [b] | 59.17±1.23 [bc] |
| | Mineral (n=3) | 26.00±5.51 | 2.57±0.39 | 0.15±0.03 | 17.67±1.34 | 5.15±0.05 [ab] | | |
| Low | Organic (n=4) | 5.50±1.89 | 46.63±0.73 [a] | 1.06±0.07 | 44.59±2.54 | 4.44±0.08 | 2.92±0.24 [b] | 50.92±3.20 [c] |
| | Mineral (n=3) | 27.00±1.15 | 4.18±1.92 | 0.22±0.11 | 19.42±0.65 | 5.16±0.20 [ab] | | |
| Int. | Organic (n=3) | 3.67±0.67 | 40.59±2.43 [ab] | 1.17±0.25 | 38.38±8.85 | 4.69±0.41 | 4.08±0.25 [b] | 61.88±1.19 [ab] |
| | Mineral (n=3) | 23.67±2.03 | 2.58±0.49 | 0.14±0.02 | 18.58±1.45 | 5.01±0.04 [a] | | |
| Deep | Organic (n=4) | 6.00±3.70 | 36.51±4.27 [b] | 1.40±0.07 | 26.27±3.41 | 5.61±0.21 | 6.49±0.20 [a] | 65.42±1.49 [a] |
| | Mineral (n=4) | 24.00±4.42 | 1.65±0.19 | 0.10±0.01 | 16.41±0.56 | 5.83±0.17 [b] | | |

Table 2. Statistical analysis of beta diversity using adonis and Mantel tests. Bray Curtis distance matrices of bacterial communities for each sample were compared between soil layers (Organic, Transition, Mineral) and snow accumulation treatments (Control, Deep, Int., Low), and to soil chemical properties. Sample sizes were n=15 for "Organic", n=13 for "Mineral", and n=41 for "All layers". Significance is indicated by asterisks (* = p<0.05, ** = p<0.01, *** = p<0.001).

| | | adonis | | | Mantel test | |
|---|---|---|---|---|---|---|
| | Samples | $R^2$ | df | p-value | r statistic | p-value |
| Soil layers | All | 0.320 | 2 | <0.001 *** | NA | NA |
| Snow pack | All | 0.126 | 3 | 0.017* | NA | NA |
| | Organic only | 0.421 | 3 | <0.001*** | NA | NA |
| | Mineral only | 0.485 | 3 | 0.003** | NA | NA |
| %C | All | 0.239 | 1 | <0.001*** | 0.633 | <0.001*** |
| | Organic only | 0.212 | 1 | 0.004** | 0.490 | 0.008** |
| | Mineral only | 0.055 | 1 | 0.720 | 0.047 | 0.791 |
| %N | All | 0.141 | 1 | <0.001*** | 0.341 | <0.001*** |
| | Organic only | 0.111 | 1 | 0.131 | -0.0245 | 0.883 |
| | Mineral only | 0.051 | 1 | 0.788 | 0.032 | 0.844 |
| C:N | All | 0.191 | 1 | <0.001*** | 0.415 | <0.001*** |
| | Organic only | 0.165 | 1 | 0.022* | 0.180 | 0.269 |
| | Mineral only | 0.108 | 1 | 0.195 | -0.063 | 0.629 |
| pH | All | 0.147 | 1 | <0.001*** | 0.490 | <0.001*** |
| | Organic only | 0.368 | 1 | <0.001*** | 0.709 | <0.001*** |
| | Mineral only | 0.297 | 1 | 0.004** | 0.526 | <0.001*** |

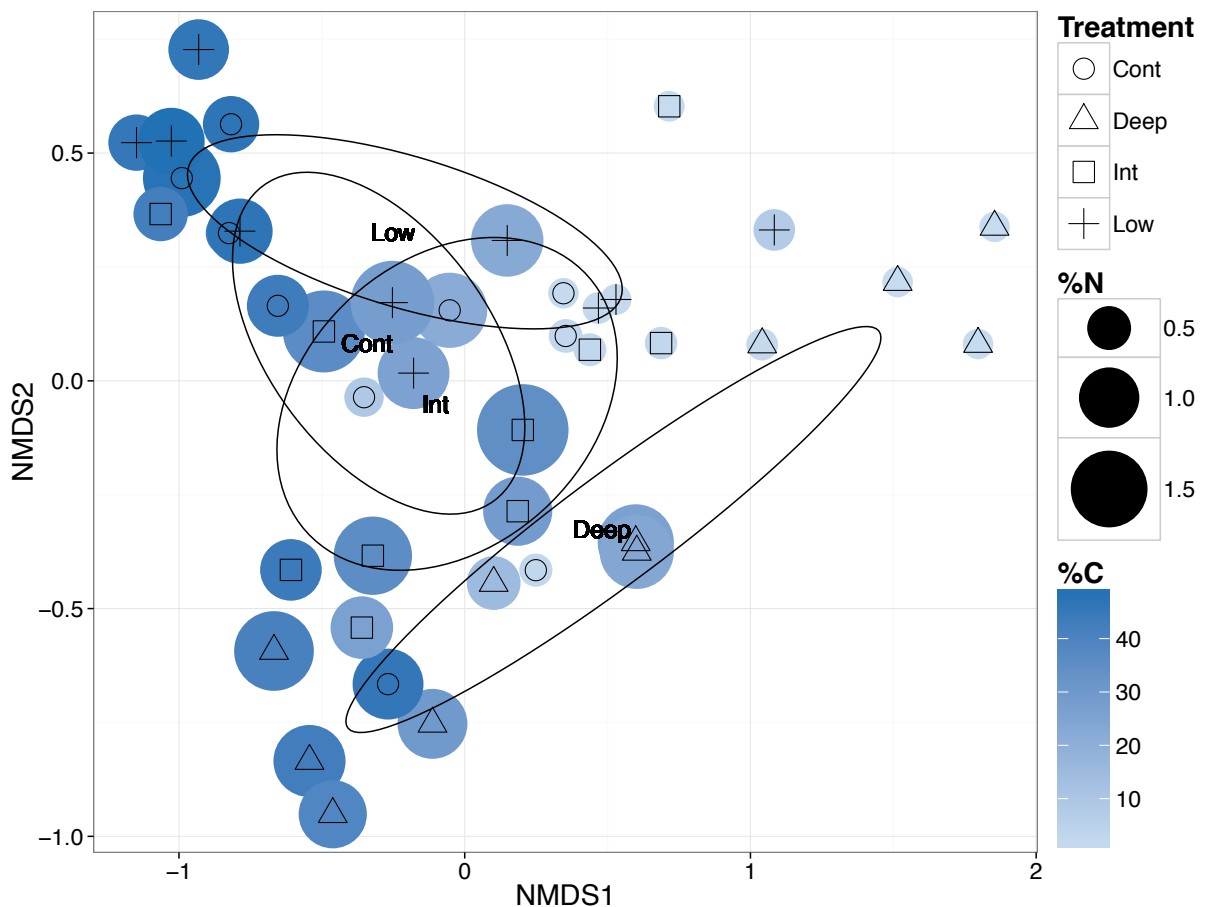

Figure 2. Non-metric multidimensional scaling (NMDS) plot using Bray-Curtis dissimilarity

matrices (Stress=0.090, Shepard plot non-metric $R^2$=0.992). Each point represents the

 bacterial community structure within one of the 41 total samples used for DNA extraction from

all soil depths (Organic, Transition, and Mineral). Colours indicate %C ranging from 1.4% (light

blue) to 48.6% (dark blue), bubble size indicates %N ranging from 0.09% (small) to 1.95%

(large), and shapes indicate snow accumulation treatments (Control, Deep, Int., Low). Ellipse

centroids represent treatment group means while the shape is defined by the covariance within

each group.

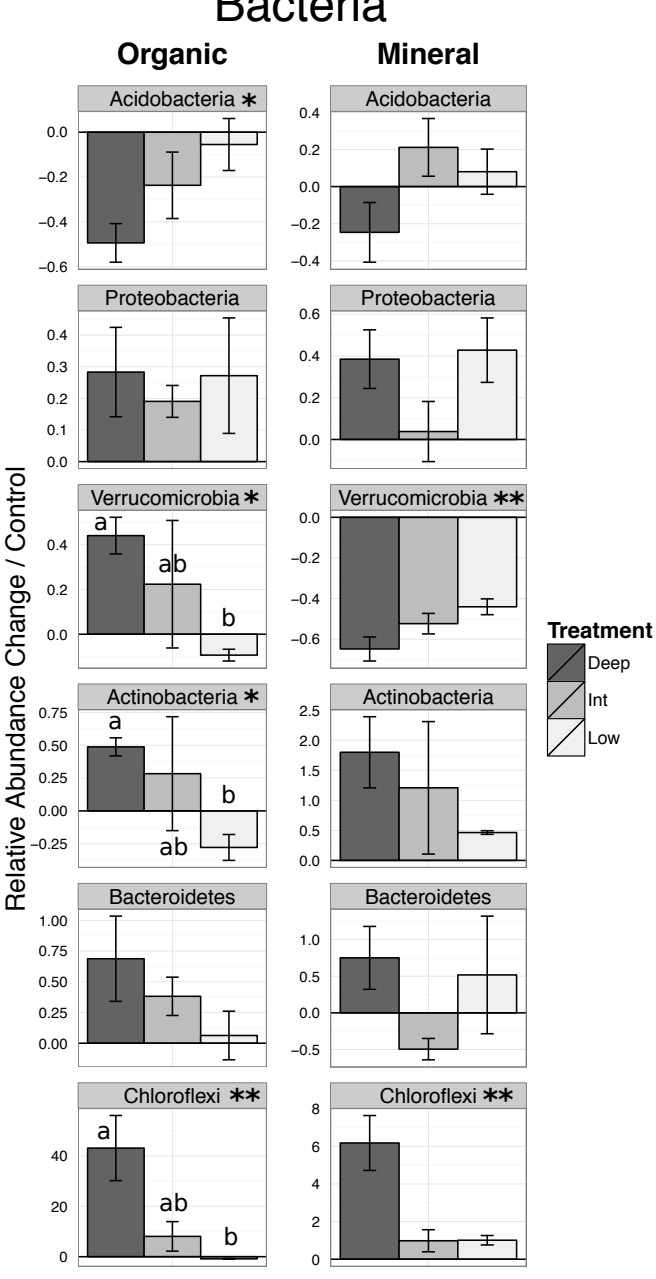

Figure 3. Averaged relative abundance of the six most abundant bacterial phylum relative to the control, separated by snow accumulation treatment, and in order of greatest abundance (top to bottom). Error bars represent standard error (standard error of controls ranged from 12.929 in Chloroflexi to 0.026 in Verrucomicrobia). Significance determined by Kruskal-Wallis tests is indicated by asterisks (* = p<0.1, ** = p<0.05), while post-hoc Nemenyi test results are indicated by "a, b, ab", except where significant differences were to the control.

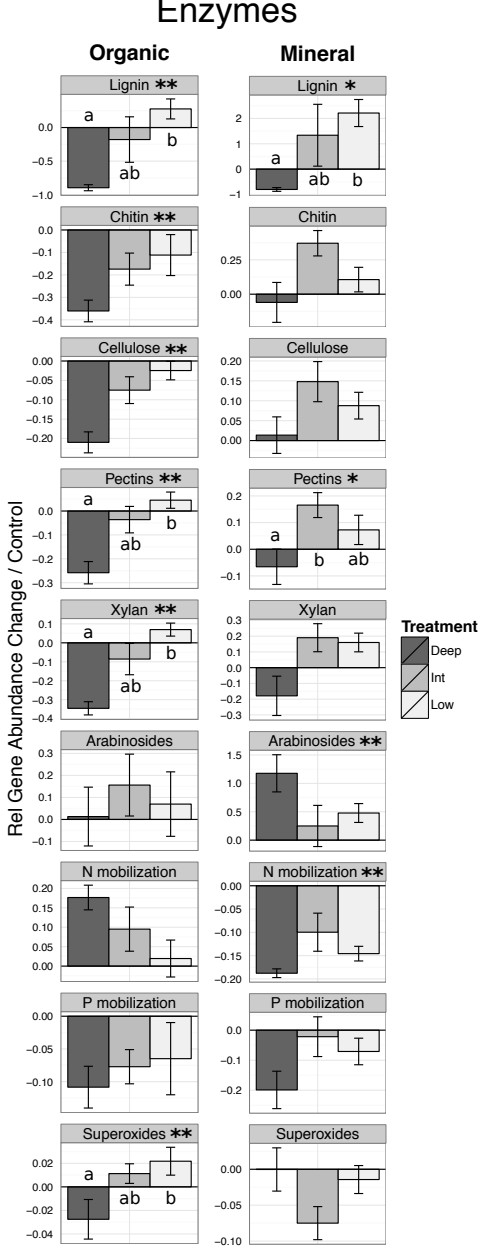

Figure 4. Averaged relative abundance of genes for enzyme functional groups relative to the control and separated by snow accumulation treatment. Functional groups involved in soil organic matter decomposition are ordered from recalcitrant to labile substrates (top to bottom). Error bars represent standard error (standard error of controls ranged from 1.220 in the lignin group to 0.008 in the superoxides group). Significance determined by Kruskal-Wallis tests is indicated by asterisks (* = p<0.1, ** = p<0.05), while post-hoc Nemenyi test results are indicated by "a, b, ab", except where significant differences were to the control.