# Peer review of "Soil bacterial community and functional shifts in response to"

_SOIL, 2015_

## Referee Comment (RC1) · Anonymous Referee #1 · 7 Feb 2016

In this manuscript, the authors describe the changes in bacterial community composition as a result of increased snow cover in a moist arctic Tundra. The study shows that increased snow cover led to changes in bacterial community composition along changes in soil chemistry and the plant community. The authors conclude that the observed changes in bacterial community composition and function might lead to reduced decomposition of SOM in these arctic systems. The manuscript is well written and structured and the story is for the most part easy to follow. After careful revisions the manuscript should be of great interest to the readership of SOIL. However there are some issues that need to be addressed or discussed in more detail to improve the manuscript. 1. Soil depth: As the authors point out, that there is a huge difference in

edaphic factors between organic and mineral horizons in this study. Such depth related differences have been shown to potentially influence microbial community structure and function and the potential controls on those (Eilers 2012 SBB, Schnecker 2015 SBB). The authors should also test the effects of depth as well as treatment and potential interactions on the individual bacterial groups, their relations to soil factors and beta diversity using Adonis and perform the mantel tests with the edaphic factors separately for organic and mineral horizons. 2. Vegetation and decomposition: The authors state that an increased snow cover ultimately leads to reduced decomposition and C loss from the system since NPP is increased and might offset potential losses of C. While their results show a reduced potential for decomposition in the bacterial community and other studies have found increased NPP in shrubby tundra compared to tussock tundra, the C contents in organic and mineral horizons decreased significantly. This huge loss could have either happened during the transition from tussock to shrubby vegetation, which would mean that NPP did not offset decomposition or during the transition into a sedge dominated fen, which would indicate that decomposition was not reduced despite the reduction of the bacterial potential for decomposition. 3. Fungi and oxidative enzymes: The authors should more strongly point out that this study is focused on bacterial community composition and function throughout the text and that fungi might play an important part especially in the production of oxidative enzymes which have been found in arctic soils (Tveit 2012 ISMEJ). 4. The authors should be more careful with the interpretation of the ancestral state reconstruction, since these results are strictly based on the sequencing results of the bacterial community. Changes in the so obtained functions can only be interpreted as changes in the bacterial community composition. Any statements concerning enzyme kinetics, enzyme transcription, activity or even in situ functional gene copy number can only be speculated on and should be clearly marked as speculation (especially Page 17 Lines 1-19) 5. The authors should consider that any changes in the bacterial community composition could be independent of SOM properties and be a result of changes in temperature, moisture vegetation length and so on and could vary with depth (Schnecker 2014 Plos One,
[Figure]

Gittel 2013 ISMEJ)

Detailed comments: Title: Since multiple environmental factors are changed with increased snow cover, "thermal insulation" should be replaced with "altered snow cover" or similar. Introduction: Page 3 Lines 17-23: Since there is another paragraph on SOM in the arctic this one could be omitted. Especially since the numbers for global C storage here and in the paragraph on arctic C storage are not the same.

Page 4 Line 1-2: The nutrient limitation of Arctic soils has recently been challenged (Wild 2015 GBC, Melle 2015 SBB). Page 5 Lines 5-18: The Authors should consider using testable hypotheses instead. Structure and O2 availability were not measured in this study. The change in plant species composition might not be a consequence of increased nutrient availability but the result of changed water status. With the experimental setup it cannot be distinguished between substrate effects and environmental effects. Material and Methods: Please mention which program was used to perform the statistical analyses. As mentioned before the measured parameters, including beta diversity should be tested for depth effects and interactions of depth and treatment. All correlative tests should also be performed separately for organic and mineral horizons. Results: Page 10 Lines 26- Page 11 Line 2: These results should be presented in a separate table. Page 11 Lines 10-11: The reported p-values are not significant. Page 11 Lines 24-25: This interpretation should be moved in the Discussion section of the manuscript and "microbial communities" should be replaced with "bacterial communities". Discussion: Page 12 Line 11: As stated before, while the bacterial functional potential might indicate reduced SOM decomposition, the decrease in C content from control to DEEP suggests otherwise. Page 12 Lines 12-17: An alternative explanation might be that the microbial community composition is shaped by the environmental factors and less so by SOM properties. Page 12 Line 16: Blanc-Bates et al. 2015 is missing in the Reference list. Is this the same that is listed as submitted in Page 16 Line 12. If this is the case and if this study was conducted at the same site, mentioning this and a short description of the findings would help the reader understanding the

author's arguments about changes in SOC dynamics. Page 13 Line 26: The strong correlation of Acidobactera with pH and the non-significant correlation with C:N questions that statement. Page 13 Lines 28-30: This sentence can be omitted. Page 14 Line 28: This could be a depth effect and not a result of the altered snow cover. Page 15 Lines 7-11: Is there any indication that increased tanning occurred at the studied site? Page 15 Lines 23-27: Binding of enzymes to tannins could happen to any enzyme. Oxidative enzymes could actually degrade tannins and might thus be upregulated. Page16 Line 13: Sistla et al 2013 did not use a snowfence study. Page 16 Lines 13-28: While this study explains to some extent some of the author's statements, it is over represented for its current publication status. Page 17 Lines 1-19: see general comments above.

---

## Referee Comment (RC2) · Anonymous Referee #2 · 15 Feb 2016

Ricketts et al. examined an interesting question: How do changes in snow cover affect soil bacterial community structure and function? They sampled and analyzed soil material from an interesting long-term snow depth manipulation experiment and applied up-to-date methods for bacterial community characterization. They claimed out that an increase in snow depth resulted in an increase in soil insulation that led to changes in bacterial community structure, to a decrease in enzyme encoding genes and in C%. The MS presents results of a relevant experiment and the topic is within the scope of the Journal.

The Introduction is well written. However, in my opinion more information about the experimental set-up and the sampling design are required in the Methods section. How

did they ensure that the snow depth was continuously increased for 100%, 50%, or decreased for 25% along the study period of 18 yrs? Did they continuously measure snow depth each year? Did they remove or add snow in cases with more or less than e.g. 100% than control? Or is the treatment rather a distribution of increased/decreased snow depth around e.g. 100% (+/- SD) than a fixed treatment level? How did they monitor the annual input of snow at each point? Was it equal for all years? The applied non-parametric statistics seem to be appropriate for the experiment and data set, however, re-analysis of the data might be necessary because organic and mineral soil samples seem to be included within one analysis (without accounting for differences in sampling depth). In my opinion, too many results were mentioned as significant effects even though the p values were above 0.05. This is problematic and partly lead to a rather speculative discussion and conclusion. To sum up, I recommend thorough revision of the manuscript in order to focus on the observed effects of snow depth on soil bacterial communities.

At least four references cited in the text are not included in the reference list.

Please, find my specific comments below:

Abstract Line 4-5 "(i.e. more or less snow), resulting in increased winter insulation" This statement is partly contradictive. Omit "or less" or add "increased or decreased winter insulation". L8 "context of ecosystem response to climate change." Please change to "context of expected ecosystem response to . . ." L15 "most abundant phyla" requires a value about the contribution of these phyla on total abundance. "20% or 80% of total detected phyla?" L26 The authors did not study the temperature sensitivity of extracellular enzymes (sensu str.) they are requested to omit any statement/conclusion about this.

Introduction Page 3 L 1: How can the stability of the structure be threatened? I suggest to change to more ecological terminology. L8 Anisimov and Vaughan must be changed to Anisimov et al. or the respective reference must be added to reference list. L11

needs reference L12 needs reference P4 L16 omit activity or kinetics – I prefer the use of the term "kinetics". P5 L11 Why should microbes be unable to degrade SOM from shrubs? Needs further explanation or changing in the sense of" the potential to degrade SOM might be reduced".

Methods Are detailed vegetation surveys available for each plot? How far away were the replications at each treatment located from each other? Did they sample more than one soil core for each replicated plot and compiled composite samples or not? Figure 1 is important and helps to understand the experimental set-up. However, where is the control located? P5 L24 "strategically" needs further explanation. P5 L29-30 Soil Survey 2015 is not listed in references P6 L1 "regime": the tested climate change scenario is: variable precipitation (that may induce differences in soil temperature) but constant air temperature. P6 treatment/factor levels: -25% vs. +50% (vs. +100%) are not equally selected. This might be problematic for ANOVA. Please check. P6 L9: n=3? Total number of sampled cores = 12? P6 L18: unclear, how did they use the 2 cm depth segments for further analysis since they presented data for "organic" and "mineral" soil only. Did they calculate the average value of C% etc. for each of the two strata by considering the data of the single segments? P7 L 11: Please provide the absolute sampling depths for each treatment (average value and variation) in the Methods section. Sampling depth might be considered as co-variable in non-parametric ANCOVA in order to account for any effect. L22 Caporaso et al is not included in the reference list P8 L 4: "six most abundant phyla" this requires a quantitative documentation for the six phyla. P8 L 18-20 needs reference(s) P8 L 24 – P9 L 14 Selected types of statistical analysis seem to be appropriate for the experimental set-up and data. (Maybe non-parametric ANCOVA is required for the consideration of sampling depth as co-variable). However, P9 L1 the selection of linear regression analysis is inconsistent since the authors applied non-parametric tests for the data. Taken into account that the primary assumptions for parametric tests are not full-filled (they used the non-parametric tests) then the use of linear regression seems not to be adequate. In addition, the use of median and median absolute deviation might be more robust

estimates (and consistent) of the central tendency and variation of the data than mean and SD. Table 1: The lower case letters used indicate that organic and mineral soil material was included in one analysis. (?) I suggest comparing the treatment effects on the two strata (organic and mineral) independently of each other (for both KW-ANOVA and Nemenyi-test; performing the respective tests for the treatment effect on e.g., C% of "organic"). Add a brief description of the treatments (-25% of snow cover compared to control etc.) to the table description. P9 L7-8: two-sample t-test for the comparison of four groups? I do not understand. The selected measure of dissimilarity as well as the criteria for NMDS seems to be adequate / sufficient. How many dimensions were included / considered for NMDS? P9 L10-11: Do I understand right? The analysis of associations between explanatory variables (i.e., soil chemical properties) and bacterial data were further analyzed by Mantel test? Which statistical software was used?

Results P9 L18ff I thought the test statistic of KW-ANOVA is the "H-value" and not $Chi^2$? Please check. Please calculate the effect sizes for each tested factor and variable. P9 L30: p=0.32 I would not consider this as a significant difference. P10 L1: p=0.14 I would not consider this as a significant difference. L2 p=0.06 indicates a tendency L11 Verrucomicrobia and Actinobacteria p-values indicate tendencies. Section 3.2. in many cases the order of magnitude of the relationships were rather low. P11 L5-8 p-values of which comparisons? CTL to DEEP or LOW to DEEP. Or do the p-values represent the results of the KW-ANOVAs? L10 "N mobilization genes" Do the genes mobilize N? Please, correct. L12-13 It might be more meaningful to write as follows: " . . . included an increase in genes encoding enzymes involved in . . .".see P15 L16-17 L25: omit "simple cellulosic and"

Discussion P12 L10: only moderate changes and in many cases they observed only trends. Please, focus on significant results p<0.05. L12: "towards more labile sources" if it is an important "pathway" then the term/concept "labile" requires definition in the introduction section. L13: What is "SOM enzyme activity"? L14: The positive relationship between gene copies and enzyme machinery requires a reference. L15/16 Which limitations of enzyme kinetics? Change "enzymatic decomposition reactions" to "enzyme functioning" . Blanc-Betes et al. is not included within the reference list L18-19 But in the present account a decrease in C% was observed. Discuss this issue in contrast to the results in the literature. Is the decrease only found in C% or also in C-stock? L22/23 Significant changes were observed only for a few groups. L28/29: What are the possible links to enzyme production and functioning? L29: Sangwan et al. is not included within the reference list. P13 L7 "cold saturated soil" Did you mean "cold, water-saturated soil"? L7 Costello and Schmidt . is not included within the reference list. L28-30: Omit. P14 L12-13: Results and no discussion. L26 omit "(Stress . . .) L28/29 Results and no discussion. P15 L1-2 Speculative. The authors did not measure enzyme kinetics. Omit "Rate" in "Rate of enzyme kinetics" since enzyme kinetics are the substrate-dependent rate of enzyme-substrate interaction. L7-8: Is this a logic relationship: increases in tannins and increases in N availability? L9 The decrease in C% might indicate exactly the opposite of the mechanism described above. L18: Are any data available about fungal biomass? Is the microbial biomass dominated by bacterial or by fungal biomass? (Did you investigate effects of snow depth on microbial biomass?) L23-27: Does an interaction between tannins and extracellular enzymes necessarily induce shifts in gene abundances? Statement requires a reference. P16 L1-2: Does the decrease correspond to a decrease in C% or in C-stock or to a decrease in microbial biomass (as a factor for enzyme production)? Typically, enzyme activities are normalized with C%. Would this data treatment (gene abundances / C%) lead to a disappearance of treatment effects? L4-5 Sullivan 2008 is not included within the reference list L8-9: Is the system nutrient limited? Reference. N% increased in DEEP treatment - why did enzyme production not (that requires N and P)? L12 Blanc-Betes et al. Submitted: not included in reference list L19: Which alternate energetic pathway? L21 change to "genes encoding enzymes involved in organic N degradation" L22: Microbial biomass was frequently reported to be positively related to enzyme production and decomposition. Please, describe more precisely under which circumstances an

increase in N availability and microbial biomass results in a decrease in decomposition rate. P17 L6 needs a reference - for example: Razavi et al. Front. Microbiol., 14 October 2015 | http://dx.doi.org/10.3389/fmicb.2015.01126 L7-8 Speculative. If in-situ substrate availability is low, Vmax will not be reached and enzyme functioning is controlled by Km (Michaelis-Menten constant). Alternatively, authors may change to "reach the same catalytic rate". L26: needs reference L26-27 This statement should be reformulated in order to account for the low number of observed effects (the low number of repetitions) and the revised results of the KW-ANOVA (separate data from "organic" from those of "mineral soil").

References IPCC should be shifted to I and not to W.(?)

Table 1: Describe in Methods how many repetitions and analytical replicates were used for the data. a> Or < b, please add information. No effect on N% - that differs from the description in the results section Why is there no effect on C:N in Table 1? Figure 2: remove the ellipses from the graph. I counted 11 triangles but 10 circles etc. Why? How many sample points were included within the NMDS? NMDS requires detailed description within the results section. What are the main gradients observed? The NMDS optimized the illustration of the dissimilarity in beta diversity data but not in explanatory variables. Therefore, the combined illustration is somehow misleading (but the Mantel test is the appropriate method of choice). Figure 3: No clear effect on bacterial phyla in organic samples (only some tendencies). Mineral soil: effect on Verrucmicrobia. Figs 3 and 4: I would prefer to change the order of treatments within the graph (from low (left) to deep (right) and the use of the same scale (y-axis) for all panels.) Add information about the treatments to Figure captions. Fig 4 Superoxides +2 to -3% difference to control? This might be a very low effect.

[Figure]

---

## Referee Comment (RC3) · Anonymous Referee #3 · 18 Feb 2016

The manuscript by Ricketts et al. addresses the effect of climate change predictions on the soil bacterial communities in Arctic tundra soils which are important global C sinks. The experiment was carried out in a long-term experimental field site offering a snow depth gradient from 25% lower to 100% more snow than the surrounding (control). Due to increasing snow depth significant changes in abiotic soil parameters (e.g. active layer thaw depth, T, C/N ratio) were observed as well as a shift in the bacterial community structure. The taxonomic information from 16S rRNA amplicon sequence data was further used to estimate the functional gene abundance. These results indicate a decreased SOM decomposition potential under predicted climate change conditions which might help to further optimize current climate models. Therefore the study

is of interest for readers of SOIL journal. Authors give a good overview about the current literature and make the aim of the study clear. In general, the manuscript is well-written and follows a logic flow. However, there are some points which were not clear to me and should be addressed before publication. The prediction of microbial functional composition from phylogeny is really advanced and delivers new insights for studies where only 16S rRNA genes were sequenced. However, I would have liked to have first more information on taxonomic composition of bacterial communities in soil treatments and not only tests on six dominant phyla. The response of bacterial taxa belonging to the same phylum might be completely different. I recommend adding a table of significantly treatment-responding genera (some were already mentioned, p.12 section 4.1) or a heatmap showing relative abundance of dominant OTUs or genera across all samples. Furthermore, it should be emphasized in the discussion that all functional gene abundances are based only on predictions not taking into account horizontal gene transfer that might decouple function from phylogeny. Furthermore, there might be a lot of unknown functions due to poorly characterized taxa in Tundra soils or not yet known links between taxon and function which should be discussed. By the way, it would be also interesting to know the amount of unclassified bacteria in your samples. In this respect, I am also wondering whether you tested first for availability of nearby genome representatives for your dataset before using PICRUSt prediction (NSTI index)? Furthermore, PICRUSt outputs a gene potential and it remains unknown to which extent these genes are expressed in the end. As far as I understood, there are no treatment repetitions at the experimental site available and per treatment only 3 pseudo-replicates were taken. This makes it difficult to exclude the effect of natural variation in the bacterial community composition between sampling points. Thus conclusions can be drawn only very carefully and you should avoid to speculate too much in the Discussion section.

These are my specific comments:

Abstract: p.2, l.4 – Does "more or less snow" mean that predictions on amount of

precipitation are not sure yet? Please consider rephrasing. p.2, l.11 – I recommend writing "Microbial community DNA was extracted from soil". p.2, l. 15 - Taxonomic names should be written in italics (throughout the manuscript).

Introduction: p.3, l.3 – Do you refer to belowground "microbial" community structure? p.5, l.1 + l.16 – Please change microbial into bacterial community. p.5, l.14 – bacterial functions (plural).

Methods: p.6, l.1 – How far away from the snow fence was the CTL sampled? p.6, l.9 – What was the distance between the replicates per treatment? p.7, l.4 – I couldn't find the results of the bacterial communities from the transition zone; and was DNA extracted from all three replicated cores per treatment? p.7, l.16- Could you please add the reference for the primers? p.7, l.17 delete – in Mi-Seq but add to p.8, l.7 Bray-Curtis. p.8, l.6- Why did you determine adequate sampling depth? – I could not find that result later on. p.7, l.25/26 (and following pages)– "enzyme gene abundance" does not exist - please consider rephrasing, e.g. " relative abundance of bacterial 16S rRNA and functional genes". p.8, l.27 – later there is also a significance threshold of p<0.1 used.

Results: I assume that the main focus here is the comparison of each treatment (LOW, INT, DEEP) to the control plot. However, sometimes this is not clear to me from the type of statistical tests you did and from the description of the results. I recommend in general spending some more sentences to explain your results. You often miss to look at the n-fold change of rel. abundance to the control or the comparison between organic and mineral layer. If there is a significant difference to the control- was it observed in all treatments? Is it the same change (decrease/increase) in all treatments? e.g. p.10, l.3 – Instead of saying that the "C/N ratios became more similar" I would point out the %N trend is opposed in the organic and mineral layer. Furthermore I would suggest to add that differences in treatments are in general smaller in the mineral layer compared to organic layer. p.10, l.6 – delete "and". p.10, section 3.2. – Please point out the n-fold change of relative abundances. For instance, the change in sequences affiliated

to Chloroflexi is much stronger than for Actinobacteria. p.10, l.11 According to the text p=0.011 for Chloroflexi but there is only one * in Fig. 3. p.10, l.13 Please consider rephrasing. I guess this is what you want to say: "acidobacterial abundance was in all treatments (DEEP, INT, LOW) lower than in the control". p.10, l.19 please check p-value for Actinobacteria text vs. Fig. S4. p.11, l.6/7 Were lignin, pectin and xylan degradation not significantly different to the CTL? p.11, l.14 Please rephrase – I agree it is also a decrease in genes coding for lignin degradative enzymes over the gradient but the scale differs and both LOW and INT have higher gene abundances compared to CTL in mineral layer in contrast to the organic layer. p.11, l.18 – Are Figs. S7-S9, S11 needed since they are not mentioned in the text? p.11, l.23-25 - Please move this sentence to discussion and refer to Table S1.

Discussion: p.12, l.11 Please explain from which results the conclusion of "reduced SOM decomposition" was derived from! p.12, l/12/13 – I don't agree with explanation 1) since you did not find differences in soil moisture along the gradient. p.12, l.30 – Actinomycetales are a bacterial order containing several taxa, thus please use plural . p.13, l.1 – Is the increase in Actinomycetales, that are linked to degradation of recalcitrant compounds, contradicting to your conclusions from functional predictions? Please discuss. p.13, l.19- I suggest to delete Koyama et al. reference here because this is a totally different experiment. Instead cite Fierer et al. 2007 who tried an ecological classification of soil bacteria. p.14, l.9 and following- I suggest to delete R2 and p-values from the discussion. p.14, l.25 and following- I recommend to transfer results of Fig.2 to Results section. Regarding the replicate size of this study I suggest to be more careful here with conclusions. p.15, l.14-27- The statements about fungi and tannic compounds are too speculative since this can not be supported by data from your study. Instead I would like to have a discussion of PICRUSt limitations here. Why is there only a decrease in rel. abundance of functional genes- which genes might be increased?

Tables & Figures: Table 1 – I don't really understand the number of replicates you refer

to here. n=4 are technical replicates? Significance was tested between treatments for each layer separately? According to Methods part you measured temperature at 4 different depths but not at 12 cm. Was there no post-hoc test done for %N, C/N, and pH? Figure 2 – Difficult to understand. Microbial communities from how many replicates and layers are plotted here? Please use "CTL" as abbreviation for control (similar to the text). Figure 3- Please indicate for which significant effect you tested here. I don't understand, why there was no post-hoc test performed for Acidobacteria or is there only a significant difference to the control and not between treatments. The same applies for Fig. 4. Supplement Fig.S1-S15 – From your Methods section it is not clear to me whether you analyzed abiotic soil parameters in the same soil sample (depth) as the bacterial community composition.

References (typos): p.22, l13 Gonzalez-Meler p.23, l34 mcrA in italics p.27, l13 CO2

---

## Author Comment (AC1) · 24 Apr 2016

Interactive comment on "Soil bacterial community and functional shifts in response to thermal insulation in moist acidic tundra of Northern Alaska" by M. P. Ricketts et al.

Anonymous Referee #1

In this manuscript, the authors describe the changes in bacterial community composition as a result of increased snow cover in a moist arctic Tundra. The study shows that increased snow cover led to changes in bacterial community composition along changes in soil chemistry and the plant community. The authors conclude that the observed changes in bacterial community composition and function might lead to reduced decomposition of SOM in these arctic systems. The manuscript is well written and structured and the story is for the most part easy to follow. After careful revisions the manuscript should be of great interest to the readership of SOIL. However there are some issues that need to be addressed or discussed in more detail to improve the manuscript.

Response: We would first like to thank the reviewer for the positive comments above and the constructive criticisms below. We will do our best to address them and are confident they will make the manuscript better.

1. Soil depth: As the authors point out, that there is a huge difference in edaphic factors between organic and mineral horizons in this study. Such depth related differences have been shown to potentially influence microbial community structure and function and the potential controls on those (Eilers 2012 SBB, Schnecker 2015 SBB). The authors should also test the effects of depth as well as treatment and potential interactions on the individual bacterial groups, their relations to soil factors and beta diversity using Adonis and perform the mantel tests with the edaphic factors separately for organic and mineral horizons.

Response: Samples were analysed separately by soil horizon / depth (Organic vs. Mineral) for most factors included in this manuscript (%C, %N, C:N, pH, bacterial abundance, predicted gene abundance). We have added Table 2 to the revision to report statistical differences in beta diversity, including separate analyses for each soil depth. We also did analyse statistical differences between soil depths / horizons for each of the six most abundant phyla and for each enzyme gene abundance. However, our primary goal for this paper was to address the effects of increasing snow pack on soil bacterial communities regardless of soil depth. Therefore, we chose to highlight the treatment effect over the depth effect.

2. Vegetation and decomposition: The authors state that an increased snow cover ultimately leads to reduced decomposition and C loss from the system since NPP is

increased and might offset potential losses of C. While their results show a reduced potential for decomposition in the bacterial community and other studies have found increased NPP in shrubby tundra compared to tussock tundra, the C contents in organic and mineral horizons decreased significantly. This huge loss could have either happened during the transition from tussock to shrubby vegetation, which would mean that NPP did not offset decomposition or during the transition into a sedge dominated fen, which would indicate that decomposition was not reduced despite the reduction of the bacterial potential for decomposition.

Response: Carbon content is not a good indicator of carbon-stock. Degrading permafrost often results in soil consolidation (loss of ice collapses soils) with associated changes in bulk density and depth redistribution of soil and C. The C-stock profile change as a result of the snow fence treatment is part of another paper. Carbon stock over the soil profile to the average active layer equivalent depth was 7% higher for the intermediate than for the control. Here we have used %C in our analyses because most of the C (if not all) is accessible to microbes (these acidic tundra soils have little to no physical aggregation, JD Jastrow personal communication). Therefore the factors affecting organic matter readiness to microbial decomposition is likely the chemistry/quality of the organic matter (%C, C/N) in addition to temperature and moisture. We hope this is made more clear in the revised manuscript. Also, we acknowledge that the use of the word "content" when referring to our %C data may have been misleading. To clarify, we have changed the phrase "C (or N) content" (i.e. C stock) to "C (or N) concentration" throughout the manuscript. Specifically at the following locations: Page 2 line 11, Page 6 line 22, Page 10 lines 7&8, and Page 14 line 16.

3. Fungi and oxidative enzymes: The authors should more strongly point out that this study is focused on bacterial community composition and function throughout the text and that fungi might play an important part especially in the production of oxidative enzymes which have been found in arctic soils (Tveit 2012 ISMEJ).

Response: We changed "microbial" to "bacterial", or "microorganisms" to "bacteria" in

the following locations in the revision: Page 5 lines 1, 6, 10, 14, 18 Page 12 lines 11, 24 Page 13 line 22 Page 15 line 18 Page 16 line 23 Page 17 line 9, Page 33, Figure 2, line 3 The role of fungi, while not highlighted in this study, is acknowledged and discussed on Page 16 lines 8-11. We also added "by specializing in the production of oxidative enzymes" to Page 16 lines 9-10.

4. The authors should be more careful with the interpretation of the ancestral state reconstruction, since these results are strictly based on the sequencing results of the bacterial community. Changes in the so obtained functions can only be interpreted as changes in the bacterial community composition. Any statements concerning enzyme kinetics, enzyme transcription, activity or even in situ functional gene copy number can only be speculated on and should be clearly marked as speculation (especially Page 17 Lines 1-19)

Response: This is well noted and care was taken to revise the manuscript with it in mind, including the following: We have added the following sentence to Page 15 line 30 – Page 16 lines 1-4: "While the use of PICRUSt and ancestral state reconstruction does not provide direct measurements of gene abundance (e.g., does not account for horizontal gene transfer or unknown functional / taxonomic linkages that may exist in the sampled tundra soils), it does offer valuable insights into the functional capacities of bacterial communities (Langille et al., 2013)." We have removed "found" and replaced it with "predicted" on Page 16 line 18. We have added "predicted" to Page 17 lines 17 & 27.

5. The authors should consider that any changes in the bacterial community composition could be independent of SOM properties and be a result of changes in temperature, moisture vegetation length and so on and could vary with depth (Schnecker 2014 Plos One, Gittel 2013 ISMEJ)

Response: We agree. However, untangling what is driving bacterial community shifts in this system requires isolation of these different factors in a laboratory setting, which

is outside the scope of this paper, however we are planning some of these experiments. The very nature of our experiment, altered snow pack over a long period of time, changes a variety of factors that may contribute to bacterial community change ($O_2$ diffusion via moisture or compaction, temperature, plant community, etc.).

Detailed comments: Title: Since multiple environmental factors are changed with increased snow cover, "thermal insulation" should be replaced with "altered snow cover" or similar.

Response: Agreed! We replaced "thermal insulation" to "altered snow pack" in the revision.

Introduction: Page 3 Lines 17-23: Since there is another paragraph on SOM in the arctic this one could be omitted. Especially since the numbers for global C storage here and in the paragraph on arctic C storage are not the same.

Response: While we appreciate the comment, it is important to recognize that the paragraph being referred to highlights the important role that microorganisms play in C cycling on a global scale versus the later one that specifically describes the significance of C in Arctic ecosystems. We have modified the sentence to highlight global vs. arctic C dynamics. We added "on a global scale, releasing nutrients..." on Page 3 lines 15-16.

Page 4 Line 1-2: The nutrient limitation of Arctic soils has recently been challenged (Wild 2015 GBC, Melle 2015 SBB).

Response: We have removed the sentence from the manuscript. However, while this may be the case in more recent studies, as rising temperatures in the Arctic may be initially accelerating SOM decomposition and releasing more nutrients, historically the Arctic has been observed to be a nutrient limited ecosystem (Hobbie et al., 2002; Jonasson et al., 1999; Mack et al., 2004; Shaver and Chapin, 1980, 1986; Sistla et al., 2012).

Page 5 Lines 5-18: The Authors should consider using testable hypotheses instead. Structure and O2 availability were not measured in this study. The change in plant species composition might not be a consequence of increased nutrient availability but the result of changed water status. With the experimental setup it cannot be distinguished between substrate effects and environmental effects.

Response: This is a valid point. Changes in plant species (and underlying causes), while proposed as a contributing factor to bacterial shifts, was not the focus of this study, and therefore we chose not to test for causes of vegetation shifts within our study site.

Material and Methods: Please mention which program was used to perform the statistical analyses. As mentioned before the measured parameters, including beta diversity should be tested for depth effects and interactions of depth and treatment. All correlative tests should also be performed separately for organic and mineral horizons.

Response: We added "...in the R statistical software package..." on Page 9 lines 2-3 of the revision. We added Table 2 to report beta-diversity statistics, both for all samples (All layers), and separated by soil depth (organic and mineral) in the revision.

Results: Page 10 Lines 26- Page 11 Line 2: These results should be presented in a separate table.

Response: These results are included in Table 2 of the revision.

Page 11 Lines 10-11: The reported p-values are not significant.

Response: While the p-values are not significant, we felt it was still important to acknowledge notable trends with p-values <0.1 as long as the p-values are reported.

Page 11 Lines 24-25: This interpretation should be moved in the Discussion section of the manuscript and "microbial communities" should be replaced with "bacterial communities".

Response: "Microbial" will be replaced with "bacterial" in the revision. Also, the indicated sentence has been deleted from the results and the following sentence has been altered/added to the discussion, Page 16 lines 4-9, The overall absence of bacterial genes encoding for peroxides, phenol oxidases, and laccases (which are primarily associated with the degradation of lignin and other complex plant compounds) suggests that bacterial communities either preferentially degrade microbial biomass and polysaccharide polymers, or that the decomposition of more recalcitrant forms of C in Arctic soils is performed by other microorganisms such as fungi."

Discussion: Page 12 Line 11: As stated before, while the bacterial functional potential might indicate reduced SOM decomposition, the decrease in C content from control to DEEP suggests otherwise.

Response: Please see response to general comment 2 above.

Page 12 Lines 12-17: An alternative explanation might be that the microbial community composition is shaped by the environmental factors and less so by SOM properties.

Response: This is a good point. However, this paragraph was meant to discuss the functional shifts as opposed to the phylogenetic shifts. To clarify, we will add the word "functional" to Page 12 line 23 of the revision.

Page 12 Line 16: Blanc-Bates et al. 2015 is missing in the Reference list. Is this the same that is listed as submitted in Page 16 Line 12. If this is the case and if this study was conducted at the same site, mentioning this and a short description of the findings would help the reader understanding the author's arguments about changes in SOC dynamics.

Response: This citation has been removed.

Page 13 Line 26: The strong correlation of Acidobactera with pH and the non-significant correlation with C:N questions that statement.

Response: This sentence highlights competitive interactions between bacterial phyla.

Any correlation with abiotic factors in the context of this sentence would be indirect.

Page 13 Lines 28-30: This sentence can be omitted.

Response: We appreciate your opinion and will omit this sentence in the revision.

Page 14 Line 28: This could be a depth effect and not a result of the altered snow cover.

Response: The effects of the snow accumulation treatment are statistically significant in the organic horizon alone (p=0.001), the mineral horizon alone (p=0.003), and within all samples (p=0.017; please see Table 2 in the revision). Therefore, we are conservative in reporting that snow accumulation affects bacterial community structure with a p-value of 0.017.

Page 15 Lines 7-11: Is there any indication that increased tanning occurred at the studied site?

Response: We did not test for tannin concentration in this study, however the encroaching shrub species at the site are known to produce them (DeMarco et al., 2014; Schimel et al., 1996).

Page 15 Lines 23-27: Binding of enzymes to tannins could happen to any enzyme. Oxidative enzymes could actually degrade tannins and might thus be upregulated.

Response: This is a valid point. However, if the bacterial community does not possess the functional capacity to produce these oxidative enzymes (as suggested by our data), they will not be able to increase their production. As mentioned in Page 15 lines 16-21, fungi may perform this role in this system, and thus would not show up in our 16S rRNA gene analysis.

Page16 Line 13: Sistla et al 2013 did not use a snowfence study.

Response: We added "...warming and..." to Page 17 line 2 of the revision.

Page 16 Lines 13-28: While this study explains to some extent some of the author's statements, it is over represented for its current publication status.

Response: This citation and description of the study has been removed from the revised manuscript.

Page 17 Lines 1-19: see general comments above.

Response: Please see response to general comment 4 above.

---

## Author Comment (AC2) · 24 Apr 2016

Interactive comment on "Soil bacterial community and functional shifts in response to thermal insulation in moist acidic tundra of Northern Alaska" by M. P. Ricketts et al.

Anonymous Referee #2

Ricketts et al. examined an interesting question: How do changes in snow cover affect soil bacterial community structure and function? They sampled and analyzed soil material from an interesting long-term snow depth manipulation experiment and applied up-to-date methods for bacterial community characterization. They claimed out that an

increase in snow depth resulted in an increase in soil insulation that led to changes in bacterial community structure, to a decrease in enzyme encoding genes and in C%. The MS presents results of a relevant experiment and the topic is within the scope of the Journal.

Response: We thank the reviewer for the encouraging summary and for the criticisms which are addressed below.

General comment #1- The Introduction is well written. However, in my opinion more information about the experimental set-up and the sampling design are required in the Methods section. How did they ensure that the snow depth was continuously increased for 100%, 50%, or decreased for 25% along the study period of 18 yrs? Did they continuously measure snow depth each year? Did they remove or add snow in cases with more or less than e.g. 100% than control? Or is the treatment rather a distribution of increased/decreased snow depth around e.g. 100% (+/- SD) than a fixed treatment level? How did they monitor the annual input of snow at each point? Was it equal for all years?

Response: Snow treatments are caused by the wind drift distance from the snowfence. Snow fall varies from year to year but the drift caused the relative snow accumulation at similar distances from the fence every winter. Snow depth and density have been measured sporadically and reported in other papers which we cite. To clarify, we added, "While snowfall varied from year to year, the wind drift from the fence provided consistent relative snow accumulation at similar distances from the fence every winter." to Page 5 lines 29-30 – Page 6 line 1.

General comment #2 - The applied non-parametric statistics seem to be appropriate for the experiment and data set, however, re-analysis of the data might be necessary because organic and mineral soil samples seem to be included within one analysis (without accounting for differences in sampling depth). In my opinion, too many results were mentioned as significant effects even though the p values were above 0.05. This

is problematic and partly lead to a rather speculative discussion and conclusion. To sum up, I recommend thorough revision of the manuscript in order to focus on the observed effects of snow depth on soil bacterial communities.

Response: Thank you for this suggestion, one that has also been pointed out by other referees. We have included new stats and language to distinguish the soil depth analyses and added p values to let the reader decide the statistical significance of results. We have noted results with marginal significance of $p<0.1$ in addition to the $p<0.05$.

At least four references cited in the text are not included in the reference list.

Response: We have reviewed our citations and added references in the revision.

Please, find my specific comments below:

Abstract Line 4-5 "(i.e. more or less snow), resulting in increased winter insulation" This statement is partly contradictive. Omit "or less" or add "increased or decreased winter insulation".

Response: Agreed! We omitted "or less" on Page 2 line 4 in the revision.

L8 "context of ecosystem response to climate change." Please change to "context of expected ecosystem response to : : :"

Response: We think this is a valid point and have changed it in the revision, as recommended.

L15 "most abundant phyla" requires a value about the contribution of these phyla on total abundance. "20% or 80% of total detected phyla?"

Response: This is a good idea! We added "…(ranging from 82% to 96% of total detected phyla per sample)…" to Page 2 lines 15-16 in the revision.

L26 The authors did not study the temperature sensitivity of extracellular enzymes (sensu str.) they are requested to omit any statement/conclusion about this.

Response: Thank you for your comment. However, we feel that while this sentence is speculative, it is an important possible mechanism explaining our results and is supported by outside literature (Conant et al., 2011; Davidson and Janssens, 2006; Lützow and Kögel-Knabner, 2009).

Introduction Page 3 L 1: How can the stability of the structure be threatened? I suggest to change to more ecological terminology.

Response: We have changed the wording of this sentence to "Broad and rapid environmental changes are driving both above- and belowground community shifts in the Arctic" on Page 3 lines 2-3 of the revision.

L8 Anisimov and Vaughan must be changed to Anisimov et al. or the respective reference must be added to reference list.

Response: Thank you! This was intended to be the same reference! We have fixed it in the revision.

L11 Needs reference

Response: We have added (Anisimov et al., 2007; Liston and Hiemstra, 2011) to Page 3 line 11 of the revision.

L12 Needs reference

Response: We have added (Hugelius et al., 2013; Ping et al., 2008; Schuur et al., 2009; Tarnocai et al., 2009) to Page 3 lines 12-13 of the revision.

Page 4 L16 omit activity or kinetics – I prefer the use of the term "kinetics".

Response: We have omitted ". . .activity and. . ." from Page 4 line 16 of the revision.

Page 5 L11 Why should microbes be unable to degrade SOM from shrubs? Needs further explanation or changing in the sense of" the potential to degrade SOM might be reduced".

Response: We have removed, "...is unable...", and replaced it with "...results in a reduced ability..." on Page 5 lines 10-11 of the revision.

Methods

Are detailed vegetation surveys available for each plot? How far away were the replications at each treatment located from each other? Did they sample more than one soil core for each replicated plot and compiled composite samples or not? Figure 1 is important and helps to understand the experimental set-up. However, where is the control located?

Response: Each of these of these questions have been addressed separately. Please see below: -We did not collect detailed vegetation surveys for each plot. -Replicate distance = I estimate ~15-20m. We added "approximately 15-20m apart" to Page 6 line 12. -Number of cores/compiled – Added "replicate" and "and analyzed separately" to Page 6 lines 12-13. Added "of each soil core" and deleted "soil" to Page 7 line 7. -Control – Added ">30m" to Page 6 line 4. We also added "Three soil cores were obtained from each treatment zone, labelled Deep, Intermediate, and Low, and a Control zone located >30m outside the effect of the snowfence." to the caption for Figure 1.

Page 5 L24 "strategically" needs further explanation.

Response: The word "strategically" refers to its orientation based on wind patterns and is described in the papers we referenced (Jones et al., 1998; Walker et al., 1999). Also, please see response to general comment #1 above.

L29-30 Soil Survey 2015 is not listed in references

Response: We added "Soil Survey Staff, Natural Resources Conservation Service, United States Department of Agriculture. Web Soil Survey. Available online at http://websoilsurvey.nrcs.usda.gov/. Accessed [11/17/2015]." to the references.

Page 6 L1 "regime": the tested climate change scenario is: variable precipitation (that may induce differences in soil temperature) but constant air temperature.

Response: Please see response to general comment #1 above.

P6 treatment/factor levels: -25% vs. +50% (vs. +100%) are not equally selected. This might be problematic for ANOVA. Please check.

Response: For our analysis, each treatment was defined as a categorical variable, thus avoiding any numerical percentage gradient. This is the more conservative approach.

L9: n=3? Total number of sampled cores = 12?

Response: Yes. 3 – CTL, 3 – DEEP, 3 – INT, 3 – LOW = 12 cores. We added "(for a total of 12 soil cores)" to Page 6 line 13. We also made alteration throughout the Methods section and figures to be more transparent on sample sizes.

L18: unclear, how did they use the 2 cm depth segments for further analysis since they presented data for "organic" and "mineral" soil only. Did they calculate the average value of C% etc. for each of the two strata by considering the data of the single segments?

Response: We only analysed %C, %N, and pH for the samples used for the DNA extractions. To clarify, we added "To maintain consistency, only these samples were used to analyse %C, %N, and pH relationships." to Page 7 lines 16-17 of the revision. Also, we added "To ensure accurate comparisons, soil chemical properties were measured from the same samples that DNA was extracted from." to Page 9 lines 10-11 of the revision.

Page 7 L 11: Please provide the absolute sampling depths for each treatment (average value and variation) in the Methods section. Sampling depth might be considered as co-variable in non-parametric ANCOVA in order to account for any effect.

Response: Changed "...and was more variable (ranging from 15-36cm soil depth) due the varying depths of transition." to "(mean soil depth = 25.1cm ±1.7cm)" in Page 7 lines 11-12. Added "(mean soil depth = 14.8±1.8cm)." to Page 7 line 10. Changed ",typically between 0-6cm soil depth, except in one case where the top 10cm was

primarily plant tissue" to "(mean soil depth = 5.6±1.3cm)" in Page 7 lines 8-9.

L22 Caporaso et al is not included in the reference list

Response: We added this to the references in the revision.

Page 8 L 4: "six most abundant phyla" this requires a quantitative documentation for the six phyla.

Response: We will add ",comprising 82% - 96% of total detected phyla per sample," to Page 8 lines 10-11.

L 18-20 needs reference(s)

Response: Added "(Sinsabaugh et al., 2008; Waldrop et al., 2010;. . ."to Page 8 line 25 of the revision.

L 24 – P9 L 14 Selected types of statistical analysis seem to be appropriate for the experimental set-up and data. (Maybe non-parametric ANCOVA is required for the consideration of sampling depth as co-variable). However, P9 L1 the selection of linear regression analysis is inconsistent since the authors applied non-parametric tests for the data. Taken into account that the primary assumptions for parametric tests are not full-filled (they used the non-parametric tests) then the use of linear regression seems not to be adequate. In addition, the use of median and median absolute deviation might be more robust estimates (and consistent) of the central tendency and variation of the data than mean and SD.

Response: This is a valid point and we thank the reviewer for pointing it out. The linear regression analyses were initially performed simply to give us an idea of how the abiotic soil data affected individual bacterial phylum, and gene group abundances. While there are many interesting relationships, we ultimately decided to report the more appropriate non-parametric stats for the bulk of the analyses, but to still include the regressions in the supplemental material.

Table 1: The lower case letters used indicate that organic and mineral soil material was included in one analysis. (?) I suggest comparing the treatment effects on the two strata (organic and mineral) independently of each other (for both KW-ANOVA and Nemenyi-test; performing the respective tests for the treatment effect on e.g., C% of "organic"). Add a brief description of the treatments (-25% of snow cover compared to control etc.) to the table description.

Response: For clarification, the organic and mineral horizon samples were analysed independently as suggested. We have removed the lower case letters "c" from the %C column, since that analysis was not part of the organic horizon analysis, and there were no statistical differences in the mineral horizon. Also, we added, "Organic and mineral samples were analysed separately using the Nemenyi post hoc test. Results are indicated by a,b,c only where $p<0.05$." to the Table description, Page 31 lines 5-6. We added "(Low = ∼25% less snow pack than the Control, Int = ∼50% more snow pack than the Control, Deep = ∼100% more snow pack than the Control)." to the Table description, Page 31 lines 1-3.

Page 9 L7-8: two-sample t-test for the comparison of four groups? I do not understand. The selected measure of dissimilarity as well as the criteria for NMDS seems to be adequate / sufficient. How many dimensions were included / considered for NMDS?

Response: The two-sample t-tests were used to perform pairwise comparisons of the treatment groups. We will clarify this by replacing, "The Shannon alpha diversity metric was compared across treatments using. . ." with, "Pairwise comparisons of the Shannon alpha diversity metrics from each treatment group were made using. . ." in Page 9 line 16. The ordination of the Bray-Curtis distance matrices using NMDS yielded two convergent solutions found after four tries, resulting in 2 dimensions.

L10-11: Do I understand right? The analysis of associations between explanatory variables (i.e., soil chemical properties) and bacterial data were further analyzed by Mantel test? Which statistical software was used?

Response: Yes. The QIIME script compare_distance_matrices.py was used with –method=mantel to compare the Bray-Curtis distance matrices to distance matrices created from the abiotic data using the QIIME script distance_matrix_from_mapping.py. The QIIME software was used to run these tests. To clarify, we altered the sentence on Page 9 line 13, as follows: "Bacterial diversity statistics were calculated using QIIME (Caporaso et al. 2010), specifically the compare_alpha_diversity.py, compare_categories.py, and compare_distance_matrices.py scripts."

Results L18 I thought the test statistic of KW-ANOVA is the "H-value" and not Chi2? Please check. Please calculate the effect sizes for each tested factor and variable.

Response: Yes, you are correct. Confusion arose because of the way the R package reports the values for kruskal.test ("Kruskal-Wallis chi-squared = 0.80317, df = 3, p-value = 0.8487"). After checking, we discovered that this value is indeed the Kruskal-Wallis test statistic, otherwise known as "H", although it has been reported both ways. We have corrected this in the revision. Page 9 lines 28-29. Sample sizes were added to clarify effect size on Page 9 lines 28-29 of the revision. Also, "(n=12 per treatment)" was added to Page 6 lines 30-31 in the revision.

L30: p=0.32 I would not consider this as a significant difference.

Response: While not considered significant, the increasing trend does contribute to decreasing C:N. Therefore to address this comment, we added "only slightly" to Page 10 line 11 of the revision.

Page 10 L1: p=0.14 I would not consider this as a significant difference.

Response: While the result is insignificant, we feel that the overall trend contributes to the story. We have reworded the sentence as follows: "This resulted in a decreasing trend in C:N ratios across snow accumulation treatment zones and relative to the control (CTL/DEEP - p=0.14; Table 1)." on Page 10 lines 8-10 of the revision.

L2 p=0.06 indicates a tendency

Response: We added the phrase "tended to" to Page 10 line 10 of the revision.

L11 Verrucomicrobia and Actinobacteria p-values indicate tendencies. Section 3.2. in many cases the order of magnitude of the relationships were rather low.

Response: These comments are true, and as such, we attempted to phrase our results in a way that openly and accurately reports the data, without dismissing trends that may or may not be the result of the treatment or abiotic soil conditions. Hopefully, this will allow readers to be able to openly interpret the data. We have also added n-fold changes to the Results in the revision.

Page 11 L5-8 p-values of which comparisons? CTL to DEEP or LOW to DEEP. Or do the p-values represent the results of the KW-ANOVAs?

Response: Thank you for catching this confusion. The first two, cellulose and chitin, were comparing the DEEP zone relative to the CTL. The latter three were comparing across treatments from LOW to DEEP. We will add, "in the DEEP zone" to Page 11 line 20 and separate the sentence into two to clarify the difference.

L10 "N mobilization genes" Do the genes mobilize N? Please, correct.

Response: To clarify this, we have altered the sentence as follows: "Shifts along the snow accumulation gradient were also observed in gene groups involved in nutrient mobilization with an increase in genes necessary for N mobilization (CTL/DEEP – p=0.14) and a decrease in genes necessary for phosphate mobilization (CTL/DEEP – p=0.39)." on Page 11 lines 24-26.

L12-13 It might be more meaningful to write as follows: ": : : included an increase in genes encoding enzymes involved in : : :".see P15 L16-17

Response: Great recommendation. We altered this in the revision as suggested.

L25: omit "simple cellulosic and"

Response: We deleted this in the revision as recommended.

Page 12 L10: only moderate changes and in many cases they observed only trends. Please, focus on significant results p<0.05.

Response: To address this, we replaced the word "phylogeny" with the phrase, "community structure" in Page 12 line 21. The result from the adonis test (p=0.017) supports the claim that community structure is significantly affected by the snow addition treatment. Additionally, this sentence (while broad is scope), is not false. The significant results we do find still do indicate a "change". The remainder of the discussion focuses on these significant results, occasionally using the trends in the data to speculate cause.

L12: "towards more labile sources" if it is an important "pathway" then the term/concept "labile" requires definition in the introduction section.

Response: Thank you for your comment. We added, "...bacterial communities may be forced to use more labile C sources such as microbial biomass or root exudates, ultimately causing SOC to increase over time." to the introduction, Page 5 line 12-13.

L13: What is "SOM enzyme activity"?

Response: We replaced " decreased SOM enzyme activity" with "a decreased abundance of genes associated with SOM decomposition" in Page 12 line 29.

L14: The positive relationship between gene copies and enzyme machinery requires a reference.

Response: We're glad you noticed that! We have considered this and it is acknowledged and discussed with references on Page 18 lines 3-13. We also added the reference "(Rocca et al., 2014)" to Page 13 line 1.

L15/16 Which limitations of enzyme kinetics? Change "enzymatic decomposition reactions" to "enzyme functioning" . Blanc-Betes et al. is not included within the reference list

Response: We have change the wording as recommended. Blanc-Betes reference

has been removed from this sentence in the revision.

L18-19 But in the present account a decrease in C% was observed. Discuss this issue in contrast to the results in the literature. Is the decrease only found in C% or also in C-stock?

Response: This sentence was removed from the revision. Carbon content is not a good indicator of carbon-stock. Degrading permafrost often results in soil consolidation (loss of ice collapses soils) with associated changes in bulk density and depth redistribution of soil and C. The C-stock profile change as a result of the snow fence treatment is part of another paper. Carbon stock over the soil profile to the average active layer equivalent depth was 7% higher for the intermediate than for the control. Here we have used %C in our analyses because most of the C (if not all) is accessible to microbes (these acidic tundra soils have little to no physical aggregation, JD Jastrow personal communication). Therefore the factors affecting organic matter readiness to microbial decomposition is likely the chemistry/quality of the organic matter (%C, C/N) in addition to temperature and moisture. We hope this is made more clear in the revised manuscript. Also, we acknowledge that the use of the word "content" when referring to our %C data may have been misleading. To clarify, we have changed the phrase "C (or N) content" (i.e. C stock) to "C (or N) concentration" throughout the manuscript. Specifically at the following locations: Page 2 line 11, Page 6 line 24, Page 10 lines 10-11, and Page 14 line 22.

L22/23 Significant changes were observed only for a few groups.

Response: While this is true, the sentence refers to "bacterial community structure" (or beta diversity), which is significantly altered by the treatment, as per the adonis test (p=0.017). In other words, just because significant results were only obtained for a few of the most abundant phyla, the overall community structure (which takes into account ALL organisms/OTU's) does significantly change due to increased snow depth.

L28/29: What are the possible links to enzyme production and functioning?

Response: Enzymes involved in the utilization of plant biomass and SOM are described in the discussion, Page 16 lines 22-26. We have added some of these cross-references throughout the discussion.

L29: Sangwan et al. is not included within the reference list.

Response: Thank you for catching that. We have added the citation to the reference list.

Page 13 L7 "cold saturated soil" Did you mean "cold, water-saturated soil"?

Response: Yes, we did! We have fixed it as recommended in the revision.

L7 Costello and Schmidt . is not included within the reference list.

Response: We have added the citation to the reference list.

L28-30: Omit.

Response: We appreciate your opinion and have omitted this sentence in the revision.

Page 14 L12-13: Results and no discussion.

Response: R2 and p-values have been removed in the revision. These results are meant to introduce the idea that while some individual abiotic soil factors may be correlated with specific bacterial phylum abundance, predictors may vary depending on the organism. This is discussed on Page 14 lines 29-31 and Page 15 lines 1-6.

L26 omit "(Stress : : :)

Response: It is the authors understanding that the stress value is a metric used to evaluate how well the ordination represents the original distances of the matrices, and should be reported when discussing the NMDS plot.

L28/29 Results and no discussion.

Response: R2 and p-values have been removed in the revision. These results highlight the relationship between the snow accumulation treatments and subsequent soil chemistry changes. A possible mechanism for shifting bacterial community structure is outlined on Page 15 lines 10-20.

Page 15 L1-2 Speculative. The authors did not measure enzyme kinetics. Omit "Rate" in "Rate of enzyme kinetics" since enzyme kinetics are the substrate-dependent rate of enzyme-substrate interaction.

Response: We have modified the sentence as follows, "The initial effects of increased snow pack result in altered physical factors (greater active layer thaw depth and increased soil temperatures and moisture; Blanc-Betes et al., 2016) which may lead to increased SOM availability and faster enzyme activities with the potential to enhance SOM decomposition. Higher SOM mineralization may promote the documented shifts in aboveground plant communities and increased NPP (Natali et al., 2012; Sturm et al., 2005,Anderson-Smith 2013), and vegetation shifts to more shrubby species may alter the chemistry and quality of new litter inputs, ultimately affecting decomposers." on Page 15 lines 10-16.

L7-8: Is this a logic relationship: increases in tannins and increases in N availability?

Response: The increases in tannins and N availability are meant to be two separate possible causes of reduced microbial activity in the DEEP zone, each of which are supported with references. However, in light of the comment above regarding insignificant difference in %N concentration, we have removed the concept of increasing N from the sentence.

L9 The decrease in C% might indicate exactly the opposite of the mechanism described above.

Response: To avoid confusion, we removed "%C and C:N" and replaced it with "relative abundance of genes required for SOM decomposition" on Page 15 line 26 of the revision.

L18: Are any data available about fungal biomass? Is the microbial biomass dominated by bacterial or by fungal biomass? (Did you investigate effects of snow depth on microbial biomass?)

Response: Unfortunately, we did not collect that data. However, we felt it important to acknowledge the role that fungi play in SOM decomposition, and provide a reason why we did not find peroxides, phenol oxidases, and laccases in our data.

L23-27: Does an interaction between tannins and extracellular enzymes necessarily induce shifts in gene abundances? Statement requires a reference.

Response: We have added the reference "(Rocca et al., 2014)" to Page 16 line 21 of the revision.

Page 16 L1-2: Does the decrease correspond to a decrease in C% or in C-stock or to a decrease in microbial biomass (as a factor for enzyme production)? Typically, enzyme activities are normalized with C%. Would this data treatment (gene abundances / C%) lead to a disappearance of treatment effects?

Response: Please see response to comment for Page 12 L18/19. We will examine how normalizing the gene abundances by %C effects our results.

L4-5 Sullivan 2008 is not included within the reference list

Response: Thank you for catching that. We have added the citation to the reference list.

L8-9: Is the system nutrient limited? Reference. N% increased in DEEP treatment - why did enzyme production not (that requires N and P)?

Response: Yes. Historically, the Arctic has been shown to be a nutrient limited ecosystem. We added the following references (Hobbie et al., 2002; Jonasson et al., 1999; Mack et al., 2004; Shaver and Chapin, 1980, 1986; Sistla et al., 2012) to Page 17 lines 3-4 and to the References section.

%N concentration did not significantly increase in the DEEP treatment, possibly explaining lack of enzyme production. Also, the increase in temperature may lead to decreased enzyme gene copies without altering enzymatic capacity from decomposers as explained in discussion, Page 17 line 22 – Page 18 line 3.

L12 Blanc-Betes et al. Submitted: not included in reference list

Response: We removed this citation from the revision. However, it does occur throughout the manuscript, so we have added the citation to the reference list.

L19: Which alternate energetic pathway?

Response: There are many possibilities, including fermentation, anaerobic respiration, and chemolithotrophy. The main point behind this sentence is to establish the idea that alternate and less efficient forms of metabolism may be selected for under these conditions. For clarification, the sentence on Page 17 lines 12-14 has been modified as follows"may select for microorganisms that use anaerobic metabolic pathways such as methanogenesis (Blanc-Betes et al. 2016). These hypoxic soil conditions would limit aerobic decomposition."

L21 change to "genes encoding enzymes involved in organic N degradation"

Response: This change has been made in the revision on Page 17 line 15 as follows:" genes encoding for enzymes involved in N mobilization,".

L22: Microbial biomass was frequently reported to be positively related to enzyme production and decomposition. Please, describe more precisely under which circumstances an in N availability and microbial biomass results in a decrease in decomposition rate.

Response: Bacteria in a N limited system must decompose SOM to gain access to more N, potentially increasing decomposition rates. As enzyme production for SOM decomposition is energetically demanding, there is a threshold (high microbial biomass and alleviated N limitation) where bacteria may switch to alternate sources of N, such

as microbial biomass, resulting in a decrease in SOM decomposition. This is supported by a theoretical C/N limitation model developed by Schimel, 2003. Also, In the Arctic, soil moisture is a confounding factor, as increased soil moisture may also decrease decomposition, as may be the case with the snow addition treatments. We have modified the text on Page 17 lines 14-21 to reflect these interactions referring to SOM in general rather than microbial biomass decomposition.

Page 17 L6 needs a reference - for example: Razavi et al. Front. Microbiol., 14 October 2015 | http://dx.doi.org/10.3389/fmicb.2015.01126

Response: This citation has been added as suggested.

L7-8 Speculative. If insitu substrate availability is low, Vmax will not be reached and enzyme functioning is controlled by Km (Michaelis-Menten constant). Alternatively, authors may change to "reach the same catalytic rate".

Response: We agree. Thank you clarifying this! The text has been updated to clarify these points on Page 17 lines 26-30 – Page 18 lines 1-3.

L26: needs reference

Response: This sentence has been removed from the revision.

L26-27 This statement should be reformulated in order to account for the low number of observed effects (the low number of repetitions) and the revised results of the KW-ANOVA (separate data from "organic" from those of "mineral soil").

Response: We added "From the results of our study" to Page 18 line 17. We added "in the organic soil horizon" to Page 18 line 19.

References IPCC should be shifted to I and not to W.(?)

Response: We moved the citation in the revision as suggested.

Table 1: Describe in Methods how many repetitions and analytical replicates were used

for the data. a> Or < b, please add information. No effect on N% - that differs from the description in the results section Why is there no effect on C:N in Table 1?

Response: We altered Page 7 lines 10-17 of the revision as follows: "Organic samples were collected just below where plant tissue transitioned into dark brown/black soil (mean soil depth = 5.6±1.3cm; CTL n=4, DEEP n=4, INT n=3, LOW n=4), transitional samples were taken from the visual border between organic and mineral horizons based on change in soil colour (mean soil depth = 14.8±1.8cm; CTL n=3, DEEP n=3, INT n=4, LOW n=3), and mineral samples were collected 10cm below this transition (mean soil depth = 25.1±1.7cm; CTL n=3, DEEP n=4, INT n=3, LOW n=3), totalling 41 samples. To maintain consistency, only these samples were used to analyse %C, %N, and pH relationships." We also added sample sizes to Table 1 in the revision We altered Page 10 lines 10-11 of the revision as follows: "...while the %N concentration only slightly increased (LOW/DEEP - p=0.32)." While it might seem from the numbers that there was a significant treatment effect on C:N, the p-value was 0.14. Please see Page 10 line 13 of the revision.

Figure 2: remove the ellipses from the graph. I counted 11 triangles but 10 circles etc. Why? How many sample points were included within the NMDS? NMDS requires detailed description within the results section. What are the main gradients observed? The NMDS optimized the illustration of the dissimilarity in beta diversity data but not in explanatory variables. Therefore, the combined illustration is somehow misleading (but the Mantel test is the appropriate method of choice).

Response: Please see response above. Alterations were made in the Methods of the revision to clarify replicates and sample sizes. We also altered the Figure 2 caption Page 33 lines 3-5 as follows "Each point represents the bacterial community structure within one of the 41 total samples used for DNA extraction from a variety of soil depths (Organic, Transition, and Mineral)." We have decided to keep the ellipses to better visualize the separation (or lack thereof) between the treatments.

Figure 3: No clear effect on bacterial phyla in organic samples (only some tendencies). Mineral soil: effect on Verrucmicrobia.

Figs 3 and 4: I would prefer to change the order of treatments within the graph (from low (left) to deep (right) and the use of the same scale (y-axis) for all panels.) Add information about the treatments to Figure captions. Fig 4 Superoxides +2 to -3% difference to control? This might be a very low effect.

Response: While we appreciate your opinion, we have decided not to change the order of the treatments. Also, we initially attempted using the same scale on the y-axis for all panels, however it resulted in loss of visual interpretation. Significant differences and trends became unnoticeable. We decided not to change the y-axis scales. We added the phrase "snow accumulation treatment" to the caption of both Figures.

---

## Author Comment (AC3) · 24 Apr 2016

Interactive comment on "Soil bacterial community and functional shifts in response to thermal insulation in moist acidic tundra of Northern Alaska" by M. P. Ricketts et al.

Anonymous Referee #3

The manuscript by Ricketts et al. addresses the effect of climate change predictions on the soil bacterial communities in Arctic tundra soils which are important global C sinks. The experiment was carried out in a long-term experimental field site offering a snow depth gradient from 25% lower to 100% more snow than the surrounding (control). Due

to increasing snow depth significant changes in abiotic soil parameters (e.g. active layer thaw depth, T, C/N ratio) were observed as well as a shift in the bacterial community structure. The taxonomic information from 16S rRNA amplicon sequence data was further used to estimate the functional gene abundance. These results indicate a decreased SOM decomposition potential under predicted climate change conditions which might help to further optimize current climate models. Therefore the study is of interest for readers of SOIL journal. Authors give a good overview about the current literature and make the aim of the study clear. In general, the manuscript is well-written and follows a logic flow. However, there are some points which were not clear to me and should be addressed before publication.

Response: We thank the reviewer for the summary and positive comments on the aim and approach of the study. We hope to address each of the comments in the revision to produce the best paper possible.

The prediction of microbial functional composition from phylogeny is really advanced and delivers new insights for studies where only 16S rRNA genes were sequenced. However, I would have liked to have first more information on taxonomic composition of bacterial communities in soil treatments and not only tests on six dominant phyla. The response of bacterial taxa belonging to the same phylum might be completely different. I recommend adding a table of significantly treatment-responding genera (some were already mentioned, p.12 section 4.1) or a heatmap showing relative abundance of dominant OTUs or genera across all samples.

Response: We will provide additional supplementary material to address this concern. There were 462 treatment responding Genera.

Furthermore, it should be emphasized in the discussion that all functional gene abundances are based only on predictions not taking into account horizontal gene transfer that might decouple function from phylogeny. Furthermore, there might be a lot of unknown functions due to poorly characterized taxa in Tundra soils or not yet known links

between taxon and function which should be discussed.

Response: This is a great comment and has been incorporated into the discussion of the revision on Page 16 lines 4-8 as follows: "While the use of PICRUSt and ancestral state reconstruction does not provide direct measurements of gene abundance (e.g., does not account for horizontal gene transfer or unknown functional / taxonomic linkages that may exist in the sampled tundra soils), it does offer valuable insights into the functional capacities of bacterial communities (Langille et al., 2013)."

By the way, it would be also interesting to know the amount of unclassified bacteria in your samples.

Response: Due to the nature of closed OTU picking (which is required for PICRUSt analysis), there were no unclassified bacteria in our samples. As per the QIIME website (http://qiime.org/tutorials/otu_picking.html), "In a closed-reference OTU picking process, reads are clustered against a reference sequence collection and any reads which do not hit a sequence in the reference sequence collection are excluded from downstream analyses." To clarify this in the revision we added, "Any read that did not match a sequence in the reference database was discarded." to Page 8 line 5. To satisfy your curiosity, we also did run open OTU picking, in which "reads are clustered against a reference sequence collection and any reads which do not hit the reference sequence collection are subsequently clustered de novo." (QIIME website). This resulted in 1.2% of unassigned bacteria. We did not use or include this method in the study as to maintain consistency between the bacterial abundance results and the PICRUSt functional gene abundance results.

In this respect, I am also wondering whether you tested first for availability of nearby genome representatives for your dataset before using PICRUSt prediction (NSTI index)? Furthermore, PICRUSt outputs a gene potential and it remains unknown to which extent these genes are expressed in the end.

Response: Please see above comment, and discussion on Page 18 lines 3-13.

As far as I understood, there are no treatment repetitions at the experimental site available and per treatment only 3 pseudo-replicates were taken. This makes it difficult to exclude the effect of natural variation in the bacterial community composition between sampling points. Thus conclusions can be drawn only very carefully and you should avoid to speculate too much in the Discussion section.

Response: We acknowledge the natural variation in bacterial communities at the micro-scale, and our limited replication. Through our revisions, we hope that our discussion and conclusions do not overstep their bounds.

These are my specific comments: Abstract: p.2, l.4 – Does "more or less snow" mean that predictions on amount of precipitation are not sure yet? Please consider rephrasing.

Response: We omitted "or less" on Page 2 line 4 in the revision.

p.2, l.11 – I recommend writing "Microbial community DNA was extracted from soil".

Response: We added "Soil microbial community" to Page 2 line 11-12 in the revision.

p.2, l. 15 – Taxonomic names should be written in italics (throughout the manuscript).

Response: We kindly disagree. As a rule, only genus and species taxonomic levels are italicized. All names of taxonomic levels above genus are capitalized, but not italicized.

Introduction: p.3, l.3 – Do you refer to belowground "microbial" community structure?

Response: This is meant to be a general statement. While there is evidence in the literature suggesting belowground macro community shifts, the citations provided primarily refer to microbial community shifts.

p.5, l.1 + l.16 – Please change microbial into bacterial community.

Response: These have been changed in the revision as suggested.

p.5, l.14 – bacterial functions (plural).

Response: This has been changed in the revision as suggested.

Methods: p.6, l.1 – How far away from the snow fence was the CTL sampled?

Response: We added ">30m" to Page 6 line 4 in the revision.

p.6, l.9 – What was the distance between the replicates per treatment?

Response: We added "approximately 15-20m apart" to Page 6 line 12 in the revision.

p.7, l.4 – I couldn't find the results of the bacterial communities from the transition zone; and was DNA extracted from all three replicated cores per treatment?

Response: Only phyla from the organic and mineral horizons were reported for bacterial abundance and gene abundance, while all samples (including transition) were included in the distance matrices, NMDS ordinations, and accompanying statistics. To clarify, we added, "Samples were analysed separately by soil horizon (Organic and Mineral only, not Transition)." to Page 9 lines 4-5 in the revision. Also we added "analysed between all samples, and organic and mineral horizons separately" on Page 9 line 18 of the revision. Yes, DNA was extracted from all three cores per treatment and per soil horizon. To clarify, we added "of each soil core," to Page 7 line 7 in the revision.

p.7, l.16- Could you please add the reference for the primers?

Response: Yes! We have added the appropriate reference in the revision.

p.7, l.17 delete – in Mi-Seq but add to p.8, l.7 Bray- Curtis.

Response: Thank you! We have made these correction in the revision.

p.8, l.6- Why did you determine adequate sampling depth? – I could not find that result later on.

Response: We removed, "rarefaction curves to determine adequate sampling depth," from Page 8 line 12 of the revision.

p.7, l.25/26 (and following pages)– "enzyme gene abundance" does not exist - please

consider rephrasing, e.g. " relative abundance of bacterial 16S rRNA and functional genes".

Response: We have reworded this in the revision as suggested.

p.8, l.27 – later there is also a significance threshold of p<0.1 used.

Response: Yes. We used a higher threshold in the figures to acknowledge and indicate where "marginal" significance was found.

Results: I assume that the main focus here is the comparison of each treatment (LOW, INT, DEEP) to the control plot. However, sometimes this is not clear to me from the type of statistical tests you did and from the description of the results. I recommend in general spending some more sentences to explain your results. You often miss to look at the n-fold change of rel. abundance to the control or the comparison between organic and mineral layer. If there is a significant difference to the control- was it observed in all treatments? Is it the same change (decrease/increase) in all treatments? e.g. p.10, l.3 – Instead of saying that the "C/N ratios became more similar" I would point out the %N trend is opposed in the organic and mineral layer. Furthermore I would suggest to add that differences in treatments are in general smaller in the mineral layer compared to organic layer.

Response: We have hopefully addressed these issues throughout the results in the revision. N-fold changes were added to the results. We deleted "and became more similar between treatments" from Page 10 line 15 of the revision. We added "although changes were less drastic than in the organic layers," to Page 10 lines 14-15.

p.10, l.6 – delete "and".

Response: This has been reworded as follows in the revision: "...all relative abundance of bacterial 16S rRNA and functional genes were analyzed by individual horizon." on Page 10 lines 17-18.

p.10, section 3.2. – Please point out the n-fold change of relative abundances. For

instance, the change in sequences affiliated to Chloroflexi is much stronger than for Actinobacteria.

Response: We have added n-fold changes to each reported result!

p.10, l.11 According to the text p=0.011 for Chloroflexi but there is only one * in Fig. 3.

Response: Thank you so much for catching that! We have fixed Fig. 3 in the revision.

p.10, l.13 Please consider rephrasing. I guess this is what you want to say: "acidobacterial abundance was in all treatments (DEEP, INT, LOW) lower than in the control".

Response: We have clarified this in the revision by adding, "while Acidobacteria showed decreased abundance in all treatments relative to the CTL (CTL/DEEP - p=0.055; Fig. 3)." to Page 10 lines 24-25.

p.10, l.19 please check p-value for Actinobacteria text vs. Fig. S4.

Response: Yes! Thank you. p-value corrected to "p<0.001" on Page 11 line 1 of the revision.

p.11, l.6/7 Were lignin, pectin and xylan degradation not significantly different to the CTL?

Response: No, they were only marginally significant. DEEP to CTL: lignin – p=0.119, pectin – p=0.100, and xylan – p=0.119.

p.11, l.14 Please rephrase – I agree it is also a decrease in genes coding for lignin degradative enzymes over the gradient but the scale differs and both LOW and INT have higher gene abundances compared to CTL in mineral layer in contrast to the organic layer.

Response: To acknowledge this observation, we added the following sentence at the end of the paragraph, Page 12 lines 1-2 in the revision: "However, relative to the CTL, both INT and LOW lignin-degrading genes exhibited much greater abundances than

they did in the organic horizon (Fig. 4)."

p.11, l.18 – Are Figs. S7-S9, S11 needed since they are not mentioned in the text?

Response: We believe that while these results are not significant, it is still important to include them in the supplementary information in order to maintain transparency.

p.11, l.23-25 - Please move this sentence to discussion and refer to Table S1.

Response: The Table S1 reference has been added to Page 12 line 10 in the revision. Also, the indicated sentence has been deleted from the results and the following sentence has been altered/added to the discussion, Page 16 line 9-11, "…and laccases (which are primarily associated with the degradation of lignin and other complex plant compounds) suggests that bacterial communities either preferentially degrade microbial biomass and polysaccharide polymers, or…"

Discussion: p.12, l.11 Please explain from which results the conclusion of "reduced SOM decomposition" was derived from!

Response: To clarify this, we added the following sentence to the revision Page 12 lines 25-27: "Our results indicate that increased snow pack reduced the abundance of genes associated with SOM decomposition in the organic soil horizon, suggesting a reduced SOM decomposition potential.."

p.12, l/12/13 – I don't agree with explanation 1) since you did not find differences in soil moisture along the gradient.

Response: Respectfully, there are many other factors besides moisture that contribute to O2 diffusion into the soil. In the organic horizons near the soil surface, compaction likely plays a role as well, especially in the DEEP zone where there is more snow pack. Also, no moisture differences were found because all measurements were close to 100% saturation, which may not reflect differences between submerged vs. non-submerged soil conditions. We have tried to clarify this in the revision.

p.12, l.30 – Actinomycetales are a bacterial order containing several taxa, thus please use plural.

Response: Good observation! We have changed the sentence to reflect the plural in the revision. "has" has been replace with "have" in Page 13 line 14 of the revision.

p.13, l.1 – Is the increase in Actinomycetales, that are linked to degradation of recalcitrant compounds, contradicting to your conclusions from functional predictions? Please discuss.

Response: The "stimulation of ... recalcitrant C degradation" was meant to be linked to the "stimulation of ectomycorrhizal growth...". To clarify this, we added "...which degrade..." and removed "...degradation" on Page 13 line 15 in the revision.

p.13, l.19- I suggest to delete Koyama et al. reference here because this is a totally different experiment. Instead cite Fierer et al. 2007 who tried an ecological classification of soil bacteria.

Response: We appreciate your suggestion. However, we feel this study provides a good example of oligotrophic vs copiotrophic competition in a nutrient limited vs non-limited environments. The Fierer et al. 2007 study is discussed in the following paragraph, Page 14 lines 17-22.

p.14, l.9 and following- I suggest to delete R2 and p-values from the discussion.

Response: We removed the R2 and p-values as suggested in the revision.

p.14, l.25 and following- I recommend to transfer results of Fig.2 to Results section. Regarding the replicate size of this study I suggest to be more careful here with conclusions.

Response: The sentence referred to was replaced by "The NMDS plot (Fig. 2)..." on Page 15 line 5 of the revision, and was revised and moved to the Results, Page 11 lines 9-12 of the revision as follows: "visualized by non-metric multidimensional

scaling (NMDS) plots of Bray-Curtis dissimilarity indices constructed from community matrices (Stress=0.090, Shepard plot non-metric R2=0.992; Fig. 2) revealed significant differences in community structure associated with winter snow pack"

p.15, l.14-27- The statements about fungi and tannic compounds are too speculative since this can not be supported by data from your study. Instead I would like to have a discussion of PICRUSt limitations here. Why is there only a decrease in rel. abundance of functional genes- which genes might be increased?

Response: The role of fungi and tannic compounds, while outside the scope of the data collected in this study, may help explain our results. Therefore, we feel that this paragraph should remain. The limitations of PICRUSt are acknowledged on Page 16 lines 4-8.

Tables & Figures: Table 1 I don't really understand the number of replicates you refer to here. n=4 are technical replicates? Significance was tested between treatments for each layer separately?

Response: We have altered the table caption to clarify. Page 31 lines 1-6 of the revision. We have also added specifics on the number of replicates throughout the Methods section.

According to Methods part you measured temperature at 4 different depths but not at 12 cm.

Response: This was our mistake and has been corrected in the revision Page 6 line 31 – Page 7 line 1.

Was there no post-hoc test done for %N, C/N, and pH?

Response: While the post-hoc test was done, they were not significantly different. Therefore, the subscripts were not used, as indicated by the revised caption "Results are indicated by a,b,c only where p<0.05." on Page 31 line 6 of the revision.
[Figure]

Figure 2 Difficult to understand. Microbial communities from how many replicates and layers are plotted here? Please use "CTL" as abbreviation for control (similar to the text).

Response: We altered the caption Page 33 lines 3-5 in the revision to clarify as follows: "Each point represents the bacterial community structure within one of the 41 total samples used for DNA extraction from a variety of soil depths (Organic, Transition, and Mineral)."

Figure 3 Please indicate for which significant effect you tested here. I don't understand, why there was no post-hoc test performed for Acidobacteria or is there only a significant difference to the control and not between treatments. The same applies for Fig. 4.

Response: You are correct in interpreting the significant difference to be to the control and not between treatments. This is indicated at the end of each caption by the phrase "except where significant differences were to the control."

Supplement Fig.S1-S15 From your Methods section it is not clear to me whether you analyzed abiotic soil parameters in the same soil sample (depth) as the bacterial community composition.

Response: To clarify this, we added the following sentence to Page 9 lines 11-12 of the revision. "To ensure accurate comparisons, soil chemical properties were measured from the same samples that DNA was extracted from."

References (typos): p.22, l13 Gonzalez-Meler

Response: Thank you! This has been fixed in the revision.

p.23, l34 mcrA in italics

Response: Thank you! This has been fixed in the revision.

p.27, l13 CO2

Response: Thank you! This has been fixed in the revision.

---

## Author Response (AR1)

In this manuscript, the authors describe the changes in bacterial community composition as a result of increased snow cover in a moist arctic Tundra. The study shows that increased snow cover led to changes in bacterial community composition along changes in soil chemistry and the plant community. The authors conclude that the observed changes in bacterial community composition and function might lead to reduced decomposition of SOM in these arctic systems. The manuscript is well written and structured and the story is for the most part easy to follow. After careful revisions the manuscript should be of great interest to the readership of SOIL. However there are some issues that need to be addressed or discussed in more detail to improve the manuscript.

> *Response:*
> *We would first like to thank the reviewer for the positive comments above and the constructive criticisms below. We will do our best to address them and are confident they will make the manuscript better.*

1. Soil depth: As the authors point out, that there is a huge difference in edaphic factors between organic and mineral horizons in this study. Such depth related differences have been shown to potentially influence microbial community structure and function and the potential controls on those (Eilers 2012 SBB, Schnecker 2015 SBB). The authors should also test the effects of depth as well as treatment and potential interactions on the individual bacterial groups, their relations to soil factors and beta diversity using Adonis and perform the mantel tests with the edaphic factors separately for organic and mineral horizons.

> *Response:*
> *Samples were analysed separately by soil layer / depth (Organic vs. Mineral) for most factors included in this manuscript (%C, %N, C:N, pH, bacterial abundance, predicted gene abundance). We have added Table 2 to the revision to report statistical differences in beta diversity, including separate analyses for each soil depth. We have also added Table S3 to the supplementary revision to report statistical differences in alpha diversity both between soil layers and between treatments. In addition, we did analyse statistical differences between soil depths / layers for each of the six most abundant phyla and for each enzyme gene abundance. We have added this data in Table S2 of the supplementary information. All new tables are briefly described and discussed in additions made within the text. While we believe these additions are beneficial to the manuscript, our primary goal for this paper was to address the effects of increasing snow pack on soil bacterial*

*communities regardless of soil depth. Therefore, in the text we chose to highlight the treatment effect over the depth effect.*

*Changes made to manuscript:*
- *Text added to Page 8, lines26-27.*
- *Text added to Page 8, line 29 – Page 9 line 2.*
- *Text added to Page 9, line 12.*
- *Text revised / added to Page 9, lines 14-18.*
- *Text revised / added to Page 11, lines 6-12.*
- *Text added to Page 11, lines 20-23.*
- *Text added to Page 13, lines 18-20.*
- *Text added to page 15, lines 23-24.*
- *Text added to page 15, lines 27-28.*
- *Text added to page 17, lines 15-16.*
- *Table 2 added to Page 33, lines 1-6*
- *Table S2 added to Supplementary Information Page 11.*
- *Table S3 added to Supplementary Information Page 12.*

2. Vegetation and decomposition: The authors state that an increased snow cover ultimately leads to reduced decomposition and C loss from the system since NPP is increased and might offset potential losses of C. While their results show a reduced potential for decomposition in the bacterial community and other studies have found increased NPP in shrubby tundra compared to tussock tundra, the C contents in organic and mineral horizons decreased significantly. This huge loss could have either happened during the transition from tussock to shrubby vegetation, which would mean that NPP did not offset decomposition or during the transition into a sedge dominated fen, which would indicate that decomposition was not reduced despite the reduction of the bacterial potential for decomposition.

*Response:*
*Carbon content is not a good indicator of carbon-stock. Degrading permafrost often results in soil consolidation (loss of ice collapses soils) with associated changes in bulk density and depth redistribution of soil and C. The C-stock profile change as a result of the snow fence treatment is part of another paper. Carbon stock over the soil profile to the average active layer equivalent depth was 7% higher for the intermediate than for the control.*
*Here we have used %C in our analyses because most of the C (if not all) is accessible to microbes (these acidic tundra soils have little to no physical aggregation, JD Jastrow personal communication). Therefore the factors affecting organic matter readiness to microbial decomposition is likely the chemistry/quality of the organic matter (%C, C/N) in addition to temperature and moisture. We hope this is made more clear in the revised manuscript.*
*Also, we acknowledge that the use of the word "content" when referring to our %C data may have been misleading. To clarify, we have changed the phrase "C (or N) content" (i.e. C stock) to "C (or N) concentration" throughout the manuscript.*

*Changes made to manuscript:*
- *Change made Page 2 line 11*
- *Change Page 6 line 19*
- *Change Page 10 lines 7-8*
- *Clarification and changes made to Page 15 lines 5-10.*

3. Fungi and oxidative enzymes: The authors should more strongly point out that this study is focused on bacterial community composition and function throughout the text and that fungi might play an important part especially in the production of oxidative enzymes which have been found in arctic soils (Tveit 2012 ISMEJ).

>*Response: The role of fungi, while not highlighted in this study, is acknowledged and discussed on Page 16 lines 8-11.*
>
>*We changed "microbial" to "bacterial", or "microorganisms" to "bacteria" throughout the manuscript:*
>
>*Changes made to manuscript:*
>- *Page 4 line 30*
>- *Page 5 lines 4, 8, 11*
>- *Page 12 lines 26*
>- *Page 15 linse 18 & 27*
>- *Page 16 line 10*
>- *Change to Page 17 line 21*
>- *Change to Page 33, Figure 2, line 4*

4. The authors should be more careful with the interpretation of the ancestral state reconstruction, since these results are strictly based on the sequencing results of the bacterial community. Changes in the so obtained functions can only be interpreted as changes in the bacterial community composition. Any statements concerning enzyme kinetics, enzyme transcription, activity or even in situ functional gene copy number can only be speculated on and should be clearly marked as speculation (especially Page 17 Lines 1-19)

>*Response:*
>*This is well noted and care was taken to revise the manuscript with it in mind, including the following:*
>
>*Changes made to manuscript:*
>- *Addition to Page 16 lines 21-31*
>- *We have added "predicted" to Page 17 lines 15 amd Page 18 line 16.*

5. The authors should consider that any changes in the bacterial community composition could be independent of SOM properties and be a result of changes in temperature, moisture vegetation length and so on and could vary with depth (Schnecker 2014 Plos One, Gittel 2013 ISMEJ)

>*Response:*
>*We agree. However, untangling what is driving bacterial community shifts in this system requires isolation of these different factors in a laboratory setting, which is outside the scope of this paper, however we are planning some of these experiments. The very nature of our experiment, altered snow pack over a long period of time, changes a variety of factors that may contribute to bacterial community change (O2 diffusion via moisture or compaction, temperature, plant community, etc.).*

**Detailed comments:**
**Title:**
Since multiple environmental factors are changed with increased snow cover, "thermal insulation" should be replaced with "altered snow cover" or similar.

> *Response:*
> *Agreed! We replaced "thermal insulation" to "altered snow pack" in the revision.*
>
> *Changes made to manuscript:*
> - *Page 1 line 2*

**Introduction:**
Page 3 Lines 17-23: Since there is another paragraph on SOM in the arctic this one could be omitted. Especially since the numbers for global C storage here and in the paragraph on arctic C storage are not the same.

> *Response:*
> *While we appreciate the comment, it is important to recognize that the paragraph being referred to highlights the important role that microorganisms play in C cycling on a global scale versus the later one that specifically describes the significance of C in Arctic ecosystems. We have modified the sentence to highlight global vs. arctic C dynamics. We added "on a global scale, releasing nutrients…" on Page 3 lines 15-16.*
>
> *Changes made to manuscript:*
> - *We added "on a global scale, releasing nutrients…" on Page 3 lines 15-16.*

Page 4 Line 1-2: The nutrient limitation of Arctic soils has recently been challenged (Wild 2015 GBC, Melle 2015 SBB).

> *Response:*
> *We have removed the sentence from the manuscript.*
> *However, while this may be the case in more recent studies, as rising temperatures in the Arctic may be initially accelerating SOM decomposition and releasing more nutrients, historically the Arctic has been observed to be a nutrient limited ecosystem (Hobbie et al., 2002; Jonasson et al., 1999; Mack et al., 2004; Shaver and Chapin, 1980, 1986; Sistla et al., 2012).*
>
> *Changes made to manuscript:*
> - *Sentence removed from Page 4 lines 1-2.*

Page 5 Lines 5-18: The Authors should consider using testable hypotheses instead. Structure and O2 availability were not measured in this study. The change in plant species composition might not be a consequence of increased nutrient availability but the result of changed water status. With the experimental setup it cannot be distinguished between substrate effects and environmental effects.

> *Response:*
> *This is a valid point. Changes in plant species (and underlying causes), while proposed as a contributing factor to bacterial shifts, was not the focus of this study, and therefore we chose not to test for causes of vegetation shifts within our study site.*

*Changes made to manuscript:*
- *None.*

**Material and Methods:**
Please mention which program was used to perform the statistical analyses. As mentioned before the measured parameters, including beta diversity should be tested for depth effects and interactions of depth and treatment. All correlative tests should also be performed separately for *organic and mineral horizons.*

*Response / Changes made to manuscript:*
- *We added "...in the R statistical software package..." on Page 8 lines 28-29 of the revision.*
- *We added Table 2 to report beta-diversity statistics, both for all samples (All layers), and separated by soil depth (organic and mineral) on Page 33 lines 1-6.*

**Results:**
Page 10 Lines 26- Page 11 Line 2: These results should be presented in a separate table.

*Response / Changes made to manuscript:*
- *These results are included in Table 2 of the revision Page 33 lines 1-6.*

Page 11 Lines 10-11: The reported p-values are not significant.

*Response:*
- *While the p-values are not significant, we felt it was still important to acknowledge notable trends with p-values <0.1 as long as the p-values are reported.*

*Changes made to manuscript:*
- *None.*

Page 11 Lines 24-25: This interpretation should be moved in the Discussion section of the manuscript and "microbial communities" should be replaced with "bacterial communities".

*Response:*
*"Microbial" will be replaced with "bacterial" in the revision.*

*Changes made to manuscript:*
- *See comments for general comment # 3.*
- *The indicated sentence has been deleted and replaced on Page 17 lines 1-5.*

**Discussion:**
Page 12 Line 11: As stated before, while the bacterial functional potential might indicate reduced SOM decomposition, the decrease in C content from control to DEEP suggests otherwise.

*Response:*
*Please see response to general comment 2 above.*

Page 12 Lines 12-17: An alternative explanation might be that the microbial community composition is shaped by the environmental factors and less so by SOM properties.

*Response:*
*This is a good point. However, this paragraph was meant to discuss the functional shifts as opposed to the phylogenetic shifts. See below changes.*

*Changes made to manuscript:*
- *Addition of the word "functional" to Page 13 line 9 of the revision.*

Page 12 Line 16: Blanc-Bates et al. 2015 is missing in the Reference list. Is this the same that is listed as submitted in Page 16 Line 12. If this is the case and if this study was conducted at the same site, mentioning this and a short description of the findings would help the reader understanding the author's arguments about changes in SOC dynamics.

*Response / Changes made to manuscript:*
- *This citation has been removed.*

Page 13 Line 26: The strong correlation of Acidobactera with pH and the non-significant correlation with C:N questions that statement.

*Response:*
*This sentence highlights competitive interactions between bacterial phyla. Any correlation with abiotic factors in the context of this sentence would be indirect.*

*Changes made to manuscript:*
- *None.*

Page 13 Lines 28-30: This sentence can be omitted.

*Response: We appreciate your opinion and will omit this sentence in the revision.*

*Changes made to manuscript:*
- *This sentence has been omitted on Page 14 line 27.*

Page 14 Line 28: This could be a depth effect and not a result of the altered snow cover.

*Response:*
*The effects of the snow accumulation treatment are statistically significant in the organic horizon alone (p=0.001), the mineral horizon alone (p=0.003), and within all samples (p=0.017; please see Table 2 in the revision). Therefore, we are conservative in reporting that snow accumulation affects bacterial community structure with a p-value of 0.017.*

*Changes made to manuscript:*
- *Table 2 added on Page 33 lines 1-6*

Page 15 Lines 7-11: Is there any indication that increased tanning occurred at the studied site?

*Response: We did not test for tannin concentration in this study, however the encroaching shrub species at the site are known to produce them (DeMarco et al., 2014; Schimel et al., 1996).*

*Changes made to manuscript:*
- *None.*

Page 15 Lines 23-27: Binding of enzymes to tannins could happen to any enzyme. Oxidative enzymes could actually degrade tannins and might thus be upregulated.

*Response:*
*This is a valid point. However, if the bacterial community does not possess the functional capacity to produce these oxidative enzymes (as suggested by our data), they will not be able to increase their production. As mentioned in Page 15 lines 16-21, fungi may perform this role in this system, and thus would not show up in our 16S rRNA gene analysis.*

*Changes made to manuscript:*
- *None.*

Page16 Line 13: Sistla et al 2013 did not use a snowfence study.

*Response / Changes made to manuscript:*
- *We added "…warming and…" to Page 17 line 29 of the revision.*

Page 16 Lines 13-28: While this study explains to some extent some of the author's statements, it is over represented for its current publication status.

*Response:*
*This citation and description of the study has been removed from the revised manuscript.*

*Changes made to manuscript:*
- *Major revisions have been made from Page 17 line 28 - Page 18 line 12.*

Page 17 Lines 1-19: see general comments above.

*Response:*
*Please see response to general comment 4 above.*

Ricketts et al. examined an interesting question: How do changes in snow cover affect soil bacterial community structure and function? They sampled and analyzed soil material from an interesting long-term snow depth manipulation experiment and applied up-to-date methods for bacterial community characterization. They claimed out that an increase in snow depth resulted in an increase in soil insulation that led to changes in bacterial community structure, to a decrease in enzyme encoding genes and in C%. The MS presents results of a relevant experiment and the topic is within the scope of the Journal.

> *Response:*
> *We thank the reviewer for the encouraging summary and for the criticisms which are addressed below.*

General comment #1- The Introduction is well written. However, in my opinion more information about the experimental set-up and the sampling design are required in the Methods section. How did they ensure that the snow depth was continuously increased for 100%, 50%, or decreased for 25% along the study period of 18 yrs? Did they continuously measure snow depth each year? Did they remove or add snow in cases with more or less than e.g. 100% than control? Or is the treatment rather a distribution of increased/decreased snow depth around e.g. 100% (+/- SD) than a fixed treatment level? How did they monitor the annual input of snow at each point? Was it equal for all years?

> *Response:*
> *Snow treatments are caused by the wind drift distance from the snowfence. Snow fall varies from year to year but the drift caused the relative snow accumulation at similar distances from the fence every winter. Snow depth and density have been measured sporadically and reported in other papers which we cite.*
>
> *Changes made to manuscript:*
> *- Addition to Page 5 lines 23-251.*

General comment #2 - The applied non-parametric statistics seem to be appropriate for the experiment and data set, however, re-analysis of the data might be necessary because organic and mineral soil samples seem to be included within one analysis (without accounting for differences in sampling depth). In my opinion, too many results were mentioned as significant effects even though the p values were above 0.05. This is problematic and partly lead to a rather speculative discussion and conclusion. To sum up, I recommend thorough revision of the manuscript in order to focus on the observed effects of snow depth on soil bacterial communities.

> *Response:*
> *Thank you for this suggestion, one that has also been pointed out by other referees. We have included new stats and language to distinguish the soil depth analyses and added p*

*values to let the reader decide the statistical significance of results. We have noted results with marginal significance of p<0.1 in addition to the p<0.05.*

*Changes made to manuscript:*
*\* Please see response and changes made to anonymous referee #1 general comment #1.*
- *Text added to Page 8, lines26-27.*
- *Text added to Page 8, line 29 – Page 9 line 2.*
- *Text added to Page 9, line 12.*
- *Text revised / added to Page 9, lines 14-18.*
- *Text revised / added to Page 11, lines 6-12.*
- *Text added to Page 11, lines 20-23.*
- *Text added to Page 13, lines 18-20.*
- *Text added to page 15, lines 23-24.*
- *Text added to page 15, lines 27-28.*
- *Text added to page 17, lines 15-16.*
- *Table 2 added to Page 33, lines 1-6*
- *Table S2 added to Supplementary Information Page 11.*
- *Table S3 added to Supplementary Information Page 12.*

At least four references cited in the text are not included in the reference list.

*Response:*
*We have reviewed our citations and added references in the revision.*

*Changes made to manuscript:*
- *Please see additions / changes in References section, Pages 21 – 30.*

Please, find my specific comments below:

**Abstract**
Line 4-5 "(i.e. more or less snow), resulting in increased winter insulation" This statement is partly contradictive. Omit "or less" or add "increased or decreased winter insulation".

*Response:*
*Agreed! Please see changes below.*

*Changes made to manuscript:*
- *We omitted "or less" on Page 2 line 4 in the revision.*

L8 "context of ecosystem response to climate change." Please change to "context of expected ecosystem response to : : :"

*Response:*
*We think this is a valid point and have changed it in the revision, as recommended.*

*Changes made to manuscript:*
- *Revised as recommended on Page 2 line 8.*

L15 "most abundant phyla" requires a value about the contribution of these phyla on total abundance. "20% or 80% of total detected phyla?"

*Response:*
*This is a good idea! Please see changes below.*

*Changes made to manuscript:*
- *We added "…(ranging from 82% to 96% of total detected phyla per sample)…" to Page 2 lines 15-16 in the revision.*

L26 The authors did not study the temperature sensitivity of extracellular enzymes (sensu str.) they are requested to omit any statement/conclusion about this.

*Response:*
*Thank you for your comment. However, we feel that while this sentence is speculative, it is an important possible mechanism explaining our results and is supported by outside literature (Conant et al., 2011; Davidson and Janssens, 2006; Lützow and Kögel-Knabner, 2009).*

*Changes made to manuscript:*
- *None.*

**Introduction**
**Page 3**
L 1: How can the stability of the structure be threatened? I suggest to change to more ecological terminology.

*Response / Changes made to manuscript:*
- *We have changed the wording of this sentence to "Broad and rapid environmental changes are driving both above- and belowground community shifts in the Arctic" on Page 3 lines 2-3 of the revision.*

L8 Anisimov and Vaughan must be changed to Anisimov et al. or the respective reference must be added to reference list.

*Response:*
*Thank you! This was intended to be the same reference! We have fixed it in the revision.*

*Changes made to manuscript:*
- *Fixed reference on Page 2 line 11 of the revision.*

L11 Needs reference

*Response / Changes made to manuscript:*
- *We have added (Anisimov et al., 2007; Liston and Hiemstra, 2011) to Page 3 line 11 of the revision.*

L12 Needs reference

*Response / Changes made to manuscript:*
- *We have added (Hugelius et al., 2013; Ping et al., 2008; Schuur et al., 2009; Tarnocai et al., 2009) to Page 3 lines 12-13 of the revision.*

**Page 4**
L16 omit activity or kinetics – I prefer the use of the term "kinetics".

*Response / Changes made to manuscript:*
- *We have omitted "…activity and…" from Page 4 line 14 of the revision.*

**Page 5**
L11 Why should microbes be unable to degrade SOM from shrubs? Needs further explanation or changing in the sense of" the potential to degrade SOM might be reduced".

*Response / Changes made to manuscript:*
- *This sentence has been deleted from the revised manuscript on Page 5 line 7.*

**Methods**

Are detailed vegetation surveys available for each plot? How far away were the replications at each treatment located from each other? Did they sample more than one soil core for each replicated plot and compiled composite samples or not? Figure 1 is important and helps to understand the experimental set-up. However, where is the control located?

*Each of these of these questions have been addressed separately. Please see below:*

*Response:*
*We did not collect detailed vegetation surveys for each plot.*

*Changes made to manuscript:*
- *None.*

*Response:*
*Replicate distance = estimated ~15-20m.*

*Changes made to manuscript:*
- *We added "approximately 15-20m apart" to Page 6 line 7.*

*Response:*
*Clarification of number of cores/compiled*

*Changes made to manuscript:*
- *Added "replicate" and "and analyzed separately" to Page 6 line 8. Added "of each soil core" and deleted "soil" to Page 7 line 2.*

*Response:*
*For the location of the control, please see below.*

*Changes made to manuscript:*
- *Added ">30m" to Page 5 line 28.*
- *Added "Three soil cores were obtained from each treatment zone, labelled Deep, Intermediate, and Low, and a Control zone located >30m outside the effect of the snowfence." to the caption for Figure 1, Page 31 lines 3-5.*

**Page 5**
L24 "strategically" needs further explanation.

> *Response:*
> *The word "strategically" refers to its orientation based on wind patterns and is*
> *described in the papers we referenced (Jones et al., 1998; Walker et al., 1999).*
> *Also, please see response to general comment #1 above.*

> *Changes made to manuscript:*
> - *None.*

L29-30 Soil Survey 2015 is not listed in references

> *Response / Changes made to manuscript:*
> - *We added "Soil Survey Staff, Natural Resources Conservation Service, United States*
>   *Department of Agriculture. Web Soil Survey. Available online at*
>   *http://websoilsurvey.nrcs.usda.gov/. Accessed [11/17/2015]." to the references, Page*
>   *28 lines 27-29.*

**Page 6**
L1 "regime": the tested climate change scenario is: variable precipitation (that may induce
differences in soil temperature) but constant air temperature.

> *Response:*
> *Please see response to general comment #1 above.*

P6 treatment/factor levels: -25% vs. +50% (vs. +100%) are not equally selected. This might be
problematic for ANOVA. Please check.

> *Response:*
> *For our analysis, each treatment was defined as a categorical variable, thus avoiding*
> *any numerical percentage gradient. This is the more conservative approach.*

> *Changes made to manuscript:*
> - *None.*

L9: n=3? Total number of sampled cores = 12?

> *Response:*
> *Yes. 3 – CTL, 3 – DEEP, 3 – INT, 3 – LOW = 12 cores.*

> *Changes made to manuscript:*
> *\*Alteration throughout the Methods section and figures were made to be more*
> *transparent on sample sizes.*
> - *Page 6 line 25*
> - *Page 7 lines 5-12*
> - *Page 32 lines 5-7*
> - *Page 33 lines 4-6*

L18: unclear, how did they use the 2 cm depth segments for further analysis since they presented
data for "organic" and "mineral" soil only. Did they calculate the average value of C% etc. for

each of the two strata by considering the data of the single segments?

> *Response:*
> *We only analysed %C, %N, and pH for the samples used for the DNA extractions.*
> *Clarifications were made in the revised manuscript. See below.*
>
> *Changes made to manuscript:*
> - *Added "To maintain consistency, only these samples were used to analyse %C, %N, and pH relationships." to Page 7 lines 11-12 of the revision.*
> - *Added "To ensure accurate comparisons, soil chemical properties were measured from the same samples that DNA was extracted from." to Page 9 lines 7-9 of the revision.*

**Page 7**

L 11: Please provide the absolute sampling depths for each treatment (average value and variation) in the Methods section. Sampling depth might be considered as co-variable in non-parametric ANCOVA in order to account for any effect.

> *Response / Changes made to manuscript:*
> - *Changed ",typically between 0-6cm soil depth, except in one case where the top 10cm was primarily plant tissue" to "(mean soil depth ± standard error [S.E.] = 5.6±1.3cm)" in Page 7 lines 6-7.*
> - *Changed to "(mean soil depth ± S.E. = 14.8±1.8cm)." on Page 7 lines 8-9.*
> - *Changed "...and was more variable (ranging from 15-36cm soil depth) due the varying depths of transition." to "(mean soil depth ± S.E. = 25.1±1.7cm)" in Page 7 line 10.*

L22 Caporaso et al is not included in the reference list

> *Response / Changes made to manuscript:*
> - *We added this to the references in the revision, Page 21, lines 24-29.*

**Page 8**

L 4: "six most abundant phyla" this requires a quantitative documentation for the six phyla.

> *Response / Changes made to manuscript:*
> - *We added ",comprising 82% - 96% of total detected phyla per sample," to Page 8 lines 5-6.*

L 18-20 needs reference(s)

> *Response / Changes made to manuscript:*
> - *Added "(Sinsabaugh et al., 2008; Waldrop et al., 2010;..."to Page 8 line 20 of the revision.*

L 24 – P9 L 14 Selected types of statistical analysis seem to be appropriate for the experimental set-up and data. (Maybe non-parametric ANCOVA is required for the consideration of sampling depth as co-variable). However, P9 L1 the selection of linear regression analysis is inconsistent since the authors applied non-parametric tests for the data. Taken into account that the primary assumptions for parametric tests are not full-filled (they used the non-parametric tests) then the

use of linear regression seems not to be adequate. In addition, the use of median and median absolute deviation might be more robust estimates (and consistent) of the central tendency and variation of the data than mean and SD.

> *Response:*
> *This is a valid point and we thank the reviewer for pointing it out. The linear regression analyses were initially performed simply to give us an idea of how the abiotic soil data affected individual bacterial phylum, and gene group abundances. While there are many interesting relationships, we ultimately decided to report the more appropriate non-parametric stats for the bulk of the analyses, but to still include the regressions in the supplemental material.*

**Table 1:**

The lower case letters used indicate that organic and mineral soil material was included in one analysis. (?) I suggest comparing the treatment effects on the two strata (organic and mineral) independently of each other (for both KW-ANOVA and Nemenyi-test; performing the respective tests for the treatment effect on e.g., C% of "organic"). Add a brief description of the treatments (-25% of snow cover compared to control etc.) to the table description.

> *Response:*
> *For clarification, the organic and mineral layer samples were analysed independently as suggested. See changes that were made below.*
>
> *Changes made to manuscript:*
> - *We have removed the lower case letters "c" from the %C column in Table 1, Page 32 line 7, since that analysis was not part of the organic horizon analysis, and there were no statistical differences in the mineral horizon.*
> - *We added, "Organic and mineral samples were analysed separately using the Nemenyi post hoc test. Results are indicated by $_{a,b,c}$ only where $p<0.05$." to Table 1 description, Page 32 lines 5-6.*
> - *We added, "(Low = ~25% less snow pack than the Control, Int = ~50% more snow pack than the Control, Deep = ~100% more snow pack than the Control)." to Table 1 description, Page 32 lines 1-3.*

**Page 9**

L7-8: two-sample t-test for the comparison of four groups? I do not understand. The selected measure of dissimilarity as well as the criteria for NMDS seems to be adequate / sufficient. How many dimensions were included / considered for NMDS?

> *Response:*
> *The two-sample t-tests were used to perform pairwise comparisons of the treatment groups. The ordination of the Bray-Curtis distance matrices using NMDS yielded two convergent solutions found after four tries, resulting in 2 dimensions.*
>
> *Changes made to manuscript:*
> - *We replaced, "The Shannon alpha diversity metric was compared across treatments using…" with, "Pairwise comparisons of the Shannon alpha diversity metrics from each treatment group were made using…" in Page 9 linse 12-13.*

L10-11: Do I understand right? The analysis of associations between explanatory variables (i.e.,

soil chemical properties) and bacterial data were further analyzed by Mantel test? Which statistical software was used?

> *Response:*
> *Yes. The QIIME script compare_distance_matrices.py was used with --method=mantel to compare the Bray-Curtis distance matrices to distance matrices created from the abiotic data using the QIIME script  distance_matrix_from_mapping.py.  The QIIME software was used to run these tests.*
>
> *Changes made to manuscript:*
> - *We altered the sentence on Page 9 lines 10-12, as follows: "Bacterial diversity statistics were calculated using QIIME (Caporaso et al. 2010), specifically the compare_alpha_diversity.py, compare_categories.py, and compare_distance_matrices.py scripts."*

**Results**
L18  I thought the test statistic of KW-ANOVA is the "H-value" and not Chi2? Please check. Please calculate the effect sizes for each tested factor and variable.

> *Response:*
> *Yes, you are correct.  Confusion arose because of the way the R package reports the values for kruskal.test ("Kruskal-Wallis chi-squared = 0.80317, df = 3, p-value = 0.8487").  After checking, we discovered that this value is indeed the Kruskal-Wallis test statistic, otherwise known as "H", although it has been reported both ways.*
>
> *Changes made to manuscript:*
> - *We have corrected this in the revision.  Page 9 lines 26-27.*
> - *Sample sizes were added to clarify effect size on Page 9 lines 26-27 of the revision.*
> - *"(n=12 per treatment)" was added to Page 6 line 25 in the revision.*

L30: p=0.32 I would not consider this as a significant difference.

> *Response:*
> *While not considered significant, the increasing trend does contribute to decreasing C:N.*
>
> *Changes made to manuscript:*
> - *To address this comment, we added "only slightly" to Page 10 line 8 of the revision.*

**Page 10**
L1: p=0.14 I would not consider this as a significant difference.

> *Response:*
> *While the result is insignificant, we feel that the overall trend contributes to the story.*
>
> *Changes made to manuscript:*
> - *We have reworded the sentence as follows: "This resulted in a decreasing trend in C:N ratios across snow accumulation treatment zones and relative to the control (CTL/DEEP - p=0.14; Table 1)." on Page 10 lines 8-10 of the revision.*

L2 p=0.06 indicates a tendency

*Response / Changes made to manuscript:*
- *We added the phrase "tended to" to Page 10 line 10 of the revision.*

L11 Verrucomicrobia and Actinobacteria p-values indicate tendencies. Section 3.2. in many cases the order of magnitude of the relationships were rather low.

*Response:*
*These comments are true, and as such, we attempted to phrase our results in a way that openly and accurately reports the data, without dismissing trends that may or may not be the result of the treatment or abiotic soil conditions. Hopefully, this will allow readers to be able to openly interpret the data. We have added the below changes to more accurately address the orders of magnitude of the relationships.*

*Changes made to manuscript:*
- *We added n-fold changes to all appropriate results in the revision, from Page 10 line 19 –to- Page 12 line 11.*

**Page 11**
L5-8 p-values of which comparisons? CTL to DEEP or LOW to DEEP. Or do the p-values represent the results of the KW-ANOVAs?

*Response:*
*Thank you for catching this confusion. The first two, cellulose and chitin, were comparing the DEEP zone relative to the CTL. The latter three were comparing across treatments from LOW to DEEP.*

*Changes made to manuscript:*
- *We added, "in the DEEP zone" to Page 11 line 28.*
- *Altered and separated the sentence into two to clarify, Page 11 line 28 –to- Page 12 line 1.*

L10 "N mobilization genes" Do the genes mobilize N? Please, correct.

*Response / Changes made to manuscript:*
- *To clarify this, we have altered the sentence as follows: "Shifts along the snow accumulation gradient were also observed in gene groups involved in nutrient mobilization with a 1.18-fold increase in genes necessary for N mobilization (p=0.14) 1.12-fold decrease in genes necessary for phosphate mobilization (p=0.39) in the DEEP zone relative to the CTL." on Page 12 lines 3-6.*

L12-13 It might be more meaningful to write as follows: ": : : included an increase in genes encoding enzymes involved in : : :".see P15 L16-17

*Response:*
*Great recommendation!*

*Changes made to manuscript:*
- *Altered as recommended, Page 12 lines 7-8 of the revision.*

L25: omit "simple cellulosic and"

> *Response / Changes made to manuscript:*
> - *This entire sentence has been removed from the manuscript, Page 12 line 21..*

**Page 12**

L10: only moderate changes and in many cases they observed only trends. Please, focus on significant results p<0.05.

> *Response:*
> *The result from the adonis test (p=0.017) supports the claim that community structure is significantly affected by the snow addition treatment. Additionally, this sentence (while broad is scope), is not false. The significant results we do find still do indicate a "change". The remainder of the discussion focuses on these significant results, occasionally using the trends in the data to speculate cause.*
>
> *Changes made to manuscript:*
> - *To address this confusion, we replaced the word "phylogeny" with the phrase, "community structure" in Page 13 line 5.*

L12: "towards more labile sources" if it is an important "pathway" then the term/concept "labile" requires definition in the introduction section.

> *Response:*
> *While the term "labile" may not be familiar to a broad audience, we feel that it does not require defining in the context of this journal.*

L13: What is "SOM enzyme activity"?

> *Response / Changes made to manuscript:*
> - *We replaced " decreased SOM enzyme activity" with "a decreased abundance of genes associated with SOM decomposition" in Page 13 line 11.*

L14: The positive relationship between gene copies and enzyme machinery requires a reference.

> *Response:*
> *We're glad you noticed that! We have considered this and it is acknowledged and discussed with references on Page 17 lines 24-26.*
>
> *Changes made to manuscript:*
> - *We also added the reference "(Rocca et al., 2014)" to Page 13 line 12 and Page 17 line 12.*

L15/16 Which limitations of enzyme kinetics? Change "enzymatic decomposition reactions" to "enzyme functioning" . Blanc-Betes et al. is not included within the reference list

> *Response / Changes made to manuscript:*
> - *We have changed the wording as recommended on Page 13 line 14 in the revision.*
> - *Blanc-Betes reference has been removed from this sentence, Page 13 line 14.*

L18-19 But in the present account a decrease in C% was observed. Discuss this issue in contrast to the results in the literature. Is the decrease only found in C% or also in C-stock?

> *Response:*
> *Please see response to anonymous referee #1 general comment #2.*
>
> *Changes made to manuscript:*
> - *This sentence was removed from the revision, Page 13 line 15.*
> - *Please see changes from anonymous referee #1 general comment #2.*

L22/23 Significant changes were observed only for a few groups.

> *Response:*
> *While this is true, the sentence refers to "bacterial community structure" (or beta diversity), which is significantly altered by the treatment, as per the adonis test (p=0.017). In other words, just because significant results were only obtained for a few of the most abundant phyla, the overall community structure (which takes into account ALL organisms/OTU's) does significantly change due to increased snow depth.*
>
> *Changes made to manuscript:*
> - *None.*

L28/29:  What are the possible links to enzyme production and functioning?

> *Response:*
> *Enzymes involved in the utilization of plant biomass and SOM are described in the discussion, Page 16 lines 22-26.*
>
> *Changes made to manuscript:*
> - *None.*

L29: Sangwan et al. is not included within the reference list.

> *Response:*
> *Thank you for catching that. We have added the citation to the reference list.*
>
> *Changes made to manuscript:*
> - Citation added to Page 27 lines 18-21.

**Page 13**

L7 "cold saturated soil" Did you mean "cold, water-saturated soil"?

> *Response:*
> *Yes, we did! We have fixed it as recommended in the revision.*
>
> *Changes made to manuscript:*
> - *Added ", water-saturated" to Page 14 line 3.*

L7 Costello and Schmidt . is not included within the reference list.

*Response:*
*We have added the citation to the reference list.*

*Changes made to manuscript:*
- *Citation added to Page 22 lines 8-10.*

L28-30: Omit.

*Response:*
*We appreciate your opinion and have omitted this sentence in the revision.*

*Changes made to manuscript:*
- *Sentence has been omitted from the manuscript, Page 14 line 27.*

**Page 14**
L12-13: Results and no discussion.

*Response:*
*These results are meant to introduce the idea that while some individual abiotic soil factors may be correlated with specific bacterial phylum abundance, predictors may vary depending on the organism.  This is discussed on Page 14 lines 28-30 and Page 15 lines 1-5.*

*Changes made to manuscript:*
- *$R^2$ and p-values have been removed in the revision, Page 15 lines 8, 9, and 11.*

L26 omit "(Stress : : :)

*Response:*
*It is the authors understanding that the stress value is a metric used to evaluate how well the ordination represents the original distances of the matrices, and should be reported when discussing the NMDS plot.*

*Changes made to manuscript:*
- *None.*

L28/29 Results and no discussion.

*Response:*
*These results highlight the relationship between the snow accumulation treatments and subsequent soil chemistry changes.  A possible mechanism for shifting bacterial community structure is outlined on Page 15 lines  29-31 and Page 16 lines 1-14.*

*Changes made to manuscript:*
- *None.*

**Page 15**
L1-2 Speculative. The authors did not measure enzyme kinetics. Omit "Rate" in "Rate of enzyme kinetics" since enzyme kinetics are the substrate-dependent rate of enzyme-substrate interaction.

*Response / Changes made to manuscript:*

- *We have modified the sentence as follows, "The initial effects of increased snow pack result in altered physical factors (greater active layer thaw depth and increased soil temperatures and moisture; Blanc-Betes et al., 2016) which may lead to increased SOM availability and faster enzyme activities with the potential to enhance SOM decomposition. Higher SOM mineralization may promote the documented shifts in aboveground plant communities and increased NPP (Natali et al., 2012; Sturm et al., 2005, Anderson-Smith 2013), and vegetation shifts to more shrubby species may alter the chemistry and quality of new litter inputs, ultimately affecting decomposer communities.." on Page 15 lines 29-31 and Page 16 lines 1-5.*

L7-8: Is this a logic relationship: increases in tannins and increases in N availability?

*Response:*
*The increases in tannins and N availability are meant to be two separate possible causes of reduced microbial activity in the DEEP zone, each of which are supported with references. However, in light of the comment above regarding insignificant difference in %N concentration (Results L30), we have removed the concept of increasing N from the sentence.*

*Changes made to manuscript:*
- *Alteration removing 'increased N availability" made to sentence, Page 16 line 10.*

L9 The decrease in C% might indicate exactly the opposite of the mechanism described above.

*Response:*
*To avoid confusion, we removed "%C and C:N" and replaced it with "relative abundance of genes required for SOM decomposition".*

*Changes made to manuscript:*
- *Above change made on Page 16 lines 10-11 of the revision.*

L18: Are any data available about fungal biomass? Is the microbial biomass dominated by bacterial or by fungal biomass? (Did you investigate effects of snow depth on microbial biomass?)

*Response:*
*Unfortunately, we did not collect fungal data. However, we felt it important to acknowledge the role that fungi play in SOM decomposition, and provide a reason why we did not find peroxides, phenol oxidases, and laccases in our data. This is discussed on Page 17 lines 3-5.*

*Changes made to manuscript:*
- *Slight alterations made to sentence, Page 17 lines 3-5.*

L23-27: Does an interaction between tannins and extracellular enzymes necessarily induce shifts in gene abundances? Statement requires a reference.

*Response / Changes made to manuscript:*
- *We have added the reference "(Rocca et al., 2014)" to Page 17 line 12 of the*

*revision.*

**Page 16**

L1-2: Does the decrease correspond to a decrease in C% or in C-stock or to a decrease in microbial biomass (as a factor for enzyme production)? Typically, enzyme activities are normalized with C%. Would this data treatment (gene abundances / C%) lead to a disappearance of treatment effects?

> *Response:*
> *Please see response to anonymous referee #1 general comment #2, regarding differences between %C and C-stock.*
>
> *Changes made to manuscript:*
> - *Please see changes from anonymous referee #1 general comment #2.*

L4-5 Sullivan 2008 is not included within the reference list

> *Response / Changes made to manuscript:*
> - *Thank you for catching that. We have added the citation to the reference list on Page 29 lines 1-3.*

L8-9: Is the system nutrient limited? Reference. N% increased in DEEP treatment - why did enzyme production not (that requires N and P)?

> *Response:*
> *Yes. Historically, the Arctic has been shown to be a nutrient limited ecosystem. See below for changes.*
> *%N concentration did not significantly increase in the DEEP treatment, possibly explaining lack of enzyme production. Also, the increase in temperature may lead to decreased enzyme gene copies without altering enzymatic capacity from decomposers as explained in discussion, Page 18 lines 13-24.*
>
> *Changes made to manuscript:*
> - *We added the following references (Hobbie et al., 2002; Jonasson et al., 1999; Mack et al., 2004; Shaver and Chapin, 1980, 1986; Sistla et al., 2012) to Page 17 lines 25-26*
> - *We added the following references to the References section, Page 24 lines 7-9, Page 24 lines 23-25, Page 25 lines 23-25, Page 28 lines 11-15, Page 28 lines 21-23.*

L12 Blanc-Betes et al. Submitted: not included in reference list

> *Response / Changes made to manuscript:*
> - *We removed this citation from the revision, Page 17 line 30.*
> - *However, it does occur throughout the manuscript, so we have added the citation to the reference list, Page 21 lines 18-20.*

L19: Which alternate energetic pathway?

> *Response:*
> *There are many possibilities, including fermentation, anaerobic respiration, and*

*chemolithotrophy. The main point behind this sentence is to establish the idea that alternate and less efficient forms of metabolism may be selected for under these conditions.*

*Changes made to manuscript:*
- *For clarification, the sentence has been modified as follows,"may select for microorganisms that use anaerobic metabolic pathways such as methanogenesis (Blanc-Betes et al. 2016). These hypoxic soil conditions would limit aerobic decomposition." on Page 18 lines 4-6.*

L21 change to "genes encoding enzymes involved in organic N degradation"

*Response / Changes made to manuscript:*
- *This change has been made in the revision as follows: "genes encoding for enzymes involved in N mobilization," on Page 18 lines 6-7*

L22: Microbial biomass was frequently reported to be positively related to enzyme production and decomposition. Please, describe more precisely under which circumstances an in N availability and microbial biomass results in a decrease in decomposition rate.

*Response:*
*Bacteria in a N limited system must decompose SOM to gain access to more N, potentially increasing decomposition rates. As enzyme production for SOM decomposition is energetically demanding, there is a threshold (high microbial biomass and alleviated N limitation) where bacteria may switch to alternate sources of N, such as microbial biomass, resulting in a decrease in SOM decomposition. This is supported by a theoretical C/N limitation model developed by Schimel, 2003. Also, In the Arctic, soil moisture is a confounding factor, as increased soil moisture may also decrease decomposition, as may be the case with the snow addition treatments.*

*Changes made to manuscript:*
- *We have modified the text to better reflect these interactions on Page 18 lines 1-12.*

**Page 17**
L6 needs a reference - for example: Razavi et al. Front. Microbiol., 14 October 2015 | http://dx.doi.org/10.3389/fmicb.2015.01126

*Response / Changes made to manuscript:*
- *This citation has been added as suggested, Page 18 line 19.*
- *Citation added to Reference list, Page 27 lines 8-10.*

L7-8 Speculative. If insitu substrate availability is low, Vmax will not be reached and enzyme functioning is controlled by Km (Michaelis-Menten constant). Alternatively, authors may change to "reach the same catalytic rate".

*Response:*
*We agree. Thank you clarifying this!*

*Changes made to manuscript:*
- *The text has been updated to clarify these points on Page 18 lines 17-24.*

L26: needs reference

*Response / Changes made to manuscript:*
- *This sentence has been removed from Page 18 line28 of the revised manuscript.*

L26-27 This statement should be reformulated in order to account for the low number of observed effects (the low number of repetitions) and the revised results of the KW-ANOVA (separate data from "organic" from those of "mineral soil").

*Response / Changes made to manuscript:*
- *We added "From the results of our study" to Page 18 line 28.*
- *We added "in the organic soil layers" to Page 18 line 30.*

**References**
IPCC should be shifted to I and not to W.(?)

*Response / Changes made to manuscript:*
- *We moved the citation in the revision as suggested*, from Page 30 line 2 to Page 24 line 20.

**Table 1:**
Describe in Methods how many repetitions and analytical replicates were used for the data. a> Or < b, please add information. No effect on N% - that differs from the description in the results section Why is there no effect on C:N in Table 1?

*Response / Changes made to manuscript:*
- *We altered Page 7 lines 5-12 of the revision as follows: "Organic samples were collected just below where plant tissue transitioned into dark brown/black soil (mean soil depth = 5.6±1.3cm; CTL n=4, DEEP n=4, INT n=3, LOW n=4), transitional samples were taken from the visual border between organic and mineral horizons based on change in soil colour (mean soil depth = 14.8±1.8cm; CTL n=3, DEEP n=3, INT n=4, LOW n=3), and mineral samples were collected 10cm below this transition (mean soil depth = 25.1±1.7cm; CTL n=3, DEEP n=4, INT n=3, LOW n=3), totalling 41 samples. To maintain consistency, only these samples were used to analyse %C, %N, and pH relationships."*
- *We added sample sizes to Table 1 in the revision, Page 32 line 7.*
- *We altered Page 10 lines 7-8 of the revision as follows: "...while the %N concentration only slightly increased (LOW/DEEP - p=0.32)."*
- *While it might seem from the numbers that there was a significant treatment effect on C:N, the p-value was 0.14. Please see Page 10 line 8 of the revision.*

**Figure 2:**
remove the ellipses from the graph. I counted 11 triangles but 10 circles etc. Why? How many sample points were included within the NMDS? NMDS requires detailed description within the results section. What are the main gradients observed? The NMDS optimized the illustration of the dissimilarity in beta diversity data but not in explanatory variables. Therefore, the combined illustration is somehow misleading (but the Mantel test is the appropriate method of choice).

*Response:*
- *We have decided to keep the ellipses to better visualize the separation (or lack*

*thereof) between the treatments.*

*Changes made to manuscript:*
- *Please see response above for Table 1. Alterations were made in the Methods of the revision to clarify replicates and sample sizes on Page 7 lines 5-12, and Page 32 line 7.*
- *We altered the Figure 2 caption Page 34 lines 3-5 as follows "Each point represents the bacterial community structure within one of the 41 total samples used for DNA extraction from all soil depths (Organic, Transition, and Mineral)."*

**Figure 3:**
No clear effect on bacterial phyla in organic samples (only some tendencies). Mineral soil: effect on Verrucmicrobia.

**Figs 3 and 4:**
I would prefer to change the order of treatments within the graph (from low (left) to deep (right) and the use of the same scale (y-axis) for all panels.) Add information about the treatments to Figure captions. Fig 4 Superoxides +2 to -3% difference to control? This might be a very low effect.

*Response:*
*While we appreciate your opinion, we have decided not to change the order of the treatments. Also, we initially attempted using the same scale on the y-axis for all panels, however it resulted in loss of visual interpretation. Significant differences and trends became unnoticeable. We decided not to change the y-axis scales.*

*Changes made to manuscript:*
- *We added the phrase "snow accumulation treatment" to the caption of Figure 3, Page 35 line 3.*

The manuscript by Ricketts et al. addresses the effect of climate change predictions on the soil bacterial communities in Arctic tundra soils which are important global C sinks. The experiment was carried out in a long-term experimental field site offering a snow depth gradient from 25% lower to 100% more snow than the surrounding (control). Due to increasing snow depth significant changes in abiotic soil parameters (e.g. active layer thaw depth, T, C/N ratio) were observed as well as a shift in the bacterial community structure. The taxonomic information from 16S rRNA amplicon sequence data was further used to estimate the functional gene abundance. These results indicate a decreased SOM decomposition potential under predicted climate change conditions which might help to further optimize current climate models. Therefore the study is of interest for readers of SOIL journal. Authors give a good overview about the current literature and make the aim of the study clear. In general, the manuscript is well-written and follows a logic flow. However, there are some points which were not clear to me and should be addressed before publication.

> *Response: We thank the reviewer for the summary and positive comments on the aim and approach of the study. We hope to address each of the comments in the revision to produce the best paper possible.*

The prediction of microbial functional composition from phylogeny is really advanced and delivers new insights for studies where only 16S rRNA genes were sequenced. However, I would have liked to have first more information on taxonomic composition of bacterial communities in soil treatments and not only tests on six dominant phyla. The response of bacterial taxa belonging to the same phylum might be completely different. I recommend adding a table of significantly treatment-responding genera (some were already mentioned, p.12 section 4.1) or a heatmap showing relative abundance of dominant OTUs or genera across all samples.

> *Response:*
> *There were 462 significant treatment responding Genera.*
>
> *Changes made to manuscript:*
> *- Heatmaps have been included in the supplementary information, Pages 13-17.*

Furthermore, it should be emphasized in the discussion that all functional gene abundances are based only on predictions not taking into account horizontal gene transfer that might decouple function from phylogeny. Furthermore, there might be a lot of unknown functions due to poorly characterized taxa in Tundra soils or not yet known links between taxon and function which should be discussed.

*Response:*
*This is a great comment and has been incorporated into the discussion of the revision.*

*Changes made to manuscript:*
- *Added, "While this method does not provide direct measurements of gene abundance (e.g. does not account for horizontal gene transfer or unknown functional / taxonomic linkages that may exist in the sampled tundra soils), it does offer valuable insights into the functional capacities of bacterial communities using 16S rRNA data (Langille et al., 2013)." Page 16 lines 20-23 of the revision.*

By the way, it would be also interesting to know the amount of unclassified bacteria in your samples.

*Response:*
*Due to the nature of closed OTU picking (which is required for PICRUSt analysis), there were no unclassified bacteria in our samples. As per the QIIME website (http://qiime.org/tutorials/otu_picking.html), "In a closed-reference OTU picking process, reads are clustered against a reference sequence collection and any reads which do not hit a sequence in the reference sequence collection are excluded from downstream analyses." However, to satisfy your curiosity, we did run open OTU picking, in which "reads are clustered against a reference sequence collection and any reads which do not hit the reference sequence collection are subsequently clustered de novo." (QIIME website). This resulted in 1.2% of unassigned bacteria. We did not use or include this method in the study as to maintain consistency between the bacterial abundance results and the PICRUSt functional gene abundance results.*

*Changes made to manuscript:*
- *To clarify this in the revision we added, "Any read that did not match a sequence in the reference database was discarded." to Page 7 line 29.*

In this respect, I am also wondering whether you tested first for availability of nearby genome representatives for your dataset before using PICRUSt prediction (NSTI index)? Furthermore, PICRUSt outputs a gene potential and it remains unknown to which extent these genes are expressed in the end.

*Response:*
*Please see response directly above and discussion on Page 16 lines 24-28.*

*Changes made to manuscript:*
*- Sentences revised and moved from Page 18 line 24 to Page 16 lines 24-28.*

As far as I understood, there are no treatment repetitions at the experimental site available and per treatment only 3 pseudo-replicates were taken. This makes it difficult to exclude the effect of natural variation in the bacterial community composition between sampling points. Thus conclusions can be drawn only very carefully and you should avoid to speculate too much in the Discussion section.

*Response:*
*We acknowledge the natural variation in bacterial communities at the micro-scale, and our limited replication. Through our revisions, we hope that our discussion and*

*conclusions do not overstep their bounds.*

These are my specific comments:
**Abstract:**
p.2, l.4 – Does "more or less snow" mean that predictions on amount of precipitation are not sure yet? Please consider rephrasing.

> *Response / Changes made to manuscript:*
> - *We omitted "or less" on Page 2 line 4 in the revision.*

p.2, l.11 – I recommend writing "Microbial community DNA was extracted from soil".

> *Response / Changes made to manuscript:*
> - *We added "Soil microbial community" to Page 2 lines 11-12 in the revision.*

p.2, l. 15 – Taxonomic names should be written in italics (throughout the manuscript).

> *Response:*
> *We kindly disagree. As a rule, only genus and species taxonomic levels are italicized. All names of taxonomic levels above genus are capitalized, but not italicized.*
>
> *Changes made to manuscript:*
> - *None.*

**Introduction:**
p.3, l.3 – Do you refer to belowground "microbial" community structure?

> *Response:*
> *This is meant to be a general statement. While there is evidence in the literature suggesting belowground macro community shifts, the citations provided primarily refer to microbial community shifts.*
>
> *Changes made to manuscript:*
> - *None.*

p.5, l.1 + l.16 – Please change microbial into bacterial community.

> *Response:*
> *The word "microbial" has been changed to "bacterial" where appropriate throughout the revision of the manuscript.*
>
> *Changes made to manuscript:*
> - *Page 4 line 30*
> - *Page 5 lines 4, 8, 11*
> - *Page 12 lines 26*
> - *Page 15 linse 18 & 27*
> - *Page 16 line 10*
> - *Change to Page 17 line 21*
> - *Change to Page 33, Figure 2, line 4*

p.5, l.14 – bacterial functions (plural).

> *Response / Changes made to manuscript:*
> - *This has been changed in the revision as suggested on Page 5 line 9.*

**Methods:**
p.6, l.1 – How far away from the snow fence was the CTL sampled?

> *Response / Changes made to manuscript:*
> - *We added ">30m" to Page 5 line 28 in the revision.*

p.6, l.9 – What was the distance between the replicates per treatment?

> *Response / Changes made to manuscript:*
> - *We added "approximately 15-20m apart" to Page 6 line 7 in the revision.*

p.7, l.4 – I couldn't find the results of the bacterial communities from the transition zone; and was DNA extracted from all three replicated cores per treatment?

> *Response:*
> *Only phyla from the organic and mineral horizons were reported for bacterial abundance and gene abundance, while all samples (including transition) were included in the distance matrices, NMDS ordinations, and accompanying statistics.*
> *Yes, DNA was extracted from all three cores per treatment and per soil horizon. We hope the changes below will clarify these points.*
>
> *Changes made to manuscript:*
> - *Text was altered as follows, "Due to significant differences between soil layers (Table S2), each layer was analysed separately. Only organic and mineral layers are reported." to Page 8 line 29 and Page 9 lines lines 1-2 in the revision.*
> - *We significantly altered text from Page 9 lines 12-18 of the revision.*
> - *We added "of each soil core," to Page 7 line 2 in the revision.*

p.7, l.16- Could you please add the reference for the primers?

> *Response / Changes made to manuscript:*
> - *Yes! We have added the appropriate reference to Page 7 line 16 of the revision.*
> - *Added to reference list, Page 21 lines 30-33.*

p.7, l.17 delete – in Mi-Seq but add to p.8, l.7 Bray- Curtis.

> *Response:*
> *Thank you! We have made these correction in the revision.*
>
> *Changes made to manuscript:*
> - *Deleted "-" in MiSeq, Page 7 line 18.*
> - *Added "-" in Bray-Curtis, Page 8 lines 7-8.*

p.8, l.6- Why did you determine adequate sampling depth? – I could not find that result later on.

*Response / Changes made to manuscript:*
- *We removed, "rarefaction curves to determine adequate sampling depth," from Page 8 line 7 of the revision.*

p.7, l.25/26 (and following pages)– "enzyme gene abundance" does not exist - please consider rephrasing, e.g. " relative abundance of bacterial 16S rRNA and functional genes".

*Response / Changes made to manuscript:*
- *We have reworded this in the revision as suggested, Page 8 lines 27-28 and Page 9 lines 2-3.*

p.8, l.27 – later there is also a significance threshold of p<0.1 used.

*Response:*
*Yes. We used a higher threshold in the figures to acknowledge and indicate where "marginal" significance was found.*

*Changes made to manuscript:*
- *None.*

**Results:**
I assume that the main focus here is the comparison of each treatment (LOW, INT, DEEP) to the control plot. However, sometimes this is not clear to me from the type of statistical tests you did and from the description of the results. I recommend in general spending some more sentences to explain your results. You often miss to look at the n-fold change of rel. abundance to the control or the comparison between organic and mineral layer. If there is a significant difference to the control- was it observed in all treatments? Is it the same change (decrease/increase) in all treatments?
e.g. p.10, l.3 – Instead of saying that the "C/N ratios became more similar" I would point out the %N trend is opposed in the organic and mineral layer. Furthermore I would suggest to add that differences in treatments are in general smaller in the mineral layer compared to organic layer.

*Response:*
*We have hopefully addressed these issues throughout the results in the revision.*

*Changes made to manuscript:*
- *We added n-fold changes to all appropriate results in the revision, from Page 10 line 19 –to- Page 12 line 11.*
- *The manuscript was revised as follows, "The changes in the mineral soil layers were less pronounced than in the organic layers. C:N ratios again showed a decreasing trend as snow accumulation increased, while soil pH increased in the DEEP zone but did not show a trend along the treatment gradient (Table 1)." on Page 10 lines 12-15.*

p.10, l.6 – delete "and".

*Response / Changes made to manuscript:*
- *This sentence has been removed from the manuscript, Page 10 line 15.*

p.10, section 3.2. – Please point out the n-fold change of relative abundances. For instance, the change in sequences affiliated to Chloroflexi is much stronger than for Actinobacteria.

*Response / Changes made to manuscript:*
- *We added n-fold changes to all appropriate results in the revision, from Page 10 line 19 –to- Page 12 line 11.*

p.10, l.11 According to the text p=0.011 for Chloroflexi but there is only one * in Fig. 3.

*Response:*
*Thank you so much for catching that!*

*Changes made to manuscript:*
- *We have fixed Fig. 3 in the revision, Page 35 line 1.*

p.10, l.13 Please consider rephrasing. I guess this is what you want to say: "acidobacterial abundance was in all treatments (DEEP, INT, LOW) lower than in the control".

*Response / Changes made to manuscript:*
- *We have clarified this in the revision by adding, "Acidobacteria showed decreased abundance in all treatments relative to the CTL, with the DEEP zone exhibiting the largest difference with a 1.98-fold decrease (p=0.055; Fig. 3)." to Page 10 lines 22-24.*

p.10, l.19 please check p-value for Actinobacteria text vs. Fig. S4.

*Response / Changes made to manuscript:*
- *Yes! Thank you. p-value corrected to "p<0.001" on Page 10 line 30 of the revision.*

p.11, l.6/7 Were lignin, pectin and xylan degradation not significantly different to the CTL?

*Response:*
*No, they were only marginally significant. DEEP to CTL: lignin – p=0.119, pectin – p=0.100, and xylan – p=0.119.*

*Changes made to manuscript:*
- *None.*

p.11, l.14 Please rephrase – I agree it is also a decrease in genes coding for lignin degradative enzymes over the gradient but the scale differs and both LOW and INT have higher gene abundances compared to CTL in mineral layer in contrast to the organic layer.

*Response / Changes made to manuscript:*
- *To acknowledge this observation, we added the following sentence, "However, relative to the CTL, both INT and LOW lignin-degrading genes exhibited much greater abundances than they did in the organic horizon (Fig. 4)." on Page 12 lines 12-14 of the revision:*

p.11, l.18 – Are Figs. S7-S9, S11 needed since they are not mentioned in the text?

*Response:*
*We believe that while these results are not significant, it is still important to include them in the supplementary information in order to maintain transparency.*

*Changes made to manuscript:*
- *None.*

p.11, l.23-25 - Please move this sentence to discussion and refer to Table S1.

*Response / Changes made to manuscript:*
- *The "(Table S1)" reference has been added to Page 12 line 22 in the revision.*
- *The indicated sentence has been altered and moved to Page 17 lines 1-4.*

**Discussion:**
p.12, l.11 Please explain from which results the conclusion of "reduced SOM decomposition" was derived from!

*Response / Changes made to manuscript:*
- *To clarify this, we added the following sentence, "Our results indicate that increased snow pack reduced the abundance of genes associated with SOM decomposition in the organic soil layers, suggesting a reduced SOM decomposition potential." to the revision Page 13 lines 8-10:*

p.12, l/12/13 – I don't agree with explanation 1) since you did not find differences in soil moisture along the gradient.

*Response:*
*Respectfully, there are many other factors besides moisture that contribute to $O_2$ diffusion into the soil. In the organic horizons near the soil surface, compaction likely plays a role as well, especially in the DEEP zone where there is more snow pack. Also, no moisture differences were found because all measurements were close to 100% saturation, which may not reflect differences between submerged vs. non-submerged soil conditions. We have tried to clarify this in the revision.*

*Changes made to manuscript:*
- *None.*

p.12, l.30 – Actinomycetales are a bacterial order containing several taxa, thus please use plural.

*Response:*
*Good observation! We have changed the sentence to reflect the plural in the revision.*

*Changes made to manuscript:*
- *We replaced "has" with "have" in Page 13 line 28 of the revision.*

p.13, l.1 – Is the increase in Actinomycetales, that are linked to degradation of recalcitrant compounds, contradicting to your conclusions from functional predictions? Please discuss.

*Response:*
*The "stimulation of … recalcitrant C degradation" was meant to be linked to the "stimulation of ectomycorrhizal growth…".*

*Changes made to manuscript:*

- *To clarify this, we added "…which degrade…" and removed "…degradation" on Page 13 line 29 in the revision.*

p.13, l.19- I suggest to delete Koyama et al. reference here because this is a totally different experiment. Instead cite Fierer et al. 2007 who tried an ecological classification of soil bacteria.

*Response:*
*We appreciate your suggestion. However, we feel this study provides a good example of oligotrophic vs copiotrophic competition in a nutrient limited vs non-limited environments. The Fierer et al. 2007 study is discussed in the following paragraph, Page 14 lines 17-22.*

p.14, l.9 and following- I suggest to delete R2 and p-values from the discussion.

*Response / Changes made to manuscript:*
- *We removed the R2 and p-values as suggested in the revision, Page 15 lines 9-10, and 12.*

p.14, l.25 and following- I recommend to transfer results of Fig.2 to Results section. Regarding the replicate size of this study I suggest to be more careful here with conclusions.

*Response / Changes made to manuscript:*
- *The sentence referred to was replaced by "The NMDS plot (Fig. 2)…" on Page 15 lines 26-27 of the revision.*
- *It was revised as follows: "visualized by a NMDS plot of Bray-Curtis dissimilarity indices constructed from community matrices (Stress=0.090, Shepard plot non-metric $R^2$=0.992; Fig. 2) revealed significant differences in community structure between all samples (organic, transition, and mineral) associated with winter snow pack", and moved to the Results, Page 11 lines 14-17 of the revision.*

p.15, l.14-27- The statements about fungi and tannic compounds are too speculative since this can not be supported by data from your study. Instead I would like to have a discussion of PICRUSt limitations here. Why is there only a decrease in rel. abundance of functional genes- which genes might be increased?

*Response:*
*The role of fungi and tannic compounds, while outside the scope of the data collected in this study, may help explain our results. Therefore, we feel that this paragraph should remain. The limitations of PICRUSt are acknowledged on Page 16 lines 4-8.*

*Changes made to manuscript:*
- *None.*

**Tables & Figures:**
**Table 1**
I don't really understand the number of replicates you refer to here. n=4 are technical replicates? Significance was tested between treatments for each layer separately?

*Response / Changes made to manuscript:*
- *We have altered the table caption to clarify. Page 32 lines 1-6 of the revision*

- *Added specifics on the number of replicates throughout the Methods section, Page 7 lines 6-12.*

According to Methods part you measured temperature at 4 different depths but not at 12 cm.

*Response / Changes made to manuscript:*
- *This was our mistake and has been corrected in the revision, Page 6 line 26-27.*

Was there no post-hoc test done for %N, C/N, and pH?

*Response:*
*While the post-hoc test was done, they were not significantly different. Therefore, the subscripts were not used, as indicated by the revised caption.*

*Changes made to manuscript:*
- *We added, "Results are indicated by $_{a,b,c}$ only where p<0.05." on Page 32 line 6 of the revision.*

**Figure 2**
Difficult to understand. Microbial communities from how many replicates and layers are plotted here?
Please use "CTL" as abbreviation for control (similar to the text).

*Response / Changes made to manuscript:*
- *We altered the caption as follows: "Each point represents the bacterial community structure within one of the 41 total samples used for DNA extraction from all soil depths (Organic, Transition, and Mineral).", on Page 34 lines 3-5 in the revision.*

**Figure 3**
Please indicate for which significant effect you tested here.
I don't understand, why there was no post-hoc test performed for Acidobacteria or is there only a significant difference to the control and not between treatments. The same applies for **Fig. 4.**

*Response:*
*You are correct in interpreting the significant difference to be to the control and not between treatments. This is indicated at the end of each caption by the phrase "except where significant differences were to the control.", Page 35 line 7 and Page 36 line 8.*

*Changes made to manuscript:*
- *None.*

**Supplement Fig.S1-S15**
From your Methods section it is not clear to me whether you analyzed abiotic soil parameters in the same soil sample (depth) as the bacterial community composition.

*Response / Changes made to manuscript:*
- *To clarify this, we added the following sentence, "To ensure accurate comparisons, soil chemical properties were measured from the same samples that DNA was extracted from.", to Page 9 lines 8-9 of the revision.*

> *Response / Changes made to manuscript:*
> - *Thank you! This has been fixed in the revision, Page 30 line 2.*

[revised manuscript text omitted]

Michael Ricketts 3/27/2016 4:37 PM

mmeler 4/22/2016 8:13 PM

Michael Ricketts 4/23/2016 12:07 PM

mmeler 4/22/2016 8:13 PM

Michael Ricketts 3/27/2016 5:02 PM

mmeler 4/22/2016 8:14 PM

mmeler 4/22/2016 8:14 PM

Michael Ricketts 4/19/2016 9:06 PM

Michael Ricketts 2/24/2016 6:07 PM

fence manipulation experiment that mimics changes in winter precipitation by creating a gradient of snow depths from much deeper than ambient to shallower than ambient levels (Jones et al., 1998; Pattison and Welker, 2014; Welker et al., 2000). We postulated that increased soil thermal insulation from deeper winter snow accumulation would elicit bacterial community response via: 1) altered soil physical characteristics such as soil temperature, moisture, or $O_2$ availability, and 2) altered soil chemistry produced by increased microbial mineralization of SOM resulting in increased nutrient availability and changes in plant species composition and litter. Here we evaluated phylum level shifts in bacterial community phylogeny using 16S rRNA gene analysis and predicted bacterial functions using the program PICRUSt (Langille et al., 2013) to test whether increased snow accumulation and associated changes in soil conditions (warmer temperatures, altered plant inputs, and increased hypoxia) would cause shifts in bacterial community structure and functional potential that reflect increased SOM decomposition and nutrient mineralization.

**2    Methods**

**2.1    Site description and sample collection**

[revised manuscript text omitted]

While the alpha diversity of soil bacterial communities via the Shannon index did differ between soil layer, it did not differ between snow pack treatment zones. Also, it does not elucidate community structural or functional differences between samples, and it fails to distinguish shifts in genetic potential among treatments. In contrast, beta diversity analyses better revealed soil bacterial community responses to snow accumulation. Bacterial community structure significantly shifted between snow pack treatment zones at all soil depths / layers (Table 2). The NMDS plot (Fig. 2) shows bacterial community structures to be associated with the snow accumulation treatment as soil chemical properties changed (%C, %N, C:N, and pH), indicating that bacterial β-diversity may respond to indirect changes in soil chemistry in response to winter snow accumulation. The initial effects of increased snow pack result in altered physical factors (greater active layer thaw depth and increased soil temperatures and moisture; Blanc-Betes et al.,

Michael Ricketts 5/17/2016 8:50 PM

mmeler 4/22/2016 8:31 PM

Michael Ricketts 5/28/2016 11:16 PM

mmeler 4/22/2016 8:33 PM

Michael Ricketts 5/28/2016 11:18 PM

mmeler 4/22/2016 8:33 PM

Michael Ricketts 5/28/2016 11:18 PM

mmeler 4/22/2016 8:33 PM

Michael Ricketts 4/18/2016 3:54 PM

mmeler 4/22/2016 8:35 PM

Michael Ricketts 5/17/2016 8:58 PM

mmeler 4/22/2016 8:36 PM

Michael Ricketts 5/28/2016 11:42 PM

mmeler 4/22/2016 8:36 PM

Michael Ricketts 4/23/2016 1:34 PM

mmeler 4/22/2016 8:36 PM

Michael Ricketts 4/13/2016 7:58 PM

mmeler 4/22/2016 8:38 PM

Michael Ricketts 6/3/2016 3:26 PM

mmeler 4/22/2016 8:39 PM

Michael Ricketts 4/23/2016 1:35 PM

mmeler 4/22/2016 8:38 PM

Michael Ricketts 4/6/2016 7:26 PM

2016) which may lead to increased SOM availability and faster enzyme activities with the potential to enhance SOM decomposition. Higher SOM mineralization may promote the documented shifts in aboveground plant communities and increased NPP (Natali et al., 2012; Sturm et al., 2005, Anderson-Smith 2013), and vegetation shifts to more shrubby species may alter the chemistry and quality of new litter inputs, ultimately affecting decomposer communities. Moreover, soil moisture and compaction can reduce $O_2$ diffusion into the soil, inhibiting aerobic SOM decomposition (Blanc-Betes et al., 2016; O'Brien et al., 2010), and altering bacterial community composition by selecting for microorganisms that utilize simple C substrates, leaving behind complex organic compounds and plant polymers. In addition, tannins produced by expanding woody shrubs may act to inhibit microbial activity (Schimel et al., 1996), further slowing decomposition. This is supported by the lower relative abundance of genes required for SOM decomposition in the DEEP snow accumulation zone where we observed the most significant shifts in bacterial community composition (Figs. 3&4). The balance between these two competing processes, and the functional shifts associated with them, will ultimately influence the C balance of the system.

**4.2 Functional shifts**

To examine the influence of shifting bacterial abundances on soil community functioning and the C balance of Arctic ecosystems, we focused on the genetic potential of the bacterial community to produce enzymes required for the degradation of various forms of SOM. We did this by using PICRUSt software to estimate functional gene abundance via ancestral state reconstruction (Langille et al., 2013). While this method does not provide direct measurements of gene abundance (e.g. does not account for horizontal gene transfer or unknown functional / taxonomic linkages that may exist in the sampled tundra soils), it does offer valuable insights into the functional capacities of bacterial communities using 16S rRNA data (Langille et al., 2013). Furthermore, gene abundance in itself is not a direct measurement of gene expression or enzyme activity (Wood et al., 2015). However it does provide a measure of genetic potential and may be positively correlated to enzyme activity and gene expression (Morris et al., 2014; Neufeld et al., 2001; Rocca et al., 2014). To accurately measure enzymatic functional potential or gene expression would require a targeted metagenomic and metatranscriptomic approach.

mmeler 4/22/2016 8:40 PM

Michael Ricketts 4/6/2016 8:07 PM

mmeler 4/22/2016 8:42 PM

Michael Ricketts 4/6/2016 8:38 PM

mmeler 4/22/2016 8:43 PM

Michael Ricketts 2/24/2016 6:25 PM

mmeler 4/22/2016 8:44 PM

Michael Ricketts 4/23/2016 1:58 PM

Michael Ricketts 5/12/2016 2:06 PM

Michael Ricketts 5/12/2016 2:04 PM
**Moved (insertion) [1]**

Michael Ricketts 5/12/2016 2:20 PM

[revised manuscript text omitted]

Michael Ricketts 4/19/2016 8:54 PM
mmeler 4/22/2016 8:58 PM
Michael Ricketts 5/29/2016 12:51 AM
mmeler 4/22/2016 8:58 PM
Michael Ricketts 4/19/2016 8:57 PM
mmeler 4/22/2016 8:59 PM
Michael Ricketts 4/10/2016 8:13 PM
mmeler 4/22/2016 8:59 PM
Michael Ricketts 4/23/2016 2:26 PM
mmeler 4/22/2016 9:00 PM
Michael Ricketts 2/24/2016 6:27 PM
mmeler 4/22/2016 9:00 PM
Michael Ricketts 5/24/2016 1:22 PM
mmeler 4/22/2016 9:00 PM
Michael Ricketts 4/23/2016 2:26 PM
mmeler 4/22/2016 9:01 PM
Michael Ricketts 4/23/2016 2:26 PM
mmeler 4/22/2016 9:02 PM
Michael Ricketts 4/23/2016 2:26 PM
mmeler 4/22/2016 9:02 PM
Michael Ricketts 4/22/2016 9:59 PM
Formatted ... [38]
Michael Ricketts 4/23/2016 2:29 PM
Michael Ricketts 4/22/2016 9:59 PM
Formatted ... [39]
mmeler 4/22/2016 9:03 PM
Michael Ricketts 4/22/2016 9:59 PM
Formatted ... [40]
Michael Ricketts 4/19/2016 8:58 PM
mmeler 4/22/2016 9:03 PM
Michael Ricketts 5/24/2016 3:54 PM
mmeler 4/22/2016 9:04 PM
Michael Ricketts 5/26/2016 11:02 AM

mmeler 4/22/2016 9:04 PM
... [44]
Michael Ricketts 4/23/2016 2:32 PM
Formatted ... [45]
mmeler 4/22/2016 9:08 PM
... [46]
Michael Ricketts 5/24/2016 1:24 PM
... [47]
mmeler 4/22/2016 9:10 PM

Michael Ricketts 4/23/2016 2:46 PM

chemistry through selective decomposition remains unclear. From the results of our study, it is clear that increased snow accumulation may lead to changes in both bacterial community composition and SOM chemistry in the organic soil layers (Table 1 and Fig. 3). Unlike other ecosystems where plants are the first responders to abiotic climate change factors, in the Arctic, microbes are likely the first responders to changes in temperature by initially increasing nutrient mineralization. These released nutrients facilitate plant community shifts and increase ecosystem NPP (Chapin III et al., 1995). Over time, the combination of increased snow accumulation and soil compaction may lead to hypoxic/anaerobic soil conditions (e.g., Blanc-Betes et al., 2016) and further vegetative shifts to wet-sedge (*Carex)* species, limiting SOM decomposition. This in combination with a recent history of more recalcitrant plant litter inputs could result in re-accrual of SOC (e.g., Sistla et al., 2012), ultimately mitigating the positive feedback loop hypothesized in current literature (Davidson and Janssens, 2006; Natali et al., 2014; Schuur et al., 2009; Sturm et al., 2005).

**5    Conclusions**

The results presented here support the hypothesis that bacterial community structure and function shift as a result of consistently deepened snowpack. Increases in soil hypoxia under deepened snow may have resulted in an increased abundance of anaerobic or facultative bacteria, slowing decomposition. Decreases in PICRUSt predicted gene copies suggest that SOM decomposition may be slowed under accumulated snow, and bacterial community substrate preference may shift to more labile compounds. Concentrations of C and N, as opposed to C:N, better explained bacterial community responses to snow pack treatments. Together these results strongly suggest that soil decomposers of moist acidic tundra are key in determining the direction and magnitude of permafrost C feedbacks on the climate system.

**Author Contributions**

M. P. Ricketts, J. M. Welker, and M. A. Gonzalez-Meler designed the experiment. R. S. Poretsky provided expertise and insight into the bioinformatics and data analyses. M. P. Ricketts performed all sample collections, lab work, and data analyses. M. P. Ricketts prepared the manuscript with contributions from all co-authors.

**Data Availability**

All data used in this report are publically accessible through two separate data repositories. The 16S rRNA gene sequences derived from Illumina Mi-Seq sequencing have been deposited in the NCBI Sequence Read Archive (SRA) under accession number SRP068302. All computer script text files used in QIIME and R packages, as well as BIOM and Excel files, are available via the NSF Arctic Data Center, doi:10.18739/A2FS9J.

**Acknowledgements**

This research was supported by funding from the Department of Energy, Terrestrial Ecosystem Science Program to M. A. Gonzalez-Meler (DE-SC 0006607). J. M. Welker initiated and maintained the snow fence experiment with NSF funds beginning in 1994 (9412782, 0632184, 0612534, 0856728). Michael Ricketts received additional support from the Elmer Hadley Graduate Research Award, the W.C. and May Preble Deiss Award for Graduate Research, and the Bodmer International Travel Award. We would like to thank members of the Stable Isotope lab at the University of Illinois at Chicago (UIC), particularly Jessica Rucks and Gerard Aubanell for assistance with field work/sample collection and processing. We thank Elena Blanc-Betes, Douglas Johnston, Dr. Douglas Lynch, and Dr. Charlie Flower for advice on experimental design, analyses, and comments on the manuscript. We would also like to thank Dr. D'Arcy Meyer Dombard (Earth and Environmental Sciences department, UIC) for use of her lab and soil expertise, and Imrose Kauser (Microbial Ecology lab, UIC) for help with the bioinformatics. Finally we thank Olivia L Miller (Purdue University), Niccole Van Hoey (University of Alaska, Anchorage), all of the staff at Toolik Field Station, and field and lab UIC undergraduate assistants: Ben Thurnhoffer, Andres Davila, and Briana Certa.

Michael Ricketts 4/21/2016 12:54 PM

Michael Ricketts 5/29/2016 1:14 AM

Michael Ricketts 5/16/2016 4:29 PM

mmeler 4/22/2016 9:30 PM

mmeler 4/22/2016 9:30 PM

mmeler 4/22/2016 9:31 PM

mmeler 4/22/2016 9:31 PM

mmeler 4/22/2016 9:31 PM

mmeler 4/22/2016 9:32 PM

mmeler 4/22/2016 9:33 PM

mmeler 4/22/2016 9:32 PM

[revised manuscript text omitted]

Michael Ricketts 5/29/2016 9:57 PM
Michael Ricketts 5/31/2016 9:32 AM

Michael Ricketts 5/29/2016 10:01 PM

[Figure]

**Figures S1-to-S6.** Linear fit regressions of bacterial phyla abundance with respect to soil properties such as soil carbon-to-nitrogen ratio (C/N), soil carbon mass concentration (%C), soil nitrogen mass concentration (%N), and soil pH across all snow treatment sites and all soil depths. The bacterial relative abundance is ordered from most abundant phylum (Fig. S1) to least abundant (Fig S6). See table S2 for statistics and significance.

Fig. S1 – Linear relationships between Acidobacteria relative abundance and soil C:N, C and N concentrations, and pH from all 41 samples. Shaded areas indicate 95% confidence intervals.

Fig. S2 – Linear relationships between Proteobacteria relative abundance and soil C:N, C and N concentrations, and pH from all 41 samples. Shaded areas indicate 95% confidence intervals.

Michael Ricketts 5/31/2016 9:32 PM
Formatted
Gonzalez-Meier, Mi…, 5/31/2016 8:18 AM
Gonzalez-Meier, Mi…, 5/31/2016 8:14 AM
Formatted
Gonzalez-Meier, Mi…, 5/31/2016 8:15 AM
Gonzalez-Meier, Mi…, 5/31/2016 9:32 AM
Michael Ricketts 5/31/2016 9:32 AM
Formatted
Gonzalez-Meier, Mi…, 5/31/2016 9:32 AM
Gonzalez-Meier, Mi…, 5/31/2016 8:17 AM
Michael Ricketts 5/24/2016 10:10 PM
Gonzalez-Meier, Mi…, 5/31/2016 8:15 AM
Michael Ricketts 5/25/2016 11:59 AM
Gonzalez-Meier, Mi…, 5/31/2016 8:17 AM
Gonzalez-Meier, Mi…, 5/31/2016 9:32 AM
Michael Ricketts 5/31/2016 9:32 AM
Formatted
Gonzalez-Meier, Mi…, 5/31/2016 9:32 AM
Michael Ricketts 5/31/2016 9:32 AM
Gonzalez-Meier, Mi…, 5/31/2016 8:17 AM
Michael Ricketts 5/24/2016 9:56 PM
Moved down [1]: Fig. S1
Michael Ricketts 5/24/2016 9:56 PM
Moved (insertion) [1]
Michael Ricketts 5/31/2016 9:36 AM
Michael Ricketts 5/29/2016 9:58 PM
Michael Ricketts 5/24/2016 9:56 PM
Moved down [2]: Fig. S2
Michael Ricketts 5/24/2016 9:56 PM
Moved (insertion) [2]
Gonzalez-Meier, Mi…, 5/31/2016 8:25 AM
Michael Ricketts 5/29/2016 10:01 PM
Formatted

[Figure]

[Figure]

**Fig. S3** – Linear relationships between Verrucomicrobia relative abundance and soil C:N, C and N concentrations, and pH from all 41 samples. Shaded areas indicate 95% confidence intervals.

**Fig. S4** – Linear relationships between Actinobacteria relative abundance and soil C:N, C and N concentrations, and pH from all 41 samples. Shaded areas indicate 95% confidence intervals.

Michael Ricketts 5/24/2016 9:56 PM
**Moved down [3]:** Fig. S3

Michael Ricketts 5/29/2016 9:58 PM

Michael Ricketts 5/24/2016 9:56 PM
**Moved (insertion) [3]**

Gonzalez-Meier, Mi..., 5/31/2016 8:25 AM

Michael Ricketts 5/24/2016 9:58 PM

Michael Ricketts 5/24/2016 9:57 PM
**Moved down [4]:** Fig. S4

Michael Ricketts 5/24/2016 9:57 PM
**Moved (insertion) [4]**

Gonzalez-Meier, Mi..., 5/31/2016 8:25 AM

Michael Ricketts 5/31/2016 9:32 AM

Michael Ricketts 5/29/2016 10:01 PM
Fig. S5

[Figure]

[Figure]

Michael Ricketts 5/24/2016 9:57 PM
Moved down [5]: Fig. S5

Michael Ricketts 5/24/2016 9:57 PM
Moved (insertion) [5]

Gonzalez-Meier, Mi..., 5/31/2016 8:25 AM

Michael Ricketts 5/31/2016 9:32 AM

Michael Ricketts 5/24/2016 9:57 PM
Moved down [6]: Fig. S6

Michael Ricketts 5/24/2016 9:57 PM
Moved (insertion) [6]

Gonzalez-Meier, Mi..., 5/31/2016 8:25 AM

Michael Ricketts 5/29/2016 9:59 PM

Michael Ricketts 5/29/2016 10:01 PM

**Figures S7-to-S15.** Linear fit regressions of relative gene copy abundance with respect to soil properties such as soil carbon-to-nitrogen ratio (C/N), soil carbon mass concentration (%C), soil nitrogen mass concentration (%N), and soil pH across all snow treatment sites and all soil depths. Enzymes were grouped by their general activities as explained in main text and they are shown on the Y-axis of the figures. See table S2 for statistics and significance.

[Figure]

**Fig. S7** - Linear relationships between the relative abundance of genes required for arabinoside degradation and soil C:N, C and N concentrations, and pH from all 41 samples. Shaded areas indicate 95% confidence intervals.

[Figure]

**Fig. S8** - Linear relationships between the relative abundance of genes required for cellulose degradation and soil C:N, C and N concentrations, and pH from all 41 samples. Shaded areas indicate 95% confidence intervals.

Michael Ricketts 5/31/2016 9:40 AM
**Moved (insertion) [16]**
Michael Ricketts 5/31/2016 9:40 AM
Michael Ricketts 5/31/2016 9:32 AM
… [12]
Michael Ricketts 5/31/2016 9:57 PM
**Moved down [7]:** Fig. S7 .
Michael Ricketts 5/24/2016 9:57 PM
**Moved (insertion) [7]**
Michael Ricketts 5/31/2016 9:42 AM
Gonzalez-Meier, Mi..., 5/31/2016 8:25 AM
… [13]
Michael Ricketts 5/31/2016 9:40 AM
**Moved up [16]: Figures S8-to-S15.** Linear fit regressions of relative gene copy abundance with respect to soil properties such as soil carbon-to-nitrogen ratio (C/N), soil carbon mass concentration (%C), soil nitrogen mass concentration (%N), and soil pH across all snow treatment sites and all soil depths. Enzymes were grouped by their gener... [14]
Michael Ricketts 5/31/2016 10:31 AM
Michael Ricketts 5/24/2016 9:57 PM
**Moved down [8]:** Fig. S8 .
Michael Ricketts 5/24/2016 9:57 PM
**Moved (insertion) [8]**
Gonzalez-Meier, Mi..., 5/31/2016 8:25 AM
Gonzalez-Meier, Mi..., 5/31/2016 8:22 AM
Michael Ricketts 5/29/2016 9:58 PM
Michael Ricketts 5/29/2016 10:01 PM
… [15]

[Figure]

[Figure]

**Fig. S9** – Linear relationships between the relative abundance of genes required for chitin degradation and soil C:N, C and N concentrations, and pH from all 41 samples. Shaded areas indicate 95% confidence intervals.

**Fig. S10** – Linear relationships between the relative abundance of genes required for lignin degradation and soil C:N, C and N concentrations, and pH from all 41 samples. Shaded areas indicate 95% confidence intervals.

Michael Ricketts 5/24/2016 9:57 PM
**Moved down [9]:** Fig. S9

Michael Ricketts 5/24/2016 9:57 PM
**Moved (insertion) [9]**

Gonzalez-Meier, Mi..., 5/31/2016 8:25 AM

Michael Ricketts 5/29/2016 9:59 PM

Michael Ricketts 5/24/2016 9:58 PM
**Moved down [10]:** Fig. S10.

Michael Ricketts 5/24/2016 9:58 PM
**Moved (insertion) [10]**

Gonzalez-Meier, Mi..., 5/31/2016 8:25 AM

Michael Ricketts 5/29/2016 10:01 PM

[Figure]

[Figure]

**Fig. S11** – Linear relationships between the relative abundance of genes required for N mobilization and soil C:N, C and N concentrations, and pH from all 41 samples. Shaded areas indicate 95% confidence intervals.

[Figure]

**Fig. S12** – Linear relationships between the relative abundance of genes required for pectin degradation and soil C:N, C and N concentrations, and pH from all 41 samples. Shaded areas indicate 95% confidence intervals.

[Figure]

Michael Ricketts 5/24/2016 9:58 PM
**Moved down [11]:** Fig. S11
Michael Ricketts 5/29/2016 9:59 PM

Michael Ricketts 5/24/2016 9:58 PM
**Moved (insertion) [11]**
Gonzalez-Meier, Mi…, 5/31/2016 8:25 AM
Michael Ricketts 5/29/2016 10:00 PM
Michael Ricketts 5/24/2016 9:58 PM
**Moved down [12]:** Fig. S12

Michael Ricketts 5/24/2016 9:58 PM
**Moved (insertion) [12]**
Gonzalez-Meier, Mi…, 5/31/2016 8:25 AM
Michael Ricketts 5/29/2016 10:00 PM
Fig. S13
Michael Ricketts 5/29/2016 10:01 PM

[Figure]

[Figure]

Fig. S13 – Linear relationships between the relative abundance of genes required for P mobilization and soil C:N, C and N concentrations, and pH from all 41 samples. Shaded areas indicate 95% confidence intervals.

Fig. S14 – Linear relationships between the relative abundance of genes required for superoxide regulation and soil C:N, C and N concentrations, and pH from all 41 samples. Shaded areas indicate 95% confidence intervals.

Michael Ricketts 5/24/2016 9:58 PM
Moved down [13]: Fig. S13

Michael Ricketts 5/24/2016 9:58 PM
Moved (insertion) [13]

Gonzalez-Meier, Mi..., 5/31/2016 8:25 AM

Michael Ricketts 5/29/2016 10:00 PM

Michael Ricketts 5/24/2016 9:58 PM
Moved down [14]: Fig. S14

Michael Ricketts 5/24/2016 9:58 PM
Moved (insertion) [14]

Gonzalez-Meier, Mi..., 5/31/2016 8:25 AM

Michael Ricketts 5/29/2016 10:01 PM

[Figure]

Michael Ricketts 5/24/2016 9:58 PM
**Moved down [15]:** Fig. S15.
Michael Ricketts 5/29/2016 10:00 PM
… [18]

3

Fig. S15 – Linear relationships between the relative abundance of genes required for xylan degradation and soil C:N, C and N concentrations, and pH from all 41 samples. Shaded areas indicate 95% confidence intervals.

Michael Ricketts 5/24/2016 9:58 PM
**Moved (insertion) [15]**
Michael Ricketts 5/25/2016 12:09 PM
Gonzalez-Meler, Mi..., 5/31/2016 8:25 AM
Michael Ricketts 5/25/2016 12:09 PM

Michael Ricketts 5/29/2016 10:01 PM

Table S2 – Effects of soil depth characteristics (Organic, Transition, Mineral) on soil chemistry, bacterial phylum relative abundance, and relative abundance of genes organized by functional groups, as determined by the Kruskal-Wallis test. Degrees of freedom = 2 for all analyses.   * = p<0.05, ** = p<0.01, *** = p<0.001

| | H-statistic | p-value | Soil layer w/ higher value (Org vs Min) |
|---|---|---|---|
| **Soil Chemistry** | | | |
| %C | 32.32 | $9.57 \times 10^{-8}$ *** | Organic |
| %N | 26.53 | $1.74 \times 10^{-6}$ *** | Organic |
| C:N | 21.70 | $1.94 \times 10^{-5}$ *** | Organic |
| pH | 6.81 | 0.03 * | Mineral |
| | | | |
| **Bacterial phylum abundance** | | | |
| Acidobacteria | 0.05 | 0.98 | NA |
| Proteobacteria | 14.78 | $6.17 \times 10^{-4}$ *** | Organic |
| Verrucomicrobia | 14.93 | $5.73 \times 10^{-4}$ *** | Organic |
| Actinobacteria | 20.16 | $4.12 \times 10^{-5}$ *** | Mineral |
| Bacteroidetes | 13.08 | $1.45 \times 10^{-3}$ ** | Organic |
| Chloroflexi | 24.80 | $4.13 \times 10^{-6}$ *** | Mineral |
| | | | |
| **Enzyme gene abundance** | | | |
| Lignin | 20.17 | $4.17 \times 10^{-5}$ *** | Organic |
| Chitin | 3.00 | 0.22 | NA |
| Cellulose | 8.36 | 0.02 * | Organic |
| Pectin | 17.25 | $1.80 \times 10^{-4}$ *** | Organic |
| Xylan | 15.79 | $3.72 \times 10^{-4}$ *** | Organic |
| Arabinoside | 13.62 | $1.10 \times 10^{-3}$ ** | Organic |
| N mobilization | 11.67 | $2.92 \times 10^{-3}$ ** | Organic |
| P mobilization | 3.30 | 0.19 | NA |
| Superoxide | 29.61 | $3.72 \times 10^{-7}$ *** | Organic |

Table S3 – Statistical analyses of alpha diversity using non-parametric two-sample t-tests with 999 Monte Carlo permutations. Pairwise comparisons of Shannon diversity metrics from each sample were made between each soil layer (Organic, Transition, Mineral) and each snow accumulation treatment (Control, Deep, Intermediate, Low).
* = p<0.05, ** = p<0.01, *** = p<0.001.

| Two-sample t-tests | | |
| --- | --- | --- |
| Soil layers | t- statistic | p-value |
| Organic  / Mineral | 5.58 | 0.003 ** |
| Trans / Organic | -0.26 | 1 |
| Trans / Mineral | 5.22 | 0.003** |
| | | |
| Treatment | | |
| Cont / Deep | 0.30 | 1 |
| Cont/ Int | -0.49 | 1 |
| Cont / Low | 1.70 | 0.6 |
| Deep / Int | $-4.0 \times 10^{-4}$ | 1 |
| Deep / Low | -.30 | 1 |
| Int / Low | 0.51 | 1 |

Michael Ricketts 5/29/2016 10:01 PM

**Figures S16-S20.** Heatmaps showing the relative raw abundance of operational taxonomic units (OTU) at different taxonomic levels (phyla, class, family, individual OTU) for each of the snow treatments: control (CTL), 100% more snow accumulation (DEEP), 50% more snow (INT), and 25% less snow than control (LOW).

[Figure]

Fig. S16 – Heatmap of raw gene abundance (# of OTU's) for all detected Phyla. Columns represent snow accumulation treatment groups.

[Figure]

Michael Ricketts 5/25/2016 12:14 PM

Fig. S17 - Heatmap of raw gene abundance (# of OTU's) for all detected Classes.  Columns represent snow accumulation treatment groups.

Michael Ricketts 5/29/2016 10:01 PM

[Figure]

Fig. S18 - Heatmap of raw gene abundance (# of OTU's) for all detected Orders. Columns represent snow accumulation treatment groups.

Michael Ricketts 5/29/2016 10:01 PM

[Figure]

Fig. S19 - Heatmap of raw gene abundance (# of OTU's) for all detected Families. Columns represent snow accumulation treatment groups.

Michael Ricketts 5/29/2016 10:01 PM

[Figure]

Fig. S20 - Heatmap of raw gene abundance (# of OTU's) for individual OTU assignments. Columns represent snow accumulation treatment groups.

Michael Ricketts 5/29/2016 10:01 PM

---

## Author Response (AR2)

**Topical editor Decision: Reconsider after minor revisions (review by Editor)** (24 Jun 2016)
by Karsten Kalbitz
Comments to the Author:
Dear authors,

Thank you very much for the revision of your paper. I have just three comments:

> *You are very welcome! Thank you for taking the time to edit our paper. We are confident that all of the comments and changes has succeeded in making the manuscript better. We hope we have adequately addressed the following comments as well.*

Please explain the acronyms of the snow treatments in the abstract as well (p. 2, lines 9-10).

> *We have added the requested explanations to the abstract Page 2, lines 9-11 as follows: "where Deep ~100% and Int. ~50% increased snow pack relative to the control, and Low ~25% decreased snow pack relative to the control."*

> There is a typo at page 9, line 28.

> *We have fixed the typo "orgaic" to "organic" on Page 9 line 28.*

At p. 15, lines 7-8 you stated that physical protection by soil aggregates has not been observed in Arctic soils. I would question that in mineral soil horizon.

> *Thank you for your comment. However, while it is common for aggregation and organomineral associations to protect organic matter in most soils, this may not be the case in Arctic permafrost soils. This statement is supported by correspondences with colleagues who also work in these Alaskan tundra soils. We have altered the text and provided references to address and support this uncommon soil trait, Page 15, lines 7-9.*

Furthermore, interactions of organic matter with soil minerals might play a role as well for C protection in mineral soil horizons (e.g. papers by the group of G. Guggenberger). You did not analyze these factors. Please change that appropriately.

> *See reply to comment above.*

I would strongly suggest to add the depths of the organic and mineral soil horizons to table 1 as well (not just in the text). Then, a reader can link the different properties easily to the different depths.

> *We have added a column indicating average soil depths ± standard error to Table 1. We did have to change the page orientation in order to fit the additional column. We hope this is acceptable for publication.*

Thank you very much and I look forward to your (finally) revised paper.
Karsten Kalbitz

*We truly appreciate the time and energy put into this manuscript and thank you again for your comments. Additional changes we thought might also help improve the paper include the following:*

1) *In order to maintain consistency throughout the text and figures when referring to the treatment groups, we changed all text to "Control", "Deep", "Int.", and "Low". These are established from Page 5, line 28 – Page 6, line 1, and have been subsequently changed throughout the manuscript.*

2) *We updated the DOI for the data archive to the number given to us after submitting the updated files, Page 20, line 12.*

[revised manuscript text omitted]